# Metric Automata Theory:
# A Unifying Theory of RNNs

**Adam Dankowiakowski**
University of Oxford
adankowiakowski02@gmail.com

**Alessandro Ronca**
IRIS-AI
alessandro.ronca@iris-ai.org

## Abstract

We propose *Metric Automata Theory*, an elegant generalisation of classic Automata Theory to continuous dynamical systems, that constitutes a unifying theory of all kinds of Recurrent Neural Networks (RNNs), including widely-adopted architectures such as xLSTM and State Space Models (SSMs). The theory allows one to analyse RNNs both in the finite and unbounded precision settings seamlessly, while utilising fundamental results of Automata Theory. It also provides a novel notion of *robustness* that guarantees numerical stability and contributes to stability of learning. We employ the theory to prove a comprehensive set of expressivity results for widely-adopted RNNs, with a focus on robustness and finite-precision. Notably, we contrast the capabilities of xLSTM and SSMs for *robustly* modelling all star-free regular languages—xLSTM can do so, while SSMs cannot robustly recognize the FLIP-FLOP language. Thus we give a novel perspective on the importance of non-linear recurrences, giving insight for why xLSTM shows superior performance to SSMs on several tasks. We provide an improved understanding of the capabilities of Mamba, a popular SSM model. We show that Mamba is not generally capable of recognising the star-free languages under finite-precision, which is seemingly in contrast with the existing theoretical and empirical results for SSMs. We clarify the picture, by showing that Mamba admits a piecewise-linearly separable state space that allows it to approximate star-free languages, with *some* length-generalisation abilities. At the same time, Mamba does not admit such state spaces for languages like Parity. This explains why empirically Mamba performs well on star-free languages, and fails on Parity.

## 1 Introduction

Recurrent Neural Networks (RNNs) encompass all the neural networks that process sequences by maintaining a *state* though some form of *recurrence*. Notable RNNs are Vanilla RNNs such as Elman-RNNs [Elman, 1990], LSTM [Hochreiter and Schmidhuber, 1997], GRU [Cho et al., 2014], and the more recent and now widely-adopted xLSTM [Beck et al., 2024] and the family of State Space Models (SSMs) including S4 [Gu et al., 2022], Mamba [Gu and Dao, 2023], HiPPO [Gu et al., 2020], and DeltaNet [Yang et al., 2024]. The more recent RNNs achieve state-of-the-art performance, comparable to other notable neural networks such as Transformers [Vaswani et al., 2017], by leveraging new design principles that overcome the limitations of previous RNNs, enabling key properties such as parallel training to take full advantage of modern computer architectures.

Recently, there has been an increasing interested in developing a systematic understanding of the capabilities of sequence models, including RNNs and Transformers, beyond empirical evidence. As of now, there is a rich literature of formal results regarding the expressivity of both RNN (cf. [Knorozova and Ronca, 2024a,b, Weiss et al., 2018, Merrill et al., 2020]) and Transformers (cf. [Strobl et al., 2024, Merrill and Sabharwal, 2024, Hahn, 2020]), with some studies directly comparing the two, e.g., [Bhattamishra et al., 2024]. We focus on modelling RNN expressivity in terms of formal

languages—an active area, with big impact on the new directions of research for novel architectures. For instance, Sarrof et al. [2024] showed that a family of SSM models, including Mamba [Gu and Dao, 2023], has expressivity restricted to *star-free* regular languages in the finite precision setting, due to restricting the eigenvalues of the state-update gates to be non-negative. Soon after, Grazzi et al. [2025] extended the capabilities of SSMs beyond star-free languages, by modifying the implementation of Mamba and DeltaNet models to allow negative eigenvalues—narrowing the gap between SSMs and LSTM models.

The main limitation of this recent literature is that it lacks a principled, commonly-accepted theory or framework providing the a established setting for investigations. For instance, both Sarrof et al. [2024] and Grazzi et al. [2025] provide similar arguments proving that under finite-precision, SSMs with gates with non-negative eigenvalues are restricted to star-free languages (with Sarrof et al. [2024] proving a special case). However, the details of the finite-precision frameworks used by the two are completely different, and result in different assumptions. This means that the results are hard to compare without carefully assessing the assumptions and inspecting the proofs.

We propose *Metric Automata Theory* (MAT) as an elegant and principled theory that generalises Automata Theory to continuous dynamical systems, with RNNs being a special case of particular interest. It has the ambition of being a unifying theory for the study of all kinds of RNNs, providing a common framework that allows for analysing the expressivity of RNN architectures in a uniform way, in order to guarantee solid progress in the field. First of all, MAT generalises the notion of finiteness to a general metric notion of $\eta$-*finiteness* (Definition 2), which captures the intuitive idea of the finite-precision setup, while retaining generality. Second, we develop a correspondence between $\eta$-finite systems and finite automata, thus allowing us to apply powerful algebraic results and notions of Automata Theory. Third, the theory introduces a notion of *robustness* (Definition 4) that guarantees numerical stability, contributes to stability learning, and notably allows one to prove results for real-world finite-precision implementations while abstracting away the difficulties introduced by finite-precision arithmetic. Fourth, we develop the notion of *geometrically-constrained systems* (Definition 8). This notion goes beyond the setting of finite-precision, allowing for modelling of languages beyond regular. It captures the empirical properties of systems approximating languages with length-generalization properties, which are observed in practice. Finally, we showcase the effectiveness of Metric Automata Theory by proving a comprehensive set of expressivity results for widely-adopted RNN architectures, with a focus on robustness and finite-precision. We argue that our results provide an improved understanding of the actual capabilities of RNNs as observed in practice.

## 2    Preliminaries

We present the most central and possibly lesser-known preliminary notions here, and we defer notation, additional background on metric spaces, background on Recurrent Neural Networks, and additional background on the topics presented below to Appendix A.

**Path-connectedness in metric spaces.**    A *path* in $X$ from $a$ to $b$ is a continuous map $\gamma : [0,1] \to X$ such that $\gamma(0) = a$ and $\gamma(1) = b$. We can define a relation $\sim_X$, where $a \sim_X b$ when there is a path in $X$ from $a$ to $b$. This relation is an equivalence, partitioning $X$ into disjoint equivalence classes, called *(path-connected) components*. For space $X$, we denote the set of its equivalence classes by $\overline{X}$. Path-connectedness is preserved by continuous functions, which is a crucial property to our theory. Notably, a continuous function $X \to Y$, with finite codomain $Y$, has to map all points within a component of $X$ to the same element of $Y$.

**Dynamical systems.**    Following [Knorozova and Ronca, 2024a,b], we adopt dynamical systems as our general formalism. A *(dynamical) system* is a tuple $S = \langle X, U, f, x_0, Y, h \rangle$, where $X$ is the *state space*, $U$ is the *input space*, $f : X \times U \to X$ is the *dynamics function*, $x_0 \in X$ is the *initial state*, $Y$ is the *output space* and $h : X \times U \to Y$ is the output function. We have that $X, U, Y$ are metric spaces, and $f, h$ are *continuous*. We call the tuple $D = \langle X, U, f \rangle$ the *dynamics* of $S$. Given $x_0 \in X$ and $N \in \mathbb{N}$, dynamics $D$ define a map from sequences $u_{[1..N]}$ of inputs to sequences $x_{[1..N]}$ of states with each state given by $x_n = f(x_{n-1}, u_n)$. Hence, we say that $D$ *defines* the function $D : X \times U^* \to X$ with $D(x_0, \varepsilon) = x_0$ on the empty sequence $\varepsilon$, and $D(x, u_{[1..n]}) = x_n$ on any input string $u_{[1..n]}$. We refer to the function defined by $D$ as *state-sequence function*. System $S$ defines a map from input sequences $u_{[1..N]}$ to output sequences $y_{[1..N]}$ where $y_n = h(x_n, u_n)$ for

all $n$. Hence, we say that $S$ *defines* the function $S : U^+ \to Y$ with $S(u_{[1..n]}) = y_n$. When $h$ is independent of $U$, we additionally define $S(\varepsilon) = h(x_0)$, extending the definition to $S : U^* \to Y$.

**Cascades.** The formalism of cascades provides a flexible way to describe dynamical systems consisting of subsystems forming an acyclic network. Their flexibility allows us, e.g., to consider not only feed-forward layers of SSMs as in [Grazzi et al., 2025, Sarrof et al., 2024], but also more complex architectures with, e.g., mixes of different types of neurons (see Figure 6 in Appendix A.6).

A *feed-forward cascade* $C$ is a form of dynamics $\langle X, U, f \rangle$ with $X = X_1 \times \cdots \times X_n$, and $f$ with a particular factorisation. We may see $C$ as consisting of dynamics $D_1, \ldots, D_n$ where $D_i = \langle X_i, U \times X_{[1..i-1]} \rangle$. State updates in a cascade proceed in a feed-forward fashion, with component $D_i$ having access to the updated states of the previous components $D_1, \ldots, D_{i-1}$. Details of cascades in relation to Automata Theory are deferred to Appendices B.5 and G.2.

**Finite Automata and Formal Languages.** A *(finite) alphabet* is a finite set $\Sigma$ of elements called *letters* or *symbols*. A *(formal) language* $L$ over $\Sigma$ is a subset of $\Sigma^*$. It is often convenient to characterise $L$ in terms of its indicator function $\mathbb{I}_L$. A *(finite) automaton* is a tuple $A = \langle Q, \Sigma, \delta, q_0, \Gamma, \theta \rangle$, where $Q$ is a *finite* set of elements called *states*, $\Sigma$ is the *finite input alphabet*, $\delta : Q \times \Sigma \to Q$ is called *transition function*, $\Gamma$ is an alphabet called *output alphabet*, and $\theta : Q \times \Sigma \to \Gamma$ is called *output function*. The tuple $A' = \langle Q, \Sigma, \delta \rangle$ is called a *semiautomaton*, and in particular $A'$ is the semiautomaton of $A$.

An automaton $A$ with output alphabet $\Gamma = \{0, 1\}$ is called a *language recogniser*, and it *recognises* the language $L$ whose indicator function is the one defined by $A$. The languages recognised by finite automata are the *regular languages*.

**Algebraic Automata Theory (AAT).** It studies finite automata through the lens of algebraic notions such as semigroups and groups, c.f. [Hartmanis and Stearns, 1966, Ginzburg, 1968, Arbib, 1969, Dömösi and Nehaniv, 2005]. The *Prime Decomposition Theorem* by Krohn and Rhodes [1965] shows how every semiautomaton can be decomposed into a cascade of *prime* semiautomata. One prime semiautomaton is the *flip-flop*, that describes the elementary system with the ability to store and manipulate one bit of information. Formally, $\text{FLIP-FLOP} \coloneqq \langle \{\texttt{high}, \texttt{low}\}, \{\texttt{set}, \texttt{reset}, \texttt{id}\}, \delta \rangle$ with transitions give by $\delta(q, \texttt{id}) = q$, $\delta(q, \texttt{set}) = \texttt{high}$, and $\delta(q, \texttt{reset}) = \texttt{low}$ for every state $q$.

Automata that admit a cascade decomposition into flip-flops are called *group-free*, and they are central since *group-free automata recognise the star-free languages*, cf. [Ginzburg, 1968]. To relate different automata, we adopt the notion of *realisation* for Mealy machines, cf. Definitions 1.14 and 1.15 of [Hartmanis and Stearns, 1966] and appendix B.4. *Realisation describes how a machine can imitate another machine after a renaming of inputs and outputs.*

**Recurrent Neural Network Architectures** An *Elman-RNN* has dynamics $D = \langle X, U, f \rangle$ where $f(x, u) = \tanh\left(A_X \cdot x + A_U \cdot u + b\right)$. *State Space Models* (SSMs) are based on linear recurrence with particular parametrisations such as Mamba [Gu and Dao, 2023]. To model *linear recurrence* in general, we introduce *Linear Recurrent Dynamics* (LRD), defined as dynamics $\langle X, U, f \rangle$, where $f(x, u) = A(u) \cdot x + B(u)$, with states $X \subseteq \mathbb{K}^{d_{\text{state}}}$, inputs $U = \mathbb{K}^{d_{\text{input}}}$ for $\mathbb{K} \in \{\mathbb{R}, \mathbb{C}\}$, We call $A(u) \in \mathbb{K}^{d_{\text{state}} \times d_{\text{state}}}$ the *state-transition gate* and $B(u) \in \mathbb{K}^{d_{\text{state}}}$ the *input gate*. The recently introduced model *xLSTM* [Beck et al., 2024] makes use of both non-linear and linear recurrences. xLSTM introduces two types of blocks: sLSTM and mLSTM. We provide the parametrization of mLSTM blocks in Appendix G.3.

## 3 Metric Automata Theory

We present Metric Automata Theory (MAT), a generalisation of Automata Theory to dynamical systems. Next we present preliminary considerations on automata and the preliminary notion of language recognition for dynamical systems. Then we present the central notions of the theory.

**Automata as dynamical systems.** We start by observing that *finite automata are a special case of dynamical systems*. Our goal is to establish a framework to analyse Recurrent Neural Networks (RNNs), with a focus on the study of their expressivity in terms of the ability to recognise formal



η-finite Dynamics $D$         Semiautomaton $\mathcal{C}(D)$

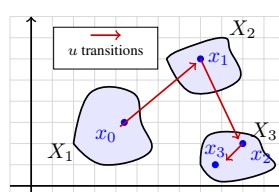 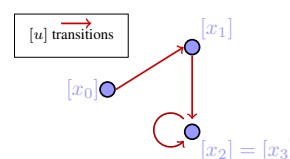



Figure 1: System dynamics and corresponding canonical semiautomaton, given in Definition 3.

languages. Every automaton $A = \langle Q, \Sigma, \delta, q_0, \Gamma, \theta \rangle$ is a dynamical system if we endow $Q, \Sigma, \Gamma$ with the *discrete metric*, giving them a discrete topology. In particular, this implies that the functions $\delta$ and $\theta$ are trivially continuous, and hence it shows that *automata are continuous systems*. The connection to dynamical systems makes it clear that a semiautomaton is a special case of dynamics, and clarifies how automata define functions $F : \Sigma^+ \to \Gamma$.

**Definition 1.** Given alphabets $\Sigma$ and $\Gamma$, and continuous functions $\mathrm{enc} : \Sigma \to U$ and $\mathrm{dec} : Y \to \Gamma$, we say that a system $S$ *implements* a function $F : \Sigma^+ \to \Gamma$, with *encoder* enc and *decoder* dec, if $F(w) = \mathrm{dec} \circ S(\mathrm{enc}(w))$, for every $w \in \Sigma^+$, where $\mathrm{enc}(w) \in U^+$ applies enc element-wise. We also say that $S$ *can-implement* $F$ if it implements $F$ for some choice of enc and dec. When $\Gamma = \{0, 1\}$, we say that $S$ *recognises* a language $L$ if it implements its indicator function $\mathbb{I}_L$, and that $S$ *can-recognise* $L$ if it can-implement $\mathbb{I}_L$.

### 3.1 The Notion of η-Finiteness for Dynamical Systems

We show that the metric setting allows for a general notion of finiteness of a given space, capturing the fact that it is *essentially finite* even if its cardinality is not—details in Appendix B.

**Definition 2.** For $X$ a set with $X \subseteq \mathbb{R}^d$ or $X \subseteq \mathbb{C}^d$, we say that $X$ is *η-finite* if it is a finite union of compact, path-connected components. Then, we say that dynamics $\langle X, U, f \rangle$ are *η-finite* if both $X$ and $U$ are η-finite. Finally, a system $S$ is *η-finite* if its dynamics are η-finite.

We refer to the components of the definition as *η-components* of the space. For example, finite alphabets are η-finite, with each element being its own η-component. As path-connectedness and compactness are preserved by continuous mappings, the notions of η-finiteness and η-component have *very favourable theoretical properties*. Any continuous mapping $f : X \to Y$, with $X$ and $Y$ η-finite, is guaranteed to map any η-component of $X$ entirely into a single η-component of $Y$.

As a result, all points within the same state η-component will be interpreted as *equivalent* states, yielding equivalent behaviours of the system; and all points within the same input or output η-component will correspond to the same inputs and outputs modulo encoding and decoding.

All automata are η-finite systems since they are discrete. Conversely, every η-finite system admits a *canonical automaton*, which fully captures its dynamics and capabilities. It gives us a way to employ the powerful characterisations and results of AAT to any η-finite system dynamics. Figure 1 visualizes the way in which the canonical (semi)automaton is a discrete interpretation of the continuous dynamics.

**Definition 3.** Any η-finite dynamical system $S = \langle X, U, f, x_0, Y, h \rangle$ admits a unique *canonical automaton*, and any η-finite dynamics $D = \langle Z, V, g \rangle$ admits a unique *canonical semiautomaton*, which are respectively given by $\mathcal{C}(S) := \langle \overline{X}, \overline{U}, \overline{f}, [x_0]_{\sim_X}, \overline{\mathrm{Im}\, h}, \overline{h} \rangle$, and $\mathcal{C}(D) := \langle \overline{Z}, \overline{V}, \overline{g} \rangle$.

**Theorem 1.** *An η-finite system $S$ can-implement the same functions as its canonical automaton, which are necessarily regular.*

Canonical automata are a *core tool we use to develop an algebraic theory of continuous systems*. Generally, it allows us to abstract away the *local* details of continuous behaviour that do not affect the *global* expressive capacity of the system. We use it to, e.g., apply the decomposition theorems of AAT to RNNs, and to create the appropriate analogue of *realisation* of a continuous system.

Strongly robust system + $\epsilon$-covering approximation

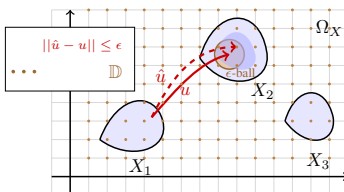 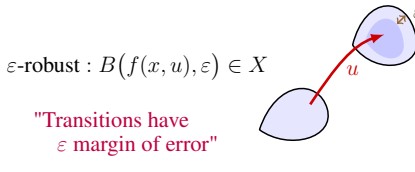

Figure 2: Given sufficient precision, the transitions of strongly $\epsilon$-robust dynamics can be realized with approximate dynamics on a finite datatype, whenever the datatype gives a $\epsilon$-covering for the state-space.

**MAT applied to finite-precision.** We explain how Metric Automata Theory provides the foundations to study dynamical systems in a principled way, with the study of RNNs as a special case that is of particular interest for us. First, we note that $\eta$-finiteness ensures that the $\eta$-components of the considered space are *bounded* and *separated* by some positive non-zero distance. Thus, in all $\eta$-finite systems, sequences of states converging to a limit will eventually lie in a single $\eta$-state—we note that this is the key property of the finite-precision arguments in [Sarrof et al., 2024]. Additionally, in every finite-precision implementation of a system (e.g., with states represented as tensors of floating-point numbers), the state space has finite cardinality, and hence it is trivially $\eta$-finite. Similar considerations apply to the input and output spaces. Altogether, Metric Automata Theory equipped with the notion of $\eta$-finiteness allows for studying finite-precision (implementations of) dynamical systems without restricting the analysis to any specific representation of the relevant spaces. The next section develops the theory in the case of system implementations based on concrete datatypes.

### 3.2 Robust Systems

The central notion that allows us to extend Metric Automata Theory to the study of finite-precision implementations is the notion of $\epsilon$-*robustness*. Intuitively, it describes stability of the dynamics under transition perturbations. It provides a way to connect $\eta$-finite systems to their floating-point implementations on real-world computer architectures, without requiring us to commit to any particular standard of floating-point operations. We let $\overline{B}_\Omega(x, r) := \{y \in \Omega : ||x - y|| \leq r\}$, which denotes the closed $\Omega$-*ball* at $x$ of radius $r$.

**Definition 4.** For $\epsilon > 0$ and $X \subseteq \Omega$, dynamics $D = \langle X, U, f \rangle$ are $\epsilon$-*robust (in $\Omega$)* if, for every $x \in X$ and every $u \in U$, it holds that $\overline{B}_\Omega(f(x, u), \epsilon) \subseteq X$—i.e., $y \in X$ for all $y \in \Omega$ s.t. $||f(x, u) - y|| \leq \epsilon$.. We say that dynamics $D$ are *strongly $\epsilon$-robust* (in $\Omega$) if they are $\epsilon$-robust (in $\Omega$) and each $\eta$-component of $X$ contains an $\Omega$-ball of radius at least $\epsilon$.

We call dynamics *robust (resp. strongly robust)*, if they are $\epsilon$-robust (resp. strongly $\epsilon$-robust) for some $\epsilon > 0$. Note that the property of robustness is with respect to the ambient space $\Omega$, which contains the state space $X$. It is possible that a dynamics is $\epsilon$-robust w.r.t. some ambient space (e.g., $\mathbb{R}$), and not $\epsilon$-robust w.r.t. another ambient space (e.g., $\mathbb{C}$). Next we discuss how our notion of robustness allows for drawing conclusions on finite-datatype implementations of a system.

**Definition 5.** A *finite datatype* is a set $\mathbb{D} \subseteq \Omega = \mathbb{R}^d$ having finite cardinality. A *finite-datatype implementation* of a system $S$ is then a system whose input, state, and output spaces are finite datatypes, and whose dynamics and output functions are implemented using floating-point operations.

**Theorem 2** (Informal version). *Every $\eta$-finite system with strongly $\epsilon$-robust dynamics, for $\epsilon > 0$, can be implemented with floating-point operations given sufficient precision.*

By *sufficient precision* we mean that the state space is sufficiently covered by the finite datatype, and that the floating-point approximation of the dynamics has error at most $\epsilon$. In Appendix C we show two examples of floating-point parametrisations for which the former condition can always be achieved using sufficiently-many bits of precision.

**Considerations on training.** Training any machine learning model that can be seen as a dynamical system amounts to optimising a parametric dynamics function $f_\theta$, with learnable parameter $\theta \in \Theta$, along with optimising the output function. In Section C.2 of Appendix C, we prove that, under some mild regularity assumptions, when dynamics $D_\theta = \langle X, U, f_\theta \rangle$ are robust, there is a $\delta > 0$ such that for all $\theta' \in \Theta$ with $\|\theta - \theta'\| \leq \delta$, the function $f_{\theta'} : X \times U \to X$ is a well-defined dynamics function, and the corresponding dynamics $D_{\theta'} = \langle X, U, f_{\theta'} \rangle$ have the same $\eta$-state dynamics as $D_\theta$–i.e., they both have the same canonical semiautomaton. Thus, replacing $f_\theta$ with $f_{\theta'}$ in a system will not change the system behaviour. Given the previous consideration, an argument should be possible by which models enjoying this form of robust parametrisation are more likely to be produced by training algorithms, compared to models that do not admit a robust parametrisation. However, a systematic development of this argument is beyond the scope of our work.

# 4 Expressivity Results for Vanilla-RNNs, xLSTM, and SSMs

Metric Automata Theory allows us to establish a rich ensemble of expressivity results in the finite-precision setting and beyond finite-precision. The elegance and generalisability of our setup enables us to compare capabilities of wildly different models. For example, Theorem 4 applies to SSMs with both *real and complex* state spaces.

## 4.1 Expressivity Results for Robust Language Recognition

We prove that linear recurrences do not admit robust dynamics, whenever they have an identity transformation on their $\eta$-states.

**Theorem 3** (Non-robustness of LRDs). *Suppose an $\eta$-finite LRD $D$ is such that its canonical semiautomaton $D_A$ has at least two states, and an input inducing an identity transformation. Then $D$ cannot be $\epsilon$-robust for any $\epsilon > 0$.*

In fact, we show that upon iterating any single input the whole state-space of an $\eta$-finite $\epsilon$-robust LRD collapses to a single $\eta$-component. We call such dynamics *contracting*. Furthermore, we show that a cascade of contracting dynamics is contracting, and that contracting dynamics cannot implement a FLIP-FLOP. We defer the technical details to Section D.3.

**Theorem 4** (LRDs cannot do FLIP-FLOP robustly). FLIP-FLOP *cannot be implemented by a cascade of $\eta$-finite $\epsilon$-robust LRDs for any $\epsilon > 0$.*

**xLSTM.** We provide constructions for strongly robust realisations of the FLIP-FLOP dynamics for Elman-RNNs and for an sLSTM block—see Appendix G.3 for the details. The Elman-RNN construction is similar to one provided in [Knorozova and Ronca, 2024a], with the `high` and `low` $\eta$-states located around the attracting fixed-points of `tanh`. For xLSTM, fixing a particular parametrisation of a sLSTM block allows us to use a very similar construction, with a `sigmoid` non-linearity. By Fact 1, this proves that all star-free languages can be implemented cascade of strongly-robust xLSTM blocks. Such a cascade is strongly-robust.

**Theorem 5** (xLSTM does start-free robustly). *All star-free languages can be recognised by xLSTM cascades, as well as by floating-point implementations of xLSTM cascades given sufficient precision.*

## 4.2 SSM Expressivity in Finite-Precision

We prove that $\eta$-finite SSMs with state-transition gates having non-negative eigenvalues are restricted to group-free dynamics, and hence can only implement star-free languages, in line with the theoretical results by Sarrof et al. [2024] and Grazzi et al. [2025], in their respective finite-precision setups.

**SSMs with non-negative eigenvalue gates are star-free.** To transfer the group-free notion into the continuous $\eta$-finite dynamics setting we introduce a notion of *aperiodic* dynamics. We say that an infinite sequence in a $\eta$-finite space $X$ is *$\eta$-convergent in $X$* if all terms of the sequence are eventually in the same $\eta$-component of $X$.

**Definition 6.** We say that $\eta$-finite dynamics $D = \langle X, U, f \rangle$ are *aperiodic* if, for every $x_0 \in X$ and every input sequence $(u_n)_{n \geq 1}$ that is $\eta$-convergent in $U$, we have that the corresponding state sequence $(x_n)_{n \geq 1}$ is $\eta$-convergent in $X$.

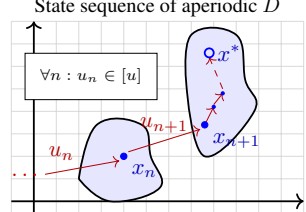
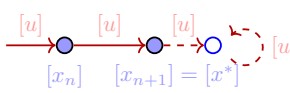

State sequence of aperiodic $D$        $\eta$-state sequence of $\mathcal{C}(D)$

Figure 3: Definition of aperiodicity means that the state sequence of aperiodic dynamics under iterated input from the same $\eta$-component always $\eta$-converges. In particular, if the state sequence always converges to some limit under iterated inputs, then the dynamics are aperiodic. This is the case e.g., for LRDs with diagonal gates with entries in $(-1, 1)$.

In Section D.2 of Appendix D, we show that cascades of aperiodic dynamics are aperiodic. Moreover, we show, that $\eta$-finite dynamics are aperiodic iff their canonical semiautomaton is group-free. Thus, aperiodic $\eta$-finite systems can implement only star-free regular languages.

**Theorem 6.** *Let $D$ be $\eta$-finite Linear Recurrent Dynamics, with its state-transition gates having all non-negative eigenvalues. Then $D$ is aperiodic.*

The proof structure is similar to the proof of Theorem 1 in [Grazzi et al., 2025], with *significant simplifications* afforded by our theory. We show that, iterating a fixed input, the state converges, by considering the Jordan Normal Form of the state-transition gate. We also show that finite context length convolutions are aperiodic. Thus, SSMs like Mamba, which are cascades of convolutions and LRDs with non-negative eigenvalue gates, can only recognise star-free languages as they are $\eta$-finite.

**Mamba cannot implement FlipFlop in finite-precision.** The FlipFlop dynamics construction presented in [Sarrof et al., 2024] makes use of the identity state-transition gate. Parametrisation of Mamba prevents it from making use of such gate. In fact, we prove that in the $\eta$-finiteness framework, Mamba blocks are *contracting* dynamics, and thus cannot implement a FLIP-FLOP.

**Theorem 7.** *SSMs with Mamba parametrisation cannot recognise* FLIP-FLOP.

## 4.3 Geometrically Constrained Systems

The case of Mamba successfully length-generalising on star-free tasks, despite being unable to model the dynamics for unbounded length inputs, motivates us to expand our theory beyond $\eta$-finite systems. The intuition behind the following setup is to allow only for output functions that are sufficiently regular to expect them to be *learnable* from short input sequences, with length-generalisation. Ultimately, this section provides an example of how Metric Automata Theory can be used to develop theories alternative to $\eta$-finiteness, motivated by phenomena observed empirically and defined by geometric properties of the dynamical systems.

**Definition 7.** Let $\Omega = \mathbb{R}^d$ or $\Omega = \mathbb{C}^d$. We call $C \subseteq \Omega$ a *convex-covering* if it is a finite union of open, convex sets in $\Omega$. Then, we say that $X \subseteq \Omega$ is *convex-covered* by $C$ if $X \subseteq C$.

A convex-covering $C$ consists of finitely-many path-connected components, which are open. The path-connected components of $C$ can be arbitrarily classified by an output function with piecewise-linear decision boundaries, with finitely many vertices. Such output functions include all feed-forward networks with ReLU activations, see Proposition 6.1 in [Zhang et al., 2018].

**Definition 8.** Let $\Omega = \mathbb{R}^d$ or $\Omega = \mathbb{C}^d$, and let $C \subseteq \Omega$. We say that dynamics $D = \langle X, U, f \rangle$ are *convex-covered* by $C$ if $X$ is convex-covered by $C$. We call a system $S_C = \langle X, U, f, C, x_0, Y, h \rangle$ *geometrically-constrained* by $C$ if its dynamics $D_C = \langle X, U, f \rangle$ are convex-covered by $C$.

*Geometrically-constrained systems* (GCS) are a generalisation of $\eta$-finite systems, as any $\eta$-finite can be extended to a geometrically-constrained system with equal capabilities. GCSs can in fact express dynamics beyond finite-state, e.g., Construction 1. The GCS framework is presented in Appendix E.

**Construction 1.** Consider Linear Recurrent Dynamics with state-space $X = \mathbb{Z}$, input space $U = \{a, b\}$ and dynamics function $f(n, a) = n + 1; \quad f(n, b) = n - 1$. The space $C = (-\infty, -0.5) \cup$

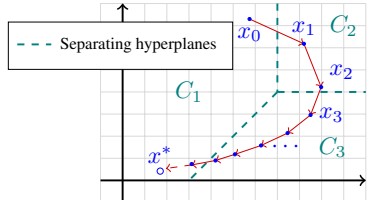 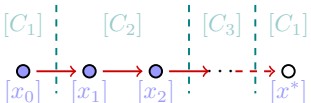

$$x_{n+1} = A_u \cdot x_n + B_u : A_u \text{ with e-values} \geq 0$$
$$X \subseteq C_1 \cup C_2 \cup C_3$$

Figure 4: SSMs with state transition gates without negative eigenvalues are not capable of alternating around any separating hyperplane under iterated input. Thus, eventually the state must be mapped to a constant output. This makes such SSMs unable to e.g., implement Parity as GCSs.

$(-0.5, 0.5) \cup (0.5, \infty)$ is a convex-covering for this dynamics. We may define the output function $h : C \rightarrow \{0, 1\}$ to map points in $(-\infty, -0.5) \cup (0.5, \infty)$ to 0 and points in $(-0.5, 0.5)$ to 1. Picking initial state $x_0 = 0$, we have that this GCS outputs 0 precisely when the input has the same number of $a$s and $b$s.

**Connection to automata.** In the case of dynamics $\langle X, U, f \rangle$ constrained by $X$, we recover the correspondence to Automata Theory via canonical semiautomata, and hence we can use the theorems of AAT—details in Section E.2. The next construction shows that, as a GCS, Linear Recurrent Dynamics with Mamba parametrisation realise FLIP-FLOP, unlike in the robust $\eta$-finite setting.

**Construction 2.** FLIP-FLOP dynamics can be realised by constrained Linear Recurrent Dynamics with diagonal state-transition gate, with entries in $[\frac{1}{4}, \frac{3}{4}]$. Take $D = \langle X, U, f \rangle$ with $X = X_l \cup X_h$, where $X_l = (-1, 0)$, $X_h = (0, 1)$, $U = \{i, l, h\}$ and $f(x, \sigma) = A_\sigma \cdot x + B_\sigma$ where $\langle A_i, B_i \rangle = \langle 3/4, 0 \rangle; \langle A_l, B_l \rangle = \langle 1/4, -1/2 \rangle; \langle A_h, B_h \rangle = \langle 1/4, 1/2 \rangle$. With output function $X_l \mapsto \texttt{low}$ and $X_h \mapsto \texttt{high}$ (indeed continuous), $D$ realizes FLIP-FLOP, and $X$ is a convex-covering of $D$.

In particular, given the realisation of FLIP-FLOP in Construction 2, we obtain the following:

**Theorem 8.** *SSMs with Mamba parametrisation can recognise all star-free languages as GCSs.*

**Modular counting.** We extend the notions of cascades to this setup, with restriction on how components depend on inputs from other components, corresponding to the idea of joining the cascade with connecting functions. Similarly, we extend the notion of aperiodic dynamics, with the modification that we require the state-sequence to be $\eta$-convergent in $C$, instead of $X$ in the usual definition. Appendix E.1 explains how aperiodicity is preserved by constrained cascades in this setup. In the GCS framework, we can no longer equate aperiodic dynamics with group-free semiautomata—the GC-system in Construction 1 is aperiodic, but implements a language which is not even regular. We can still obtain more specialised expressivity results. Aperiodicity prevents a GC-system from modelling any function for which iterating the same input can alternate between distinct outputs indefinitely. We call a function $F : \Sigma^+ \rightarrow \Gamma$ is *alternating* if, for some $\sigma \in \Sigma$, the sequence $\left(F(\sigma^n)\right)_{n \geq 1}$ changes value infinitely many times. All alternating functions are group-like. As an example, functions that perform modulo-$M$ counting are alternating.

**Theorem 9.** *Let $D$ be an $\eta$-finite Linear Recurrent Dynamics, with its state-transition gates having all non-negative eigenvalues. Let $C$ be a covex-regular covering of $D$. Then $D$ is aperiodic w.r.t. $C$.*

The proof is similar to that of Theorem 6. By considering the Jordan Normal Form of the state-transition gate, we show that the state sequence cannot alternate around the separating hyperplanes of the convex components making up $C$. Overall, we obtain that *SSMs such as Mamba are not able to implement alternating functions as geometrically-constrained systems.*

## 5 Empirical Validation of Our Results

**Mamba performance on star-free tasks.** The experiments presented by [Sarrof et al., 2024] demonstrate that Mamba can effectively learn star-free languages with length-generalisation abilities. On the benchmark from [Bhattamishra et al., 2023], Mamba performed perfectly on all 11 star-free tasks, also on out-of-distribution input lengths. This is consistent with its expressivity described by

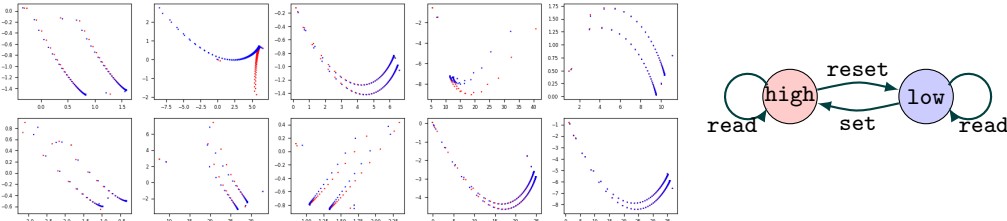

Figure 5: FLIP-FLOP task [Liu et al., 2023]. PCA of a trained 1-layer Mamba states for each channel: red and blue are state sequences under `i0` inputs, starting from `w1` and `w0` respectively. After $\approx$1000 inputs, both state sequences give the same predictions on the read instruction `r`, incorrectly.

Theorem 8. We performed additional experiments on the dataset from [Liu et al., 2023]. It introduces the task of realizing the FLIP-FLOP by predictively modelling a sequence of instructions. We found that in the case of training 1-layer Mamba, despite achieving accuracy 1 on all validation datasets, iterating the ignore instruction indeed leads to incorrect outputs, as *predicted by our results for $\eta$-finite systems, namely Theorem 7*. See Figures 5,10 and Appendix F for details.

**Non-star-free tasks.** Our negative results for SSMs in the $\eta$-finite setup predict that SSMs with non-negative eigenvalues (non-negative SSMs for short) for the state-transition gate cannot implement non-star-free tasks. The experiments performed by Sarrof et al. [2024] on Mamba with the datasets from [Bhattamishra et al., 2023] show that Mamba struggles to model non-star-free tasks. The empirical evidence presented in [Grazzi et al., 2025] similarly validates our results, with results for non-star-free languages from the Chomsky Hierarchy benchmark by Deletang et al. [2023]. Remarkably, both the Chomsky Hierarchy and Bhattamishra's benchmarks have the worst results for non-negative SSMs on languages involving modulo counting. Our negative results in the geometrically-constrained framework suggest that *this is caused by the inherent geometry of the state-space for these models.*

**Significance of robustness.** Beck et al. [2024] and Grazzi et al. [2025] evaluate their proposed architectures on the Chomsky Hierarchy benchmark [Deletang et al., 2023]. Even though, as shown in [Grazzi et al., 2025], DeltaNet with negative eigenvalues is capable of modelling the Modular Arithmetic w/o Brackets task, it falls short of perfect accuracy on all sequence lengths. On the other hand, sLSTM achieves perfect accuracy on this task, as reported by Beck et al. [2024] (although Grazzi et al. [2025] failed to reproduce these results). Theorem 3 gives a possible explanation for why linear recurrences may perform worse in practice than non-linear recurrences. This effect can also be observed for star-free tasks—we defer further discussion to Appendix F.

**Beyond regular tasks.** Contex-free and context-sensitive tasks remain challenging for the recent recurrent archtectures, as evident by the performance of xLSTM, DeltaNet and Mamba on the Chomsky Hierarchy benchmark, reported in [Beck et al., 2024] and [Grazzi et al., 2025]. This indicates that $\eta$-finite systems are largely a good model for the finite-precision setting. Sarrof et al. [2024] report that Mamba achieves good results for counter languages, but with limited length-generalisation. We conjecture that counter-like dynamics, which are permitted in the GCS framework, are not possible for Mamba, as its dynamics are space-contracting.

## 6    Limitations

The limitations of Metric Automata Theory (MAT) in its current, initial, state of develoment revolve around three aspects, that we discuss below.

**Limitations inherited from AAT.** MAT allows one to employ Algebraic Automata Theory (AAT) for the purpose of analysing RNNs. However, AAT is underdeveloped in many ways, with limitations on its current ability to describe certain fine-grained expressivity aspects, which clearly transfer to MAT. A key limitation is that AAT does not focus on the complexity of the functions that connect the stateful components in a cascade, and specifically it provides no results on how the complexity of such functions influences the expressive capacity of a model. Now that our MAT makes AAT relevant for the study of RNNs, there is a *new motivation in futher developing AAT*.

**Dynamics-dependent state space.** Requiring continuity throughout means that the main work of assigning meaning to states is done in selecting the state space $X$. Further, the dynamics $f$ need to have codomain $X$, which can make verifying constructions complicated. In the context of learning parameters for $f$, as the parameters vary, the state space must change accordingly, making it less straighforward to derive results regarding learning. Nonetheless, MAT already allows for indirect analysis of learning stability, via the notion of robustness, as discussed in Section 3.2.

**Focus on unbounded-length expressivity.** Most of our work studies the ability of RNNs to recognise languages where the length of strings is unbounded. Additional results could be proved regarding the ability of RNNs to recognise languages where strings have bounded length. Some of our notions and results—e.g., robustness or GCSs—can still be applied in this context, but otherwise MAT may require to be extended significantly.

## 7 Related Work

Our dynamical systems approach follows the framework by Knorozova and Ronca [2024a,b]. This set of results focuses on RNC$_+$, which are cascades of 1-dimensional Elman-RNN neurons, with dynamics function $f(x, u) = \tanh(w \cdot x + u)$ having $w \geq 0$. Expressivity of RNC$_+$ in terms of regular languages is shown to be exactly the star-free languages. Their setup is not directly relatable to ours under $\eta$-finiteness, but it implicitly assumes that the state-space is *compact*, and uses similar convergence arguments as Sarrof et al. [2024] and Grazzi et al. [2025], combined with AAT. The authors hope to further develop the theoretical foundations of expressivity theory, and to incorporate further theories, such as the work in [Knorozova and Ronca, 2024a,b], into Metric Automata Theory.

Related expressivity results for SSMs are given in [Sarrof et al., 2024, Grazzi et al., 2025, Merrill et al., 2024], and for ReLU-activated Elman-RNNs and LSTMs in [Weiss et al., 2018]. We defer the discussion of such results to Appendix H.

## 8 Future Work

The framework we set up fills in the gaps in the existing literature in terms of general theoretical methodology, as well as understanding of empirical phenomena. At the same time, it opens up new avenues for future research in connection to automata theory, model design, and learning. We especially see robustness as being of practical interest and as a subject of future research. Next we discuss a few concrete points that are on our research agenda. First, we plan to devise additional experiments to fully understand the impact of our results on learning models—e.g., measuring robustness trade-offs between xLSTM and SSM length-generalisation on star-free tasks. Second, we plan to study how tokenization affects the models ability to perform state-tracking and realise automata transitions. For example, Grazzi et al. [2025] (paragraph under Theorem 3, page 7) note that allowing more input symbols per transition (e.g., "$3 + 2 + 4 = 4$") allows simpler gates to implement automata. Third, we would like to explore the potential of robustness in driving design decisions behind model architectures and training algorithms. For example, the inherent non-robustness of linear RNNs suggests that the solutions that may be learned for the model's parametrisation are very sensitive, especially when it comes to length-generalisation abilities. Fourth, we plan to employ the GCS theory for investigating the ability of RNNs to recognise languages beyond regular. A notable family of languages to consider is the one of counter languages, already mentioned in Construction 1 and in the analysis of the performance of Mamba in Section 5. Finally, we would like to use the GCS theory to further clarify length-generalisation phenomena.

## 9 Conclusions

We have presented Metric Automata Theory, an elegant and principled theory that generalises classic Automata Theory to dynamical systems, and to RNNs in particular. The fundamental notions and key properties of the theory we have described, as well as the deep understanding of several widely-adopted RNNs that we were able to provide using the theory, justify the ambition of the theory to be a unifying theory for the study of RNNs, and also dynamical systems in general. The introduced notions, e.g. of robustness, leave many exciting avenues for deeper study.

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

# Appendices

The appendices provide proofs of the theorems stated in the main body, as well as more detailed exposition of preliminary notions, and illustrative figures. It is structured as a suppleental body of work which can be read from top to bottom, and which gives a detailed presentation of Metric Automata Theory and its main results. While the main body gives a big picture overview of the key notions and results, the appendices aim to serve as a foundational text, showcasing how Metric Automata Theory can be used to develop new theories and draw novel insights about RNN architectures—in addition to providing full proofs of all results stated in the main body.

**Appendix A** provides standard *preliminary notions*, required for later sections and in particular for proving our results.

**Appendix B** presents the *foundations of Metric Automata Theory (MAT)*, which build on several different fields—metric spaces, dynamical systems, algebraic and classic automata theory. Also establishing novel and fundamental connections across such fields. We prove Theorem 1 in this appendix.

**Appendix C** introduces the novel notion of *ε-robust* dynamics, which allows us to argue about real-world floating point implementations of models. It also describes numerical and parametrisation stability properties of systems, thus going beyond the phenomena which can be described by discrete systems. We provide proofs of Theorem 2 and Theorem 5.

**Appendix D** employs Metric Automata Theory and its connection to Algebraic Automata Theory to show a collection of expressivity results in the $\eta$-finite setting, including Theorems 3, 4, 6 and 7.

**Appendix E** explores the setting of *Geometrically-Constrained Systems (GCS)*, in connection to the empirical length-generalisation capabilities of Mamba, which go beyond its finite-precision expressivity. We give proofs of Theorem 8 and Theorem 9.

**Appendix F** gives further details on the visualisation experiments we conducted to showcase the state-space collapse suffered by Mamba SSMs.

**Appendix G** contains technical proofs and constructions deferred from other sections, which are not necessary to fully comprehend the overall argument they are used in.

**Appendix H** continues the discussion of related work from Section 7, notably contrasting the frameworks of Sarrof et al. [2024] and Grazzi et al. [2025].

# A   Additional Preliminaries

In this Appendix, we introduce the preliminary notions for the remainder of this work.

Section A.1 covers basic mathematical notions and notation used throughout.

Section A.2 introduces the necessary background in Metric Spaces and Topology, notably properties of *compactness* and *path-connectedness*.

Section A.3 defines the language of Dynamical Systems, which we use to describe RNNs and to build our theory.

Section A.4 shows the key Algebraic Automata Theory results and notions which we use in our work.

Finally, Section A.5 and Section A.6 cover MLPs and introduce relevant RNN architectures.

## A.1   Basic Concepts and Notation

We introduce basic mathematical concepts and notation required in later sections.

### A.1.1   Numeric Domains

We write $\mathbb{B} = \{0, 1\}$ for the Boolean domain, we write $\mathbb{N} = \{0, 1, \ldots\}$ for the natural numbers, we write $\mathbb{N}_{>0} = \{1, 2, \ldots\}$ for the natural numbers excluding zero, we write $\mathbb{R}$ for the real numbers, we write $\mathbb{R}_+$ for the positive real numbers including zero, we write $\mathbb{R}_{>0}$ for the positive real numbers

excluding zero, i.e., $\mathbb{R}_{>0} = \mathbb{R}_+ \setminus \{0\}$, and we write $\mathbb{C} = \{\langle a, b \rangle \mid a, b \in \mathbb{R}\}$ for the complex numbers—where every pair $\langle a, b \rangle$ is to be seen as the complex number $a + ib$.

For $i, j \in \mathbb{N}$ with $m \leq n$, we define the notation $[i..j] := \{i, i{+}1, \ldots, j\}$.

In the rest of the section, let $Z$ be a set.

### A.1.2 Powersets

We write $\mathcal{P}(Z)$ for the *powerset* of $Z$, and we define $\mathcal{P}_+(Z) := \mathcal{P}(Z) \setminus \{\emptyset\}$.

### A.1.3 Tuples and Matrices

For $n \in \mathbb{N}$, the set of $Z$-valued *n-vectors*, or *n-tuples* over $Z$, is $Z^n := \{\langle z_1, \ldots, z_n \rangle \mid z_i \in Z\}$. We typically write an element of $Z^n$ as $\mathbf{z} = \langle z_1, \ldots, z_n \rangle$. For $m, n \in \mathbb{N}$, the set of $Z$-valued $(m \times n)$-*vectors*, or $m \times n$ *matrices* over $Z$, is $Z^{m \times n} := \{\langle \mathbf{z}_1, \ldots, \mathbf{z}_m \rangle \mid \mathbf{z}_i \in Z^n\}$. We typically write an element of $Z^{m \times n}$ as $\mathbf{Z} = \langle \mathbf{z}_1, \ldots, \mathbf{z}_n \rangle$.

We use the compact notation $Z_{[i..j]}$ to specify the set $Z_i \times \cdots \times Z_j$ resulting from the Cartesian product of the sets $Z_i, \ldots, Z_j$, meaning that they are contextually introduced by the notation.

### A.1.4 Sequences

A *sequence* over $Z$ with indices $I \subseteq \mathbb{N}$ is a function $s : I \to Z' \subseteq Z$, which we commonly present as $(z_i)_{i \in I}$ where $z_i = s(i)$ for every $i \in I$. A sequence is *finite* if so is its index set, and it is *infinite* otherwise. When $s$ is an infinite sequence with index set of the form $I = \{m, m{+}1, \ldots\}$, we adopt a simplified notation and write the sequence as $(z_i)_{i \geq m}$, instead of $(z_i)_{i \in I}$. When $s$ is a finite sequence, the cardinality of its index set is called the *length* of $s$. The *empty sequence*, denoted by $\varepsilon$, is the sequence having length zero, i.e., the sequence with indices $I = \emptyset$. Any finite sequence $s$ with indices $I = [i..j]$ can be presented as the list $z_i, \ldots, z_j$ by letting $z_k = s(k)$ for every $k \in [i..j]$; in this case, the sequence can also be written in compact form as $z_{[i..j]}$. We write $Z^\omega$ for *the set of all infinite sequences* on $Z$, we write $Z^*$ for the set of all finite sequences on $Z$, we write $Z^+$ for the set of all non-empty finite sequences on $Z$, and we write $Z^\ell$ for the set of all sequence of a given length $\ell \in \mathbb{N}$—noting that this definition of $Z^\ell$ clearly corresponds to the definition given above of $Z^\ell$ as the set of all $\ell$-tuples over $Z$.

We often say that a property holds *eventually* for a sequence $(z_i)_{i \geq m}$ if there exists $m' \geq m$ such that it holds for the sequence $(z_i)_{i \geq m'}$. That is, the property holds for some *tail* of the sequence.

### A.1.5 Strings

A *string* over a finite set $\Sigma$ is a concatenation (juxtaposition) of elements of $\Sigma$. Namely, a string is an expression $\sigma_1 \sigma_2 \cdots \sigma_n$ with $\sigma_i \in \Sigma$, for every $i \in [1..n]$. In this context, we call $\Sigma$ an *alphabet*, and we call each element $\sigma_i$ a *letter* or *symbol* of the string $s$. We can equivalently see a string $\sigma_1 \sigma_2 \cdots \sigma_n$ as the finite sequence $\sigma_{[1..n]}$, following the definition of finite sequence given above, and hence apply all notions already introduced for finite sequences. In particular, we have that the length of a string $\sigma_1 \sigma_2 \cdots \sigma_n$ is $n$, that $\varepsilon$ is the empty string, that $\Sigma^\ell$ is the set of all strings of given length $\ell \in \mathbb{N}$ over alphabet $\Sigma$, that $\Sigma^*$ is the set of all strings over alphabet $\Sigma$, and that $\Sigma^+$ is the set of all non-empty strings over alphabet $\Sigma$.

### A.1.6 Functions and Transformations

The image of a function $f : X \to Y$ is $\operatorname{Im} f := \{f(x) \mid x \in X\} \subseteq Y$. We say that $f$ is an *identity* if $f(x) = x$ for every $x \in X$, and we say that $f$ is a *permutation* if it is a bijection. A *transformation of* $X$ is a function $f : X \to X$ where the codomain coincides with the domain. Note that every identity transformation is also a permutation, and hence it is sometimes important to distinguish permutations that are not identities by referring to them as *non-identity permutations*.

### A.1.7 Equivalence

For $\sim$ an equivalence relation on $Z$, the *equivalence class* of $z$ w.r.t. $\sim$ is the set $[z]_\sim := \{z' \in Z \mid z' \sim z\}$. We denote by $Z/\sim$ the set of equivalence classes of $Z$ w.r.t. $\sim$.

## A.2 Metric Spaces and Topology

We follow [Willard, 2012] as a general reference for this section, revisiting the notation. Let $X$ be a set fixed for the rest of this section.

### A.2.1 Metrics

A *metric*, or *distance function*, is a function $d : X \times X \to \mathbb{R}_{>0}$ that satisfies all the following properties for every $x, y, z \in X$:

a) $d(x, y) = 0 \iff x = y$

b) $d(x, y) \geq 0$                            (positivity)

c) $d(x, y) = d(y, x)$                 (symmetry)

d) $d(x, y) + d(y, z) \geq d(x, z)$      (triangle inequality)

Notable metrics, relevant to us, are the following ones.

- The *Euclidean distance*, or $\mathrm{L}^2$-*norm distance*, is defined as

$$\mathrm{L}^2_X(\mathbf{x}, \mathbf{y}) := \|\mathbf{x} - \mathbf{y}\| := \sqrt{(x_1 - y_1)^2 + \cdots + (x_n - y_n)^2}.$$

- The *discrete metric* is defined as

$$\mathcal{D}_X(\mathbf{x}, \mathbf{y}) := \begin{cases} 1 & \text{if } \mathbf{x} \neq \mathbf{y}, \\ 0 & \text{if } \mathbf{x} = \mathbf{y}. \end{cases}$$

We will omit $X$ from a metric when it is clear from the context. For instance, we will write $\mathrm{L}^2$ and $\mathcal{D}$ for $\mathrm{L}^2_X$ and $\mathcal{D}_X$, respectively.

### A.2.2 Metric spaces

A *metric space* is a tuple $\mathbf{S} = \langle X, d \rangle$ where $d : X \times X \to \mathbb{R}$ is a metric. Given metric spaces $\mathbf{X} = \langle X, d_X \rangle$ and $\mathbf{Y} = \langle Y, d_Y \rangle$, an *isometry* between $\mathbf{X}$ and $\mathbf{Y}$ (or *distance-preserving function*) is a bijective function $f : X \to Y$ such that, for every $_1, x_2 \in X$, we have $d_X(x_1, x_2) = d_Y(f(x_1), f(x_2))$. When an isometry exists, the spaces $\mathbf{X}$ and $\mathbf{Y}$ are said to be *isometric*. Intuitively, two isometric spaces are essentially the same metric space. Notable metric spaces, relevant to us, are the following ones, for $n \in \mathbb{N}_{>0}$.

- The *Euclidean $n$-space* $\langle \mathbb{R}^n, \mathrm{L}^2 \rangle$.

- The *complex $n$-space* $\langle \mathbb{C}^n, \mathrm{L}^2 \rangle$, seen as isometric to $\langle \mathbb{R}^{2n}, \mathrm{L}^2 \rangle$, by the following isometry:

$$f(a_1 + ib_1, \ldots, a_n + ib_n) = \langle \langle a_1, b_1 \rangle, \ldots, \langle a_n, b_n \rangle \rangle.$$

In particular, by the isometry above, all our results for Euclidean $n$-spaces transfer to complex $n$-spaces seamlessly.

We omit the metric when referring to metric spaces, since in the following sections we only consider Euclidean $n$-spaces $\langle \mathbb{R}^n, \mathrm{L}^2 \rangle$ and complex $n$-spaces $\langle \mathbb{C}^n, \mathrm{L}^2 \rangle$, that are always equipped with the $\mathrm{L}^2$ as described above. Thus we simply refer to them as $\mathbb{R}^n$ and $\mathbb{C}^n$, respectively.

A *subspace* $\langle Y, d_Y \rangle$ of $\langle X, d_X \rangle$ is a metric space with $Y \subseteq X$ and $d_Y$ given by restriction of $d_X$ to $Y \times Y$.

We define the *open ball* $B_X(x, r)$ and *closed ball* $\overline{B}_X(x, r)$ at $x \in X$ of radius $r \geq 0$ in $\langle X, d \rangle$ as the set of points in $X$ with distance $\delta < r$ and $\delta \leq r$ from $x$, respectively:

$$B_X(x, r) := \{ y \in X \mid d(x, y) < r \}, \qquad \overline{B}_X(x, r) := \{ y \in X \mid d(x, y) \leq r \}.$$

A subspace $(Y, d_Y)$ of $(X, d_X)$ is a metric space with $Y \subseteq X$ and $d_Y$ given by restriction of $d_X$ to $Y \times Y$. We say that a subspace $S \subseteq X$ is *bounded*, if there is some $x \in X$ and $\infty > M \geq 0$ s.t. $S \subseteq B_X(x, M)$. We call a subspace $S \subseteq X$ is *open in* $X$ if for all $s \in S$ there is some $\epsilon_s > 0$ s.t. $B_X(s, \epsilon_s) \subseteq S$. $S$ is *closed in* $X$ if $X \setminus S$ is open in $X$.

*Example* 1. The open intervals $(a, b)$ and $(a, \infty)$ are open in $\mathbb{R}$ (with the usual metric). The closed interval $[a, b]$ is closed in $\mathbb{R}$. The subspace $\{0, 2^{-n} : n \in \mathbb{N}\}$ is closed in $\mathbb{R}$, while $\{2^{-n} : n \in \mathbb{N}\}$ is neither closed nor open in $\mathbb{R}$. ∎

### A.2.3 Topology

The notion of open subspaces in terms of open balls defines a *topology* on any metric space, which determines what functions are *continuous*. Formally, a topological space is a tuple $(S, \mathcal{T})$, with $S$ being the underlying set, and $\mathcal{T} \subseteq \mathcal{P}(S)$ being the collection of open sets, such that $S$ and $\emptyset$, the union of *any* collection of open sets is open, and the intersection of any *finite* collection of open sets is open. The open sets definition in terms of open balls for a metric space satisfies these properties. Many aspects of Metric Automata Theory could be easily restated in the language of Topology Theory, but we choose a more concrete setting, to make it more accessible.

Intuitively, the closed subspaces of $X$ are precisely the ones which contain all their limit points, i.e. if $(x_n)_{n \geq 1} \subseteq S$ converges to some limit $l \in X$, then $l \in S$.

**Fact A.2.1.** For a metric space $X$, a subset $S \subseteq X$ is closed iff for all sequences $(x_n)_{n \geq 1} \subseteq S$ converging to $l \in X$ we have that $l \in S$. (see §10, Cor. 10.5 of Willard [2012], as every metric space is first-countable)

Note that the notion of openness/closeness is not inherent to the subspace $S$: it also depends on the superspace $X$, since the definition involves balls in $X$. In fact, any subspace $S \subseteq X$ is by definition *both open and closed* as a subspace of itself, regardless of whether is open or closed in $X$. Any time we use openness or open balls, we need to excercise caution and be clear which space the openness is referring to.

*Example* 2. Consider $M = \mathbb{R}^2$ and $X = \mathbb{R} \times \{0\} = \{(x, 0) \in \mathbb{R}^2 : x \in \mathbb{R}\}$. $(-1, 1) \times \{0\} \subseteq X$ is an open ball at $(0, 0)$ of radius 2 in $X$, and thus an open set. However, it is not even an open set in $M$! For any $\epsilon > 0$ we have $||(0, 0) - (0, \epsilon)|| = \epsilon$, but $(0, \epsilon) \notin S$, and so no open $X$-ball centred at $(0, 0)$ is wholly contained in $S$. ∎

In fact, any subspace $S \subseteq X$ is by definition *both open and closed* as a subspace of itself, regardless of whether is open or closed in $X$.

A *continuous function* $f : (M, d) \to (M', d')$ is the a set function $f : M \to M'$ such that for all sequences $(x_n)_{n \geq 1} \subseteq M$ converging to some $x \in M$, the mapped sequence $(f(x_n)) \subseteq M'$ converges to $f(x) \in M'$. The $\epsilon - \delta$ definition of continuity, as well as the topological definition of continuity ($Y \subseteq M'$ open $\implies f^{-1}(Y) \subseteq M$ open) are equivalent in the metric space setting.

*Example* 3. Let $S$ be a subspace of $X$. Then the inclusion map $\iota : S \to X$, given by set-theoretical inclusion $S \subseteq X$, is continuous. ∎

The topological definition of continuity makes clear the following:

**Fact A.2.2.** All functions $f : (M, d) \to (M', d')$ are continuous for a discrete metric space $(M, d)$.

Next, we introduce two elementary notions in Topology and Metric Space Theory: *compactness* and *path-connectedness*.

### A.2.4 Compactness

**Definition 9.** A space $X$ is called *compact* if all coverings of $X$ by open subsets of $X$ admit a finite subcover. For metric spaces, equivalently $X$ is (sequentially) compact, if all sequences in $X$ have a subsequence converging to a limit in $X$ (see 17G.3 of Willard [2012]). ∎

The following is a characterization of compact subspaces of $\mathbb{R}^d$.

**Fact A.2.3.** (Heine-Borel) $X \subseteq \Omega$ is a compact subspace iff. $X$ is a bounded, closed subset of $\mathbb{R}^d$ (see 17.9 of *Willard [2012]*).

*Example* 4. Subspaces $[a, b], \{a\}, \{0, 2^{-n} : n \in \mathbb{N}\}$ are compact in $\mathbb{R}$. $(a, b), \{2^{-n} : n \in \mathbb{N}\}$ are not closed, and so they are not compact. $\mathbb{R}$ is not bounded, and so it is not compact. ∎

Turns out that compactness, unlike openness, is inherent to the subspace, as demonstrated by the following theorem:

**Fact A.2.4.** A continuous image of a compact space is compact (see 17.7 of *Willard [2012]*)

Finally, *Tychonoff* Theorem tells us that compactness is a property which is preserved by cartesian products.

**Fact A.2.5.** (Tychonoff) The cartesian product of two compact spaces is compact (see 17.8 of *Willard [2012]*)

### A.2.5 Path-connectedness

**Definition 10.** A *path* in $X$ from $a$ to $b$ is a continuous function $\gamma : [0,1] \to X$ such that $\gamma(0) = a$ and $\gamma(1) = b$. A space $X$ is called *path-connected* if for all $a, b \in X$ there is a path from $a$ to $b$. ∎

Path-connectedness partitions the space into components, which we will later think of as atomic parts of the state-space for a dynamical system. - any continuous decoder assigning discrete symbols to the state-space must be *constant* on a path-connected component, see Lemma 22.

See Section 27D of Willard [2012] for the following:

**Fact A.2.6.** The relation $\sim$ on $X$ given by $a \sim b \iff$ there is a path from $a$ to $b$ in $X$ is an equivalence. The equivalence classes of $\sim$ are the maximal path-connected subspaces of $X$.

*Example* 5. Any convex subspace of $\mathbb{R}^d$ is path-connected, in particular open and closed $\mathbb{R}^d$-balls are path-connected. $(-1, 0) \cup (0, 1)$ has 2 path-connected components: $(-1, 0)$ and $(0, 1)$. ∎

Just like compactness, path-connectedness is an inherent property of the subspace, and is preserved by Cartesian products (see 27B of Willard [2012]):

**Fact A.2.7.** A continuous image of a path-connected space is path-connected.

**Fact A.2.8.** The cartesian product of two path-connected spaces is path-connected.

### A.3 Dynamical Systems

Following Knorozova and Ronca [2024a], we adopt dynamical systems as an general formalism to describe all systems that operate by maintaining a state recurrently. This allows for treating such systems in a uniform way despite their differences. In this work specifically, we will use dynamical systems to formalise Finite Automata and several RNN architectures in Section A.6.

**Definition 11.** A *(dynamical) system* is a tuple $S = \langle X, U, f, x_0, Y, h \rangle$, where $X$ is the *state space*, $U$ is the *input space*, $f : X \times U \to X$ is the *dynamics function*, $x_0 \in X$ is the *initial state*, $Y$ is the *output space* and $h : X \times U \to Y$ is the output function. We have that $X, U, Y$ are metric spaces, and $f, h$ are *continuous*. In our analysis it will be useful to refer to the tuple $D = \langle X, U, f \rangle$ as the *dynamics* of $S$, allowing us to focus on just the state transitions.

Given $x_0 \in X$, $D$ defines a map from sequences of inputs $(u_n)_{n \geq 1} \subseteq U$ to sequences of states $(x_n)_{n \geq 0} \subseteq X$, given by

$$x_{n+1} = f(x_n, u_{n+1}) \quad \text{for } n \geq 0$$

With this, we can define the *state-sequence function* $D : X \times U^* \to X$ as

$$D(x_0, \varepsilon) = x_0; \quad D(x, u_{1..n}) = x_n$$

$S$ defines a map from sequences of inputs $(u_n)_{n \geq 1} \subseteq U$ to sequences of states $(x_n)_{n \geq 1} \subseteq X$ and sequences of outputs $(y_n)_{n \geq 1} \subseteq Y$, given by

$$y_n = h(x_n, u_n) = h\big(D(x_0, u_{[1..n]}), u_n\big)$$

Hence we say that $S$ *defines* the function $U^+ \to Y$, with $S(u_{[1..n]}) = y_n$. In the special case that $h$ is independent of $U$, we may define $S(\epsilon) = h(x_0)$, extending the definition to $S : U^* \to Y$. ∎

**Lemma 10** (State continuity). *Let $S = \langle X, U, f \rangle$ be a dynamics, and for input sequence $(u_n)_{n \geq 1}^N \subseteq U$ and $x_0 \in X$ let $(x_n)_{n \geq 1}^N \subseteq X$ be the sequence of states*

$$x_n = f(x_{n-1}, u_n)$$

*Then $x_n$ is a continuous function of $x_0, u_1, \ldots, u_n$ for all $n \in 1..N$. Consequently $y_n = h(x_n, u_n)$ is also a continuous function of $x_0, u_1, \ldots, u_n$, for any continuous $h$.*

*Proof.* By induction. Writing $x_n(u_1, \ldots, u_n)$ we have that

$$x_{n+1} = f(x_n(x_0, u_1, \ldots, u_n), u_{n+1})$$

is also a continuous function of $x_0, u_1, \ldots, u_{n+1}$. ☐

The formalism of cascades provides a flexible way to describe dynamical systems consisting of subsystems forming an acyclic network. Their flexibility will allows us, e.g., to consider not only feed-forward layers of SSMs as in Grazzi et al. [2025], Sarrof et al. [2024], but also more complex architectures with, e.g., blocks in parallel, and mixes of different types of neurons.

**Definition 12.** A *feed-forward cascade* C is a form of dynamics $\langle X, U, f \rangle$ with $X = X_1 \times \cdots \times X_n$, and dynamics function of the form

$$f(\langle x_1, \ldots, x_n \rangle, u) = \langle x'_1, \ldots, x'_n \rangle$$
$$\text{where} \quad x'_i = f(x_i, \langle u, x'_1, \ldots, x'_{i-1} \rangle)$$

We may see $C$ as consisting of dynamics $D_1, \ldots, D_n$ where

$$D_i = \langle X_i, U \times X_{[1,i-1]}, f_i \rangle$$

and write $C = D_1 \rightsquigarrow \cdots \rightsquigarrow D_n$. ∎

Thus, the cascade is evaluated in a feedforward fashion: on input $u$, first the state of $D_1$ is updated, then for all subsequent components $D_i$, the state of $D_i$ is updated based on $u$ and the *updated* states of $D_1, \ldots, D_{i-1}$. This differs from some recurrent neural network literature, where $D_i$ is updated based on $u$ and the *initial* states of $D_1, \ldots, D_{i-1}$, i.e. the update happens at the same time for all components. We refer to such cascades as *serial cascades*.

**Definition 13.** A *serial cascade* $C$ is a form of dynamics $\langle X, U, f \rangle$ where states are of the form $X = X_1 \times \cdots \times X_n$, and the dynamics function is of the form

$$f(\langle x_1, \ldots, x_n \rangle, u) = \langle f_1(x_1, u_1), \ldots, f_n(x_n, u_n) \rangle, \quad \text{with} \quad u_i = \langle u, x_1, \ldots, x_{i-1} \rangle.$$

We may see $C$ as consisting of dynamics $D_1, \ldots, D_n$ where

$$D_i = \langle X_i, U \times X_{[1,i-1]}, f_i \rangle$$

and write $C = D_1 \ltimes \cdots \ltimes D_n$. ∎

Serial cascading can be achieved with feed-forward cascades, and the distinction between the two is irrelevant for our purposes. For details, see Appendix G.2.

In further sections, it will be useful to allow *connection* functions in a cascade, transforming the inputs between components. It will not alter the expressivity results, but it allows us to e.g. define one canonical FLIP-FLOP dynamics, rather than a family of FLIP-FLOP-like dynamics for every possible input and output set.

**Definition 14.** For dynamics $D_1, D_2$ with $D_i = \langle X_i, U_i, f_i \rangle$ for all $i \in [1..2]$, and for continuous $i : U \to U_1$ and $g : U \times X_1 \to U_2$, we define the *feed-forward cascade with input $i$ and connection $g$*, written $\overset{i}{\rightsquigarrow} D_1 \overset{g}{\rightsquigarrow} D_2$, and the *serial cascade with input $i$ and connection $g$*, written $\overset{i}{\ltimes} D_1 \overset{g}{\ltimes} D_2$ as the dynamics $\langle X_1 \times X_2, U, f \rangle$, $\langle X_1 \times X_2, U, f' \rangle$ respectively, where $f$ and $f'$ are given by

$$f(\langle x_1, x_2 \rangle, u) = \langle x'_1, x'_2 \rangle, \quad \text{where}$$
$$x'_1 = f_1(x_1, i(u))$$
$$x'_2 = f_2(x_2, g(u, x'_1)),$$

and $f'(\langle x_1, x_2 \rangle, u) = \langle f_1(x_1, i(u)), f_2(x_2, g(u, x_1)) \rangle$. Note that for $U_2 = U_1 \times X_2$ and $g = \mathrm{id}$, we recover the usual notion of feed-forward cascade and serial cascade/. ∎

For dynamics $D = \langle X, U, f \rangle$ and continuous function $g : Z \to U$, we define the *dynamics with input function* $D_g = \langle X, Z, (x, z) \mapsto f(x, g(z)) \rangle$. With the notation from the previous definition, note that $D_{1,i} \rightsquigarrow D_{2,g} \equiv \overset{i}{\rightsquigarrow} D_1 \overset{g}{\rightsquigarrow} D_2$, and $D_{1,i} \ltimes D_{2,g} \equiv \overset{i}{\ltimes} D_1 \overset{g}{\ltimes} D_2$. In our expressivity results we will not care about how the dynamics of a neuron interpret the input function, only about the induced transformations of the state-space. Thus, in further sections in proofs we will only consider feed-forward cascading without connection functions, without loss of generality, in order to simplify notation. Further discussion about serial cascades and connecting functions is deferred to Appendix B.5. The next lemma shows the intuitive fact, that it does not matter in which order we "connect" the components of the cascade. In the following propositions, it will be useful to view a cascade $D_1 \rightsquigarrow \cdots \rightsquigarrow D_n$ as $(D_1 \rightsquigarrow \cdots \rightsquigarrow D_{n-1}) \rightsquigarrow D_n$ for inductive proofs.

**Definition 15.** For dynamics $D_1, D_2$, where $D_i = \langle X_i, U_i, f_i \rangle$ for all $i \in [1..2]$, write $D_1 \equiv D_2$ if $X_1 = X_2, U_1 = U_2$ and $f_1 = f_2$.

**Lemma 11.** *The cascading operation is* associative, *i.e. we have*

$$D_1 \rightsquigarrow (D_2 \rightsquigarrow D_3) \equiv (D_1 \rightsquigarrow D_2) \rightsquigarrow D_3,$$

*where '$\equiv$' is as introduced in Definition 15*

*Proof.* Say we have $D_i = \langle X_i, U \times X_{[1,i]}, f_i \rangle$ for $i \in 1..3$. Both the LHS and RHS dynamics have state space $X_1 \times X_2 \times X_3$ and input space $U$. Consider a state $\langle x_1, x_2, x_3 \rangle \in X_1 \times X_2 \times X_3$ and input $u \in U$.

Write $x_1' = f_1(x_1, u), x_2' = f_2(x_2, \langle u, x_1' \rangle), x_3' = f_3(x_3, \langle u, x_1', x_2' \rangle)$. Also write $f_{23}$ for the dynamics function of $D_2 \rightsquigarrow D_3$ and $f_{12}$ for the dynamics function of $D_1 \rightsquigarrow D_2$. Then the state update of the LHS system is as follows:

$$
\begin{aligned}
f_{LHS}(\langle x_1, x_2, x_3 \rangle, u) &= \left\langle x_1', f_{23}\big(\langle x_2, x_3 \rangle, \langle u, x_1' \rangle\big) \right\rangle \\
&= \left\langle x_1', \langle x_2', f_3\big(x_3, \langle u, x_1', x_2' \rangle\big) \rangle \right\rangle \\
&= \langle x_1', x_2', x_3' \rangle.
\end{aligned}
$$

where the second line follows from the definition of cascade dynamics for $D_2 \rightsquigarrow D_3$, and the third line follows from associativity of the cartesian product. Analogously,

$$f_{\text{RHS}}(\langle x_1, x_2, x_3 \rangle, u) = \left\langle x_{12}', f_3\big(x_3, \langle u, x_{12}' \rangle\big) \right\rangle, \quad \text{where} \quad x_{12}' = f_{12}(\langle x_1, x_2 \rangle, u).$$

Now, we have $x_{12}' = f_{12}(\langle x_1, x_2 \rangle, u) = \left\langle x_1', f_2\big(x_2, \langle u, x_1' \rangle\big) \right\rangle = \langle x_1', x_2' \rangle$, and so

$$
\begin{aligned}
f_{\text{RHS}}(\langle x_1, x_2, x_3 \rangle, u) &= \left\langle x_{12}', f_3\big(x_3, \langle u, x_{12}' \rangle\big) \right\rangle \\
&= \left\langle \langle x_1', x_2' \rangle, f_3\big(x_3, \langle u, x_1', x_2' \rangle\big) \right\rangle \\
&= \langle x_1', x_2', x_3' \rangle.
\end{aligned}
$$

Thus both ways of composing the dynamics $D_1, D_2, D_3$ results in the same dynamics function. $\square$

### A.4 Algebraic Automata Theory (AAT)

We present an extended version of the background on Algebraic Automata Theory given in the preliminaries of the main body.

Algebraic Automata Theory (AAT) allows for studying finite automata through the lens of algebraic notions such as semigroups and groups, c.f. [Hartmanis and Stearns, 1966, Ginzburg, 1968, Arbib, 1969, Dömösi and Nehaniv, 2005]. Its fundamental theorem is the seminal *Prime Decomposition Theorem* by Krohn and Rhodes [1965], that shows how every semiautomaton can be decomposed into a *cascade* of elementary *prime* semiautomata. One prime semiautomaton is the *flip-flop*, that describes the elementary system with the ability to store and manipulate one bit of information.

**Definition 16.** The *flip-flop* is the two-state semiautomaton defined as

$$\text{FLIP-FLOP} := \left\langle \{\text{high}, \text{low}\}, \{\text{set}, \text{reset}, \text{id}\}, \delta \right\rangle$$

where

$$\delta(q, \text{id}) = q, \qquad \delta(q, \text{set}) = \text{high}, \qquad \delta(q, \text{reset}) = \text{low}.$$

AAT often focuses on *state transformations* rather than on the transition function $\delta$ of an automaton. State transformations are the functions $\delta_\sigma(q) := \delta(q, \sigma)$ obtained by fixing an input $\sigma$. They allow us to characterise semiautomata in terms of semigroups and groups. In particular, the transitive closure of the state transformations of an automaton forms a semigroup, and a monoid or group in special

cases. From this algebraic point of view, the flip-flop is characterised by the *flip-flop* semigroup, which is in fact given by the set of state transformations of FLIP-FLOP. All the other primes are characterised by finite simple groups, and for this reason they are called *group-like*. Specifically, their state transformations form a finite simple group.

Automata whose semiautomaton can be decomposed purely into flip-flops are called *group-free*, and they play a central role in our theory and in general, due to the following theorem whose proof also involves the celebrated theorem by Schützenberger [1965]) on aperiodic semiautomata, cf. [Ginzburg, 1968].

**Theorem 12.** *The star-free languages is the class of languages recognised by groupfree automata.*

All other automata, that do not admit the above decomposition, are called *non-group-free*, since their prime decompositions always include group-like semiautomata. They admit the following characterisation in terms of state transformations, relevant to our results.

**Theorem 13.** (Lemma 9 of [Knorozova and Ronca, 2024a][1]) *If a semiautomaton $\langle Q, \Sigma, \delta \rangle$ is not group-free, then there exist $Q' \subseteq Q$ and $\sigma \in \Sigma$ such that the state transformation $\delta_\sigma : Q \to Q$ is a non-identity permutation on $Q'$.*

Our theory will extend the applicability of AAT to the study of general dynamical systems. And in particular to analyse the structure of such systems using algebraic means like group theory. A notion from AAT that is key to our results is the notion of *realisation* for Mealy machines (cf. Definitions 1.14 and 1.15 of [Hartmanis and Stearns, 1966]).

Realisation describes how a machine can imitate another machine after a renaming of inputs and outputs—noting that actual names of inputs and outputs are not important in order to characterise what functionalities a machine is fundamentally able to implement.

We recall that a *Mealy machine* is a tuple $\langle Q, \Sigma, \delta, \Gamma, \theta \rangle$ where $\langle Q, \Sigma, \delta \rangle$ is a semiautomaton, $\Gamma$ is an output alphabet, and $\theta : Q \times \Sigma \to \Gamma$ is an output function.

A Mealy machine defines the mapping $Q \times \Sigma^+ \to \Gamma$ given by

$$M(q, w) = \theta\big(D_M(q, w), w_{-1}\big),$$

where $D_M$ is the semiautomaton of $M$.

Given a (finite) automaton $A = \langle Q, \Sigma, \delta, q_0, \Gamma, \theta \rangle$, the *associate Mealy machine* $M_A = \langle Q, \Sigma, \delta, \Gamma, \theta \rangle$ is obtained by dropping the initial state from automaton $A$.

Given a semiautomaton $D_A = \langle Q, \Sigma, \delta \rangle$ we define its *canonical* Mealy machine as

$$\mathcal{M}(D) \coloneqq \langle Q, \Sigma, \delta, \Gamma, \theta \rangle, \quad \text{where } \Gamma = Q \times \Sigma, \text{ and } \theta = \mathrm{id}.$$

**Definition 17** (Definitions 1.14 and 1.15 of [Hartmanis and Stearns, 1966]). If $M = \langle Q, \Sigma, \delta, \Gamma, \theta \rangle$ and $M' = \langle Q', \Sigma', \delta', \Gamma', \theta' \rangle$ are Mealy machines, then the triple $(\alpha, \iota, \zeta)$ is called an *assignment* of $M$ into $M'$ when the functions

$$\alpha : Q \to \mathcal{P}_+(Q'), \quad \iota : \Sigma \to \Sigma', \quad \zeta : \Gamma' \to \Gamma,$$

satisfy the two conditions below for every $q \in Q$, every $q' \in \alpha(q)$, and every $\sigma \in \Sigma$.

$$\text{I)} \quad \delta'\big(q', \iota(\sigma)\big) \in \alpha\big(\delta(q, \sigma)\big)$$

$$\text{II)} \quad \zeta \circ \theta'\big(q', \iota(\sigma)\big) = \theta\big(q, \sigma\big)$$

If an assignment of $M$ into $M'$ exists, then $M'$ is said to be a *realisation* of $M$. ∎

The following results tells us how a machine $M'$ that is a realisation of another machine $M$ actually implements its behaviour. Any trajectory through $M$ factors through $M'$, with $\iota$ and $\zeta$ acting as the encoder and decoder, respectively, and with $\alpha$ providing an initial state to start from.

**Theorem 14.** *(Theorem 1.5 in §1.3 of [Hartmanis and Stearns, 1966]) If $M' = \langle Q', \Sigma', \delta', \Gamma', \theta' \rangle$ is a realisation of $M = \langle Q, \Sigma, \delta, \Gamma, \theta \rangle$ through an assignment $(\alpha, \iota, \zeta)$, then for all $x_0 \in Q$, $w \in \Sigma^+$, and $x_0' \in \alpha(x_0)$*

$$\theta\big(D(x_0, w), w_{-1}\big) = \zeta \circ \theta'\big(D'(x_0', \iota(w)), \iota(w_{-1})\big)$$

*i.e., $M(x_0, w) = \zeta \circ M'\big(q_0', \iota(w)\big)$.*

---

[1] Lemma 9 of [Knorozova and Ronca, 2024a] can be found in the appendix of its extended version [Knorozova and Ronca, 2023].

We will use the following version of the Krohn-Rhodes decomposition theorem, presented in [Hartmanis and Stearns, 1966], which uses the notion of realisability.

**Theorem 15.** *(Theorem 7.8, §8, Hartmanis and Stearns [1966]) Let $M$ be a Mealy machine, with group-free semiautomaton. Then $M$ can be realised by a machine with serial cascade dynamics, consisting of* FLIP-FLOP *components.*

### A.5 Multilayer Perceptrons

A *Multilayer Perceptron (MLP)* is a tuple

$$N = \langle d, \mathbf{n}, U, Y, \alpha, \beta, \mathbf{W}, \mathbf{b} \rangle,$$

where $d \in \mathbb{N}_{>0}$ is called the *depth* or *number of layers*, $\mathbf{n} = \langle n, n_2, n_3, \ldots, n_d, m \rangle$ is called *architecture*, $U \subseteq \mathbb{R}^n$ is the input domain, $Y \subseteq \mathbb{R}^m$ is the output domain (or codomain), $\alpha : \mathbb{R} \to \mathbb{R}$ is called *activation function*, $\beta : \mathbb{R} \to \mathbb{R}$ is called *activation function of the last layer*, $\mathbf{W} = \langle W_1, \ldots, W_d \rangle$ with $W_i \in \mathbb{R}^{n_i \times n_{i+1}}$ called *weight matrices*, and $\mathbf{b} = \langle b_1, \ldots, b_d \rangle$ with $b_i \in \mathbb{R}^{m_i}$ called *bias vectors*. Then, $N$ defines the function $f : U \subseteq \mathbb{R}^n \to U \subseteq \mathbb{R}^m$ given by the composition $f_1 \circ \cdots \circ f_d$ of the functions $f_i : \mathbb{R}^{n_i} \to \mathbb{R}^{n_{i+1}}$ defined as

$$f_i(x) = \alpha(W_i^\mathsf{T} x + b_i) \quad \forall i \in [1..d-1],$$
$$f_d(x) = \beta(W_d^\mathsf{T} x + b_d).$$

We often identify $N$ with the function $f$, and hence see the network as a function $N : U \to Y$. The functions $f_i$ are called *layers*, with the first layer $f_1$ called the *input layer*, the last layer $f_d$ called the *output layer*, and the other layers called *hidden layers*. The (maximum) *width* of $N$ is $\max\{n_2, \ldots, n_d\}$. Typical choices for the activation function $\alpha$ are $\mathrm{sigmoid}(x) \coloneqq \frac{1}{1+\exp(-x)}$ and the *Rectified Linear Unit* $\mathrm{ReLU}(x) \coloneqq \max\{0, x\}$. The same choices are valid for the last-layer activation function $\beta$; however, as it computes the output of the network, it is often specialised by choosing $\beta$ to be: the identity function (e.g., for regression tasks), $\mathrm{sigmoid}$ (e.g., for binary classification), softmax (e.g., for modelling distributions).

MLPs are universal approximators as long as their activation function $\alpha$ is *non-polynomial*, as established by several well-known Universal Approximation Theorems for feedforward neural networks, cf. [Cybenko, 1992, Hornik et al., 1989].

**Theorem 16** (Universal Approximation). *Let $\alpha$ be any non-polynomial activation function. Additionally, let $X \subseteq \mathbb{R}^n$ be compact, and let $f : X \subseteq \mathbb{R}^n \to Y \subseteq \mathbb{R}^m$ be continuous. For every $\epsilon > 0$, there exists a 2-layer MLP $N$ with activation function $\alpha$, and identity as its last-layer activation function, such that the following inequality holds:*

$$\sup_{x \in X} \|f(x) - N(x)\| < \epsilon.$$

Note that $\mathrm{ReLU}$ and $\mathrm{sigmoid}$ are non-polynomial activation functions.

In light of the above theorem, in the rest we will focus on MLPs having non-polynomial activation function $\alpha$, as well as identity as their last-layer activation function $\beta$. This will be relevant in all expressivity results for RNNs whose architecture includes MLPs—as also discussed in Section A.6.

### A.6 Recurrent Neural Network Architectures

We present the Recurrent Neural Network (RNN) architectures studied in the following sections.

Classical RNNs are networks of neurons with hidden state $h \in \mathbb{R}^{d_{\mathrm{state}}}$ and update rule of the form

$$h_t = \phi(h_{t-1}, x_t) \quad \text{for } x \in \mathbb{R}^{d_{\mathrm{input}}}$$

where $\phi$ is commonly a linear transformation composed with a non-linearity, like $\mathrm{sigmoid}$ or $\tanh$. We model such neurons as dynamical systems, with hidden state taking values in $X$, and inputs taking values in $U$. The hidden state of the neuron at step $t$ may be available to other neurons in the network as part of their input at time $t + 1$.

In modern Machine Learning applications, notably NLP, the networks are in the form of feed-forward connections, with learnable transformations between the neurons. Also some neurons may appear in

parallel, and some neurons might additionally include residual connections. Most generally, we can model such RNNs as acyclic networks, and for nodes $N_1, \ldots, N_L$ consider the connection functions $\psi_{i,j}$, describing the transformation which is applied to the value going from neuron $N_i$ to neuron $N_j$. The network input also may be given to $N_i$, after going through some transformation $\iota_i$. As the network is acyclic, we may assume that there are no connection functions $\psi_{i,j}$ for $i > j$. Finally, the inputs to $N_i$ are accumulated by some $\alpha_i$. Now, we may express the network as a feed-forward cascade $D_1 \rightsquigarrow \cdots \rightsquigarrow D_L$, with $D_i = \langle X_i, U \times X_{[1..n]}, f_i \rangle$, where $X_i$ is the state-space of neuron $N_i$, $U$ is the input space of the network, and $f_i$ is given by

$$f_i\big(x, x_{[1..n]}\big) = \phi\Big(h, \alpha_i\big(\langle \iota_i(u), \psi_{1,i}(x_1), \ldots, \psi_{i-1,i}(x_{i-1})\rangle\big)\Big)$$

This is how our framework allows to pull the details about the state-less transformations of the input or state-space into the dynamics function.

**Classical (Vanilla) RNNs.** Vanilla RNNs are networks where the state is updated through a linear combination of the previous state and current input, followed by the application of a non-linear activation function. A prominent example of a vanilla RNN architecture is the *Elman RNN*, which is given by dynamics $D = \langle X, U, f \rangle$ with state space $X \subseteq \mathbb{R}^{\text{state}}$, input space $U \subseteq \mathbb{R}^{\text{input}}$, and dynamics function

$$f(x, u) = \tanh\big(A_X \cdot x + A_U \cdot u + b\big),$$

where $A_X \in \mathbb{R}^{\text{state} \times \text{state}}$ is a matrix defining a linear transformation of the state, $A_U \in \mathbb{R}^{\text{state} \times \text{input}}$ is a matrix defining a linear transformation of the input, and $b \in \mathbb{R}^{\text{state}}$ is the bias vector.

**State Space Models.** *State Space Models (SSMs)* are a family of models based on linear recurrence with particular parametrisation. Notable ones are *Mamba* [Gu and Dao, 2023] and *S4* [Gu et al., 2020].

To model *linear recurrence* in general, we introduce *Linear Recurrent Dynamics*, defined as dynamics $D = \langle X, U, f \rangle$, with state space $X \subseteq \mathbb{K}^{d_{\text{state}}}$, input space $U = \mathbb{K}^{d_{\text{input}}}$, where $\mathbb{K} = \mathbb{R}$ or $\mathbb{K} = \mathbb{C}$, and with dynamics function

$$f(x, u) = A(u) \cdot x + B(u),$$

where $A(u) \in \mathbb{K}^{d_{\text{state}} \times d_{\text{state}}}$ is the *state-transition gate* and $B(u) \in \mathbb{K}^{d_{\text{state}}}$ is the *input gate*.

SSM architectures often combine linear recurrence blocks with linear projections, non-linearities, residual connections and convolutions. Our theory can easily model such setups with cascade compositions—introduced in Section 2. Consider the Mamba block:

$$z_{[1..n]} = \text{SSM} \circ \sigma \circ \text{Conv} \circ \text{linear}_1(u_{[1..n]})$$
$$y_{[1..n]} = \sigma \circ \text{linear}_2(u_{[1..n]})$$
$$o_{[1..n]} = \text{linear}_3(z_{[1..n]} \times y_{[1..n]})$$

where the input sequence $u_{[1..n]} \in U^+$ and output sequence $o_{[1..n]} \in Y^+$ are processed sequentially, each $\text{linear}_i$ is a linear projection, $\sigma$ is a non-linearity, SSM is an SSM block, Conv is a *causal* convolution, and $\times$ is element-wise multiplication. Only Conv and SSM are stateful transformations here. In Figure 6, we present it in the form of a system with cascade dynamics.

We introduce a general class of dynamics as an abstraction for convolution blocks.

**Definition 18.** *Finite Context Dynamics (FCDs)* with context length $\ell$ are dynamics $D = \langle X, U, f \rangle$ such that their state depends only on the most recent $\ell$ inputs. That is, in view of Lemma 10, there is a continuous function $C : U^\ell \to X$ such that

$$D(x, w) = C(w_{-u}, \ldots, w_{-1})$$

for all $x \in X$ and $w \in U^*$ with $|w| \geq \ell$, where $w_{-i}$ is the $i$-th-to-last element of $w$.

**xLSTM.** The recently introduced model xLSTM [Beck et al., 2024] is a successor of the LSTM architecture [Hochreiter and Schmidhuber, 1997], and it achieves performance competitive with transformer architectures. It makes use of both non-linear and linear recurrences. xLSTM introduces two types of blocks: sLSTM and mLSTM. In this work we will focus on the sLSTM block.

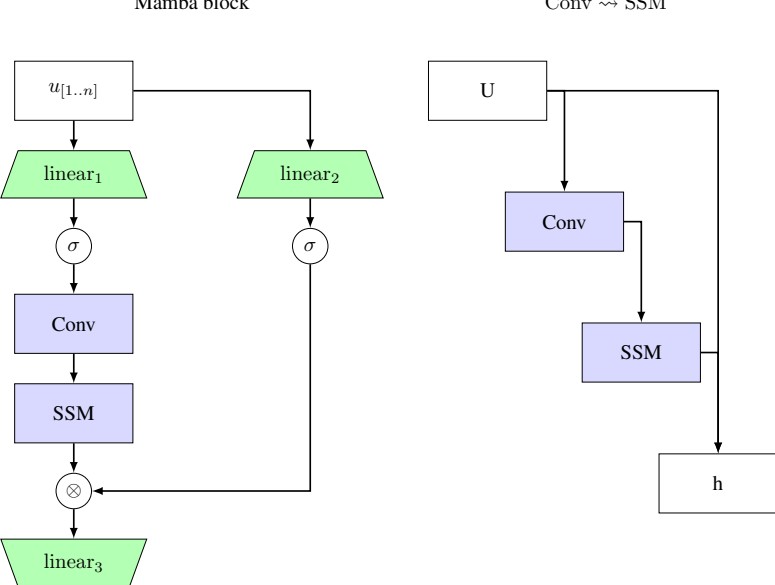

Figure 6: The feedforward cascade structure of a Mamba block. Only Conv and SSM are stateful, so the cascade has 2 components. Structure on the left as it is presented in [Gu and Dao, 2023].

The state space of a sLSTM is $\mathbb{R}^3$, and the input space is $\mathbb{R}^d$ for some $d \geq 1$. The dynamics function of the form $(\langle c, n, h \rangle, u) \mapsto \langle f_c(\langle c, n, h \rangle, u), f_n(\langle c, n, h \rangle, u), f_h(\langle c, n, h \rangle, u) \rangle$, where

$$f_c(\langle c, n, h \rangle, u) = \psi(l_f(h, u)) \cdot c + \exp(l_i(h, u)) \cdot \varphi(l_z(h, u))$$
$$f_n(\langle c, n, h \rangle, u) = \psi(l_f(h, u)) \cdot n + \exp(l_i(h, u))$$
$$f_h(\langle c, n, h \rangle, u) = \sigma(l_o(h, u)) \cdot \frac{f_c(\langle c, n, h \rangle, x)}{f_n(\langle c, n, h \rangle, x)}$$

where each $l_s : s \in o, i, z, f$ is a function of the form $w_s^t \cdot u + r_s \cdot h + b_s$, for $w_s \in \mathbb{R}^d$, $r_s, b_s \in \mathbb{R}$, $\psi$ is either $\exp$ or $\sigma$, and $\varphi$ is $\tanh$.

# B Foundations of Metric Automata Theory

In this appendix, we develop the key notions of Metric Automata Theory within the $\eta$-finiteness framework.

In Sections B.1 and B.2 we introduce the basic properties of $\eta$-finite spaces and dynamics.

In Section B.3 we develop the correspondence between $\eta$-finite systems and finite automata, which is crucial to unlocking the powerful theorems of AAT. We provide the proof for Theorem 1.

In Sections B.4 and B.5 we import the notion *realizability* to continuous systems, via the correspondence with automata, and use it to translate algebraic decomposition theorems into the setting of $\eta$-finiteness.

## B.1 The Notion of $\eta$-Finiteness

We begin by introducing $\eta$-*finiteness*, which is a central notion of Metric Automata Theory and our novel finite-precision framework.

**Definition 19.** Let $X \subseteq \Omega$ for some $d \geq 1$. Call $X$ $\eta$-*finite* if it is a finite union of compact, path-connected sets.

Immediately from the definition we have that an $\eta$-finite space is necessarily *compact*—in the case of metric spaces, finite union of bounded, closed sets is bounded and closed. The next result resolves the technicality, that the defining sets in the union of a $\eta$-finite $X$ need not be disjoint.

**Lemma 17.** *Let $X$ be $\eta$-finite. Then $X$ has finitely many path connected components, say $X_1, \ldots, X_n$, and each of $X_i$ is compact. We shall refer to them as the $\eta$-components of $X$.*

*Proof.* By def, $X = \bigcup_{i=1}^{N} Y_i$ for some compact and path-connected subsets. By induction on $N$: If $N = 1$, then the claim is immediate. Now, consider the inductive hypothesis for $N \geq 2$, that $X' = \bigcup_{i=1}^{N-1} Y_i$ has finitely many path connected components $X_1, \ldots, X_n$, each compact. The path connected components of $X$ are then unions of elements from $\{X_1, \ldots, X_n, Y_N\}$. Each of these sets is compact, and so each such finite union is compact: clearly it is still bounded, and a finite union of closed sets is still closed. $\square$

*Example* 6. *Any finite alphabet is $\eta$-finite*, with each symbol in a separate $\eta$-component. The subspace $[-2, 1] \cup \{2\} \subseteq \mathbb{R}$ is $\eta$-finite. The subspace $(-2, 1) \cup \{2\}$ is not $\eta$-finite, since it is not compact. The subspace $\{0, 2^{-n} : n \in \mathbb{N}\}$ is compact but not $\eta$-finite, since it is not a finite union of path-connected sets. $\blacksquare$

Both compactness and path-connectedness are preserved by continuous mappings and (finite) Cartesian products, see Facts A.2.4, A.2.5, A.2.7, and A.2.8. This gives us the corresponding results for $\eta$-finite spaces.

**Lemma 18.** *Continuous image of an $\eta$-finite space is $\eta$-finite.*

*Proof.* Write $X = \bigcup_{i=1}^{N} X_i$ for path-connected, compact sets $X_i$. Let $f : X \to Y$ be continuous. We have:

$$f(X) = \bigcup_{i=1}^{N} f(X_i)$$

By Facts A.2.4 and A.2.7, each $f(X_i)$ is compact and path-connected. Thus by definition $f(X)$ is $\eta$-finite. $\square$

**Lemma 19.** *The Cartesian product $X \times Y$ space of $\eta$-finite spaces is $\eta$-finite. The $\eta$-components of $X \times Y$ are the products of $\eta$-components of $X$ and $\eta$-components of $Y$.*

*Proof.* Let $X_1, \ldots, X_n$ and $Y_1, \ldots, Y_,$ be the C-components of $X, Y$ respectively. We have $X = \bigcup_{i=1}^{n} X_i, Y = \bigcup_{j=1}^{m} Y_j$ and so

$$X \times Y = \left( \bigcup_{i=1}^{n} X_i \right) \times \left( \bigcup_{j=1}^{m} Y_j \right) = \bigcup_{i=1}^{n} \bigcup_{j=1}^{m} X_i \times Y_j$$

By Facts A.2.8 and A.2.5 each $X_i \times Y_j$ is path-connected. Therefore by def. $X \times Y$ is $\eta$-finite. Moreover, the $\eta$-components of $X \times Y$ are unions of the products $X_i \times Y_j$. Now, fix $i \in [1..n], j \in [1..j]$. Let $Z$ be the $\eta$-componentof $X \times Y$ containing $X_i \times Y_j$. consider the projection map $\pi_X : X \times Y \to X$. As the projection is continuous, the image, $\pi_X(Z)$ is path-connected in $X$ by Fact A.2.7. Moreover, $X_i \in \pi_X(Z)$. Thus, as $X_i$ is a maximal path-connected subspace of $X$, we have $X_i = \pi_X(Z)$. Similarly, considering the projection $\pi_Y : X \times Y \to X$, we have $Y_j = \pi_X(Z)$. Since $X_i \times Y_j \subseteq Z$, we therefore must have $X_i \times Y_j = Z$. Therefore $X \times Y$ has finitely many $\eta$-components, and they are the products of $\eta$-components of $X$ and $\eta$-components of $Y$. $\qquad\square$

**Lemma 20.** *Let $X$ be $\eta$-finite, with $\eta$-component $X_1, \ldots, X_n$. For some $\delta > 0$ we have*

$$\inf_{x \in X_i, y \in X_j} \|x - y\| \geq \delta \quad \text{for all } i \neq j.$$

*Proof.* It is sufficient to show this in the case that $X$ has two $\eta$-components, say $X_1, X_2$. Define $f : X_1 \times X_2 \to \mathbb{R}_{\geq 0}$ by $f(x_1, x_2) = \|x_1 - x_2\|$. This is continuous, and so Im $f$ is compact, as $X_1 \times X_2$ is compact. Since $X_1, X_2$ are disjoint, $0 \notin \text{Im } f$. Thus 0 is not a limit point of Im $f$, and so for some $\delta > 0$ we have that $[0, \delta) \not\subseteq \text{Im } f$. $\qquad\square$

**Corollary 21.** *Let $X \subseteq \Omega$ be $\eta$-finite and $(x_n)_{n \geq 1} \subseteq X$ converge in $\Omega$. Then $(x_n)_{n \geq 1}$ is eventually contained in a single $\eta$-component of $X$.*

**Lemma 22.** *Let $X$ be an $\eta$-finite space and $\Sigma$ a finite alphabet. Then a function $f : X \to \Sigma$ is continuous if and only if it is constant on the $\eta$-components of $X$*

*Proof.* ($\Leftarrow$) Suppose $f : X \to \Sigma$ is constant on $\eta$-components of $X$. Let $(x_n)_{n \geq 1} \subseteq X$ converge to $x \in X$. Then by Lemma 20, $(x_n)_{n \geq 1}$ is eventually contained in the same $\eta$-component as $x$. Thus $f(x_n) = f(x)$ eventually, in particular $f(x_n) \to f(x)$ as $n \to \infty$. Hence $f$ is continuous.

($\Rightarrow$) If $f$ is continuous, then it maps $\eta$-component of $X$ to path-connected subspaces of $\Sigma$. Therefore $f$ must be constant on $\eta$-components. $\qquad\square$

## B.2 Dynamical Systems and $\eta$-Finiteness

**Definition 20.** We say that dynamics $\langle X, U, f \rangle$ are *$\eta$-finite* if both $X$ and $U$ are $\eta$-finite. A system $S$ is *$\eta$-finite* if its dynamics are $\eta$-finite.

*Example 7.* Take $X = [-1, -1/2] \cup [1/2, 1]$ and $U = \{-1, 0, 1\}$. The both $X$ and $U$ are $\eta$-finite. Define $f : X \times U \to X$ by:

$$f(x, u) = \begin{cases} x & \text{if } u = 0 \\ u & \text{if } u = 1, -1 \end{cases}$$

Thus under input $u = 0$ the dynamics function performs the identity transformation on $X$, and under inputs $u = 1, -1$, $X$ is mapped to $1, -1$ respectively. The dynamics $D = \langle X, U, f \rangle$ is $\eta$-finite. $\qquad\blacksquare$

Note, that by Lemma 19, a cascade of $\eta$-finite components is itself $\eta$-finite.

**Lemma 23.** *Let $D = \langle X, U, f \rangle$ be a $\eta$-finite dynamics, and $h : X \times U \to Y$ be continuous. Then the image of $h$, $\text{Im } h \subseteq Y$, is $\eta$-finite.*

*Proof.* Immediately follows from Lemma 18. $\qquad\square$

**Lemma 24** (Path-connected $\Rightarrow$ same state)**.** *Let $D = \langle X, U, f \rangle$ be a dynamics, and consider $x_0, x_0' \in X$, and input sequences $(u_n)_{n \geq 1}, (u_n')_{n \geq 1} \subseteq U$, and the corresponding state sequences $(x_n)_{n \geq 1}, (x_n')_{n \geq 1} \subseteq X$. Suppose that for all $n \geq 1$, $u_n \sim_U u_n'$, and $x_0 \sim_X x_0'$. Then for all $n \geq 1$ we have that $x_n \sim_X x_n'$, i.e.,*

$$D(x_0, u_{[1..n]}) \sim_X D(x_0', u_{[1..n]}')$$

*Proof.* Let $n \geq 1$. By 10, we have that there is for each $n$ a continuous function $x_n(x_0, u_1, ..u_n)$ determining the $n$-th state. Now, since each pair $u_i, u_i'$ for $i \in 1..n$ is path-connected in $U$, we have

that $\langle u_{1..n} \rangle$ and $\langle u'_{1..n} \rangle$ are path-connected in $U^n$ - the path connecting them applies the corresponding 1-d paths pointwise. Thus by continuity of $x_n$,

$$x_n = x_n(x_0, \langle u_{1..n} \rangle), \ x'_n = x_n(x_0, \langle u'_{1..n} \rangle)$$

are path-connected in $X$. $\qquad\qquad\qquad\qquad\qquad\qquad\qquad\qquad\qquad\qquad\qquad\qquad$ $\square$

**Corollary 25.** *Let $S = \langle X, U, f, x_0, Y, h \rangle$ be a $\eta$-finite system, and let us consider input sequences $(u_n)_{n \geq 1}, (u'_n)_{n \geq 1} \subseteq U$ such that for all $n$ $u_n$ and $u'_n$ are in the same path-connected component. Then the corresponding state sequences $(x_n)_{n \geq 1}, (x'_n)_{n \geq 1} \subseteq X$, and the corresponding output sequences $(y_n)_{n \geq 1}, (y'_n)_{n \geq 1} \subseteq Y$ are such that for all $n$ $x_n$ and $x'_n$ are in the same path-connected component of $X$ and $y_n$ and $y'_n$ are in the same path-connected component of $\mathrm{Im}\, h$*

In light of the above results, we introduce the notion of *equivalent* sequences, for convenience in later proofs.

**Definition 21.** *Let $X$ be a $\eta$-finite space. Call sequences $(x_n)_{n \geq 1}, (x'_n)_{n \geq 1} \subseteq X$ equivalent, if for each $n$ we have that $x_n$ and $x'_n$ are in the same component of $\bar{X}$. Call these sequences eventually equivalent, if they have equivalent tail sequences.*

Overall, the notions of $\eta$-finiteness and $\eta$-component have *very favourable theoretical properties*. Any continuous mapping $f : X \to Y$, with $X$ and $Y$ $\eta$-finite, is guaranteed to map every element of an $\eta$-component of $X$ into a single $\eta$-component of $Y$.

In the case of $\eta$-finite systems, this means that the dynamics function acts on the $\eta$-components of the state-space (referred to as $\eta$-states) in the same way for each input within an $\eta$-component of the input-space (referred to as $\eta$-input). Moreover, every point within an $\eta$-component of the output function image (which is always $\eta$-finite), must be decoded as the same alphabet symbol. We formalize these properties in the following section.

## B.3 Representing $\eta$-Finite Systems as Automata and Proof of Theorem 1

For set $A$ and equivalence $\sim$ on $A$, write $A/_\sim$ for the set of its equivalence classes. For $a \in A$ write $[a]_A$ for the $\sim$-equivalence class containing $a$.

For $\eta$-finite spaces $A$, we will write $\bar{A}$ for the set $A/_{\sim_A}$, with $\sim_A$ being the path-connectedness equivalence. For $X, Y$ being $\eta$-finite spaces, we have by Lemma 19 that $\overline{X \times Y} = \bar{X} \times \bar{Y}$.

**Definition 22.** *Any $\eta$-finite dynamical system $S = \langle X, U, f, x_0, Y, h \rangle$ defines its canonical automaton*

$$A_S = \langle \bar{X}, \bar{U}, \bar{f}, [x_0]_{\sim_X}, \overline{\mathrm{Im}\, h}, \bar{h} \rangle$$

*Similarly, any $\eta$-finite dynamics $D = \langle X, U, f \rangle$ defines its canonical semiautomaton $D_A = \langle \bar{X}, \bar{U}, \bar{f} \rangle$.* $\qquad\qquad\qquad\qquad\qquad\qquad\qquad\qquad\qquad$ $\blacksquare$

Note that by Lemma 23, $\mathrm{Im}\, h$ is indeed $\eta$-finite. $\bar{f} : (\bar{X}) \times (\bar{U}) \to (\bar{X})$ is defined as $[x]_{\sim_X}, [u]_{\sim_U} \mapsto [f(x,u)]_{\sim_X}$. $\bar{h} : \bar{X} \times \bar{U} \to \overline{\mathrm{Im}\, h}$ is defined as $[x]_{\sim_X}, [u]_{\sim_U} \mapsto [h(x,u)]_{\sim_{\mathrm{Im}\, h}}$. This is well defined by Lemma 25.

For a $\eta$-finite dynamical system $S = \langle X, U, f, x_0, Y, h \rangle$, define the *canonical* regular function $F_S : (\bar{U})^+ \to \overline{\mathrm{Im}\, h}$ to be the function defined by the FSA $A_S$. The following lemma shows that the dynamics of the canonical automaton determine—up to path-connectedness—the dynamics of the system.

**Lemma 26.** *Let $D = \langle X, U, f \rangle$ be a $\eta$-finite dynamics, and $D_A$ be its canonical semiautomaton. Then*

$$D_A([x_0]_{\sim_X}, [w]_{\sim_U}) = \left[ D(x_0, w) \right]_{\sim_X} \qquad \forall w \in U^* \tag{1}$$

*where $[w]_{\sim_U} \in U^*$ denotes the word with each letter of $w$ replaced by its equivalence class.*

*Proof.* By induction on the length of $w$. For the base case $w = \varepsilon$, we have $D_A\big([x_0]_{\sim_X}, [\varepsilon]_{\sim_U}\big) = D_A\big([x_0]_{\sim_X}, \varepsilon\big) = [x_0]_{\sim_X}$ and by definition $D(x_0, \varepsilon) = x_0$, so that $\big[D(x_0, \varepsilon)\big]_{\sim_X} = [x_0]_{\sim_X}$.

Now, suppose for $w \in U^*$ we have $D_A\big([x_0]_{\sim_X}, [w]_{\sim_U}\big) = \big[D(x_0, w)\big]_{\sim_X}$, and let $[u]_{\sim_U} \in \overline{U}$. Write $w[u]_{\sim_U}$ for the word obtained by appending $[u]_{\sim_U}$ at the end of $w$, we have

$$
\begin{aligned}
D_A\big([x_0]_{\sim_X}, [wu]_{\sim_U}\big) &= \overline{f}\big(D_A([x_0]_{\sim_X}, [w]_{\sim_U}), [u]_{\sim_U}\big) \\
&= \overline{f}\big(\big[D(x_0, w)\big]_{\sim_X}, [u]_{\sim_U}\big) \\
\text{by def. of } \overline{f} \quad &= \big[f\big(D(x_0, w, u)\big)\big]_{\sim_X} \\
&= \big[D(x_0, wu)\big]_{\sim_X}
\end{aligned}
$$

Thus by induction the statement holds for all $w \in U^*$. $\qquad\square$

**Lemma 27.** *Let $S$ be a $\eta$-finite system and $F_S$ be its canonical regular function. Then, $F_S$ is implemented by $S$ with encoder $\overline{\mathrm{enc}} : \overline{U} \to U$ given by $[u]_{\sim_U} \mapsto u'$ with $u' \in [u]_{\sim_U}$ chosen arbitrarily, and with decoder $\overline{\mathrm{dec}} : \mathrm{Im}\, h \to \overline{\mathrm{Im}\, h}$, given by $y \mapsto [y]_{\sim_{\mathrm{Im}\, h}}$.*

*Proof.* $\overline{\mathrm{enc}}$ is continuous, since $\overline{U}$ is a finite alphabet. $\overline{\mathrm{dec}}$ is continuous by Lemma 22. Let $D_A$ be the dynamics of $A_S$, and let $D_S$ be the dynamics of $S$. Then we have

$$
F_S(w) = \overline{h}\Big(D_A\big([x_0]_{\sim_X}, w\big), w_{-1}\Big) \quad \forall w \in \overline{U}^+
$$

where $w_{-1}$ denotes the last symbol in word $w$. Now consider $w \in \big(U/{\sim_U}\big)^+$ and write $[u]_{\sim_U}$ for $w_{-1}$. By Lemma 26, we have $D_A\big([x_0]_{\sim_X}, w\big) = \big[D_S\big(x_0, \overline{\mathrm{enc}}(w)\big)\big]_{\sim_X}$, so that

$$
\begin{aligned}
\overline{h}\Big(D_A\big([x_0]_{\sim_X}, w\big), w_{-1}\Big) &= \overline{h}\Big(\big[D_S\big(x_0, \overline{\mathrm{enc}}(w)\big)\big]_{\sim_X}, [u]_{\sim_U}\Big) \\
\text{as } u' = \overline{\mathrm{enc}}\big([u]_{\sim_U}\big) \in [u]_{\sim_U} \quad &= \overline{h}\Big(\big[D_S\big(x_0, \overline{\mathrm{enc}}(w)\big)\big]_{\sim_X}, [u']_{\sim_U}\Big) \\
\text{by def. of } \overline{h} \quad &= \Big[h\Big(D_S\big(x_0, \overline{\mathrm{enc}}(w)\big), u'\Big)\Big]_{\mathrm{Im}\, h} \\
&= \Big[h\Big(D_S\big(x_0, \overline{\mathrm{enc}}(w)\big), \overline{\mathrm{enc}}(w_{-1})\Big)\Big]_{\mathrm{Im}\, h} \\
&= \Big[S\big(\overline{\mathrm{enc}}(w)\big)\Big]_{\mathrm{Im}\, h} = \overline{\mathrm{dec}} \circ S\big(\overline{\mathrm{enc}}(w)\big)
\end{aligned}
$$

This concludes the proof. $\qquad\square$

**Lemma 28.** *Let $\eta$-finite system $S = \langle X, U, f, x_0, Y, h \rangle$ implement function $F : \Sigma^+ \to \Gamma$ with encoder $\mathrm{enc} : \Sigma \to U$ and decoder $\mathrm{dec} : \mathrm{Im}\, h \to \Gamma$. Then there are (continuous) functions $\mathrm{enc}' : \Sigma \to \overline{U}$ and $\mathrm{dec}' : \overline{\mathrm{Im}\, h} \to \Gamma$ such that*

$$
F(w) = \mathrm{dec}' \circ F_S(\mathrm{enc}'(w)) \quad \forall w \in \Sigma^+
$$

*where $F_S : \big(\overline{U}\big)^+ \to \big(\overline{\mathrm{Im}\, h}\big)$ is the canonical function for $S$.*

*Proof.* Define $\mathrm{enc}'$ as $\sigma \mapsto \big[\mathrm{enc}(\sigma)\big]_{\sim_U}$ for all $\sigma \in \Sigma$.

As for $\mathrm{dec}'$, define it as $[y]_{\sim_{\mathrm{Im}\, h}} \mapsto \mathrm{dec}(y)$. This is well-defined: Consider $y_1, y_2 \in \mathrm{Im}\, h$ such that $y_1, y_2 \in [y]_{\sim_{\mathrm{Im}\, h}}$. Since $y_1, y_2$ are path-connected in $\mathrm{Im}\, h$, by continuity of $\mathrm{dec} : \mathrm{Im}\, h \to \Gamma$ we have that $h(y_1), h(y_2)$ are path-connected in $\Gamma$. Therefore necessarily $h(y_1) = h(y_2)$.

Let $A_S$ be the canonical FSA of $S$. Denote the dynamics of $S$ as $D_S$ and the dynamics of $A_S$ as $D_A$. By Lemma 26, we have

$$D_A\big([x_0]_{\sim_X}, \mathrm{enc}'(w)\big) = \big[D_S\big(x_0, \mathrm{enc}(w)\big)\big]_{\sim_X} \quad \forall w \in \Sigma^+$$

Thus we have for all $w \in \Sigma^+$

$$\begin{aligned}
\mathrm{dec}' \circ F_S\big(\mathrm{enc}'(w)\big) &= \mathrm{dec}' \circ \overline{h}\Big(D_A\big([x_0]_{\sim_X}, \mathrm{enc}'(w)\big), \mathrm{enc}'(w_{-1})\Big) \\
&= \mathrm{dec}' \circ \overline{h}\Big(\big[D_S\big(x_0, \mathrm{enc}(w)\big)\big]_{\sim_X}, \big[\mathrm{enc}(w_{-1})\big]_{\sim_U}\Big) \\
&= \mathrm{dec}' \Big[h\Big(D_S\big(x_0, \mathrm{enc}(w)\big), \mathrm{enc}(w_{-1})\Big)\Big]_{\sim_{\mathrm{Im}\,h}} \\
&= \mathrm{dec} \circ S\big(\mathrm{enc}(w)\big)
\end{aligned}$$

Finally, $\mathrm{enc}'$ and $\mathrm{dec}'$ are continuous, since their domains are finite alphabets. $\qquad\square$

**Theorem 1.** *An $\eta$-finite system $S$ can-implement the same functions as its canonical automaton, which are necessarily regular.*

*Proof.* Suppose $S = \langle X, U, f, x_0, Y, h \rangle$ implements a function $F : \Sigma \to \Gamma$, with encoder $\mathrm{enc} : \Sigma \to U$ and decoder $\mathrm{dec} : Y \to \Gamma$. By Lemma 28, we have that the canonical FSA of $S$, say $A_S = \langle \overline{X}, \overline{U}, \overline{f}, [x_0]_{\sim_X}, \overline{\mathrm{Im}}h, \overline{h} \rangle$, implements $F$ with encoder $\mathrm{enc}'$ and decoder $\mathrm{dec}'$.

Moreover, consider the FSA $A' = \langle \overline{X}, \Sigma, \delta, [x_0]_{\sim_X}, \Gamma, \theta \rangle$, where $\delta : \overline{X} \times \Sigma \to \overline{X}$ is given by

$$\delta([x]_{\sim_X}, \sigma) = \overline{f}([x]_{\sim_X}, \mathrm{enc}'(\sigma))$$

and $\theta : \overline{X} \times \Sigma \to \Gamma$ is given by

$$\theta([x]_{\sim_X}, \sigma) = \mathrm{dec}' \circ \overline{h}\big([x]_{\sim_X}, \mathrm{enc}'(\sigma)\big)$$

Then we have that $F(w) = A'(w)$ for all $w \in \Sigma^+$. Thus $F$ is necessarily regular.

Now, suppose that $A_S$ implements a function $F : \Sigma \to \Gamma$, with encoder $\mathrm{enc} : \Sigma \to U$ and decoder $\mathrm{dec} : \mathrm{Im}\,h \to \Gamma$. By Lemma 27, $S$ implements $F_S$ with encoder $\overline{\mathrm{enc}}$ and decoder $\overline{\mathrm{dec}}$. Thus we have the following: for all $w \in \Sigma^+$

$$\begin{aligned}
F(w) &= \mathrm{dec} \circ A_S\big(\mathrm{enc}(w)\big) \\
&= \mathrm{dec} \circ F_S\big(\mathrm{enc}(w)\big) \\
&= \mathrm{dec} \circ \overline{\mathrm{dec}} \circ \big(\overline{\mathrm{enc}} \circ \mathrm{enc}(w)\big)
\end{aligned}$$

so that $S$ implements $F$ with encoder $\overline{\mathrm{enc}} \circ \mathrm{enc}$ and decoder $\mathrm{dec} \circ \overline{\mathrm{dec}}$. $\qquad\square$

## B.4 Algebraic Theory of $\eta$-Finite Systems

The connection between $\eta$-finite systems and canonical automata is extremely useful. It gives us a way to employ the powerful characterisations and results of AAT to any $\eta$-finite system dynamics. Namely, we can extend the notion of *realisability* to continuous $\eta$-finite systems, via the canonical automaton.

**Definition 23.** We say that $\eta$-*finite* dynamics $D'$ are a realisation of $\eta$-*finite* dynamics $D$ when $\mathcal{M}(\mathcal{C}(D'))$ is a realisation of $\mathcal{M}(\mathcal{C}(D))$ of $D$.

We that automaton $A'$ is a realisation of system $A$, if the associated machine $M_{A'}$ is a realisation of of the associated machine $M_A$ via an assignment $(\alpha, \iota, \zeta)$, and the respective initial states $x_0', x_0$ are such that $x_0' \in \alpha(x_0)$.

Say that $\eta$-*finite* system $S'$ is a realisation of system $S$, if $A_{S'}$ is a realisation of $A_S$, where $A_S, A_{S'}$ are the canonical automata. $\qquad\blacksquare$

The notion of realisation for machines is transitive. See §1.3 of Hartmanis and Stearns [1966].

**Fact B.4.1.** If $M$ is a realisation of $M'$ and $M'$ is a realisation of $M''$, then $M$ realies $M''$.

It is easy to see that the notion of realisation for dynamics and systems is also transitive.

**Lemma 29.** *Suppose that semiautomaton $D'$ is a realisation of semiautomaton $D$. Then*

*1) for any machine $M$ with dynamics $D$, the canonical machine $\mathcal{M}(D')$ of $D'$ is a realisation of $M$,*

*2) for any automaton $A$ with dynamics $D$, an initial state can be picked for $\mathcal{M}(D')$ such that the resulting automaton is a realisation of $A$.*

*Proof.* Say $D = \langle Q, \Sigma, \delta \rangle$ and $D' = \langle Q', \Sigma', \delta' \rangle$. Suppose we have an assignment $(\alpha, \iota, \zeta)$ from $D$ to $D'$. That is, $\alpha : Q \to \mathcal{P}_+(Q')$, $\iota : \Sigma \to \Sigma'$, $\zeta : Q' \times \Sigma' \to Q \times \Sigma$

Let $M = \langle Q, \Sigma, \delta, \Gamma, \theta \rangle$ be a Mealy machine with semiautomaton $D$. The canonical machine for $D'$ is

$$\mathcal{M}(D') = \langle Q', \Sigma', \delta', \Gamma' = Q' \times \Sigma', \theta' = \mathrm{id} \rangle$$

Define $\zeta' : (Q' \times \Sigma') \to \Gamma$ by $\zeta' = \theta \circ \zeta$. Want to show: $(\alpha, \iota, \zeta')$ give an assignment of $M$ into $\mathcal{M}(D')$. We already have that the condition I) is satisfied.

Now, for any $q \in Q, \sigma \in \Sigma$ and $q' \in \alpha(Q)$ we have that $\zeta \circ \theta'(q', \iota(\sigma)) = \mathrm{id}(q, \sigma) = (q, \sigma)$, since $(\alpha, \iota, \zeta)$ give an assignment of $D$ into $D'$. Thus

$$\zeta' \circ \theta'(q', \iota(\sigma)) = \theta \circ \zeta \circ \theta'(q, \sigma) = \theta(q, \sigma)$$

So $(\alpha, \iota, \zeta')$ also satisfy condition II). Thus the 1) part of the statement holds.

Now for the part 2): Let $A = \langle Q, \Sigma, \delta, q_0, \Gamma, \theta \rangle$ be a system with dynamics $D$. By part 1), the associated machine $M_A = \langle Q, \Sigma, \delta, q_0, \Gamma, \theta \rangle$ has some assignment $(\alpha, \iota, \zeta)$ into $M_{D'}$. $\alpha(x_0)$ is a non-empty set, and so we may arbitrarily pick $x_0' \in \alpha(x_0)$. Then the automaton $A' = \langle Q', \Sigma', \delta', q_0', Q' \times \Sigma', \mathrm{id} \rangle$ obtained from setting initial state $x_0'$ for machine $M_{D'}$, by definition is a realisation of $A$. $\square$

We have the following proposition to connect our notion of dynamical systems with Algebraic Automata Theory.

Before proceeding, we remark that Definition 1 must be made fully precise by saying that a decoder is a function $\mathrm{dec} : \mathrm{Im}\, h \to \Gamma$ where $h$ is the output function of system $S$, (rather than a function $\mathrm{dec} : Y \to \Gamma$).

**Theorem 30.** *Let $S$ and $S'$ be $\eta$-finite systems, and $A_S$, $A_{S'}$ their respective canonical automata. If $A_{S'}$ is a realisation of $A_S$, then $S'$ can implement all the functions that $S$ can implement.*

*Proof.* Say we have $A_S = \langle \overline{X}, \overline{U}, \overline{f}, \overline{x_0}, \overline{\mathrm{Im}\, h}, \overline{h} \rangle$ and $A_{S'} = \langle \overline{X'}, \overline{U'}, \overline{f'}, \overline{x_0'}, \overline{\mathrm{Im}\, h'}, \overline{h'} \rangle$.

Say that an assignment of $A_S$ into $A_{S'}$ is given by $\alpha : \overline{X} \to \mathcal{P}_+(\overline{X'})$, $\iota : \overline{U} \to \overline{U'}$ and $\zeta : \overline{\mathrm{Im}\, h'} \to \overline{\mathrm{Im}\, h}$. Let $F_S : (\overline{U})^+ \to \overline{\mathrm{Im}\, h}$ be the canonical regular function for $S$. By Lemma 28, it suffices to show that $A_{S'}$ can implement $F_S$.

Define the encoder $\mathrm{enc} : \overline{U} \to \overline{U'}$ as $\mathrm{enc} = \iota$ and decoder $\mathrm{dec} : \overline{\mathrm{Im}\, h'} \to \overline{\mathrm{Im}\, h}$ as $\mathrm{dec} = \zeta$. Let $D, D'$ be the dynamics of $A_S, A_{S'}$ resp. By Theorem 1.4 in §1.3 of [Hartmanis and Stearns, 1966], we have for all $x' \in \alpha(\overline{x_0})$ and all $w \in (\overline{U})^+$, that

$$h\big(D(\overline{x_0}, w), w_{-1}\big) = \zeta \circ h'\Big(D'\big(x', \iota(w)\big), \iota(w_{-1})\Big).$$

Thus, for all $w \in (\overline{\Sigma})^+$ we have

$$\begin{aligned}
F_S(w) = A_S(w) &= \overline{h}\big(D(x_0, w), w_{-1}\big) \\
&= \zeta \circ h'\Big(D'\big(x_0', \iota(w)\big), \iota(w_{-1})\Big) \\
&= \mathrm{dec} \circ S'\big(\mathrm{enc}(w)\big).
\end{aligned}$$

This concludes the proof. $\square$

*Example* 8. The reverse implication to Theorem 30 does not hold in general. Consider $\Sigma = \Sigma' = \{\sigma\}$, $Q = \{a, b\}$, $Q' = \{a\}$ and unary dynamics functions $\delta : Q \times \Sigma \to Q$ defined as $\delta(q, \sigma) = q$ for every $q \in Q$, and depicted next.

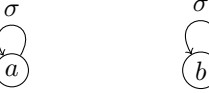

And similarly $\delta' : Q' \times \Sigma \to Q'$ defined as $\delta'(q, \sigma) = q$ for every $q \in Q'$, and depicted next.

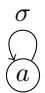

Define system $S = \langle Q, \Sigma, \delta, x_0 = a, \Gamma = Q, \theta \rangle$ with $\theta : (q, \sigma) \mapsto q$, and system $S' = \langle Q', \Sigma, \delta', q_0' = a, \Gamma' = Q', \theta' \rangle$ with $\theta' : (q, \sigma) \mapsto q$.

The only possible state trajectories for either systems are the constant trajectories $x_n = x_0 = a$ and $x_n' = x_0' = a$. Thus, a function $\Sigma^+ \to \Gamma$ can be represented by either system if and only if it is constant. So we have that both systems implement the same functions.

However, there is no assignment $(\alpha, \iota, \zeta)$ from $S$ to $S'$. This is because $\Gamma'$ is a singleton, and so any potential $\zeta : \Gamma' \to \Gamma$ must be constant. At the same time, it must hold that $\alpha(a), \alpha(b)$ are non-empty and

$$\zeta \circ \theta(q', \iota(\sigma)) = \theta(a, \sigma) = a \quad \forall q' \in \alpha(a),$$
$$\zeta \circ \theta(q', \iota(\sigma)) = \theta(b, \sigma) = b \quad \forall q' \in \alpha(b).$$

This is a contradiction, as $\zeta$ must be constant. ∎

**Theorem 31.** *Let $D, D'$ be $\eta$-finite dynamics. Suppose that $D'$ is a realisation of $D$. Then any function implemented by a system with dynamics $D$ can be implemented by some system with dynamics $D'$.*

*Proof.* Let $D_A = \langle \overline{X}, \overline{U}, \overline{f} \rangle$ and $D_{A'} = \langle \overline{X}', \overline{U}', \overline{f}' \rangle$ be the canonical semiautomata of $D$ and $D'$, respectively. Then $D_{A'}$ realises $D_A$.

Let $S$ be a system with dynamics $D$ implementing function $F$. Its canonical automaton $A_S$ has dynamics $D_A$, and so by Lemma 29 there is an automaton $A' = \langle \overline{X}', \overline{U}', \overline{f}', \overline{x}_0', \Gamma', \theta' \rangle$ with dynamics $D_{A'}$ which realises $A_S$.

Consider the system $S' = \langle X', U', f', x_0', X' \times U', \mathrm{id} \rangle$, where $x_0' \in X'$ is s.t. $[x_0']_{\sim_{X'}} = \overline{x}_0'$. Its canonical automaton is $A_{S'} = \langle \overline{X}', \overline{U}', \overline{f}', \overline{x}_0', \overline{X}' \times \overline{U}', \mathrm{id} \rangle$. $A_{S'}$ realises $A'$ with the assignment $\alpha : \overline{X}' \to \mathcal{P}_+(\overline{X}')$ g.b. $\overline{x}' \mapsto \{\overline{x}'\}$, $\iota : \overline{U}' \to \overline{U}'$ g.b. $\overline{u}' \mapsto \overline{u}'$ and finally $\zeta : \overline{X}' \times \overline{U}' \to \Gamma'$ g.b $(\overline{x}', \overline{u}') \mapsto \theta'(\overline{x}', \overline{u}')$. Thus by Theorem 30, $S'$ can implement all functions that $S$ can implement. □

## B.5 Cascade Decomposition and $\eta$-Finite Systems

In this section we bridge the gap between the AAT decomposition results, which apply to serial cascading, and our $\eta$-finite framework, which focuses on feed-forward connections. We begin by showing how taking the canonical semiautomaton 'commutes' with feed-forward cascading.

**Lemma 32.** *Let $D_1 \rightsquigarrow \cdots \rightsquigarrow D_n$ be $\eta$-finite feed-forward cascade dynamics. Then we have*

$$\mathcal{C}\left(D_1 \rightsquigarrow \cdots \rightsquigarrow D_n\right) \equiv \mathcal{C}(D_1) \rightsquigarrow \cdots \rightsquigarrow \mathcal{C}(D_n),$$

*where '$\equiv$' is as per Definition 15.*

*Proof.* By induction, it suffices to show the statement for $n = 2$.

We have $D_1 = \langle X_1, U_1, f_1 \rangle$ and $D_2 = \langle X_2, U_1 \times X_1, f_2 \rangle$. Now, $\mathcal{C}(D_1) = \langle \overline{X}_1, \overline{U}_1, \overline{f}_1 \rangle$ and $\mathcal{C}(D_2) = \langle \overline{X}_2, \overline{U_1 \times X_1}, \overline{f}_2 \rangle$. Note, that here we use that, by Lemma 19, $\overline{(U_1 \times X_1)} = \overline{U}_1 \times \overline{X}_1$.

Thus, we may write the cascade
$$\mathcal{C}(D_1) \rightsquigarrow \mathcal{C}(D_2) = \langle \overline{X}_1 \times \overline{X}_2, \overline{U}_1, \overline{f} \,\rangle$$
where $f$ is the dynamics function of the feed-forward cascade $\mathcal{C}(D_1) \rightsquigarrow \mathcal{C}(D_2)$.

At the same time, writing $D_1 \times D_2 = \langle X_1 \times X_2, U_1, f' \rangle$, we have
$$\mathcal{C}(D_1 \rightsquigarrow D_2) = \langle \overline{X_1 \times X_2}, \overline{U}_1, \overline{f'} \rangle,$$
where again we use Lemma 19 to get $\overline{(X_1 \times X_2)} = \overline{X}_1 \times \overline{X}_2$. It remains to show that $f = \overline{f'}$. For $[x_1]_{\sim_{X_1}} \in \overline{X}_1, [x_2]_{\sim_{X_2}}, [u]_{\sim_{U_1}}$ we have

$$
\begin{aligned}
f\big(\langle [x_1]_{\sim_{X_1}}, [x_2]_{\sim_{X_2}} \rangle, [u]_{\sim_{U_1}} \big) &= \Big\langle \overline{f}_1\big([x_1]_{\sim_{X_1}}, [u]_{\sim_{U_1}} \big), \\
&\qquad \overline{f}_2\big([x_2]_{\sim_{X_2}}, \langle [u]_{\sim_{U_1}}, \overline{f}_1([x_1]_{\sim_{X_1}}, [u]_{\sim_{U_1}}) \rangle \big) \Big\rangle \\
&= \Big\langle \big[f_1(x_1, u_1)\big]_{\sim_{X_1}}, \big[f_2\big(x_2, \langle u, f_1(x_1, u) \rangle \big)\big]_{\sim_{X_2}} \Big\rangle \\
&= \Big[ \langle f_1(x_1, u_1), f_2\big(x_2, \langle u, f_1(x_1, u) \rangle\big) \rangle \Big]_{\sim_{X_1 \times X_2}} \\
&= \Big[ f'(\langle x_1, x_2 \rangle, u) \Big]_{\sim_{X_1 \times X_2}} = \overline{f'}\Big( [\langle x_1, x_2 \rangle]_{\sim_{X_1 \times X_2}}, [u]_{\sim_{U_1}} \Big) \\
&= \overline{f'}\Big( \langle [x_1]_{\sim_{X_1}}, [x_2]_{\sim_{X_2}} \rangle, [u]_{\sim_{U_1}} \Big)
\end{aligned}
$$

This concludes the proof. $\qquad\square$

Note: we treat objects such as $\overline{X}_1 \times \overline{X}_2$ and $\overline{(X_1 \times X_2)}$ as identical, even though one is a product of equivalence classes, and the other is an equivalence class of a product. However, from Lemma 19, we can identify the two in a natural way, that is in a way that is consistent with applying functions component-wise.

Next, we show that cascading interacts well with realisability, up to introducing a connection function.

**Lemma 33.** *Suppose $D_i = \langle X_i, U_i, f_i \rangle, D_i' = \langle X_i', U_i', f_i' \rangle$ are such that $D_i'$ is a realisation for $D_i$, for each $i \in [1..2]$. Then, for any feed-forward cascade $\overset{i}{\rightsquigarrow} D_1 \overset{h}{\rightsquigarrow} D_2$ with input $i$ and connection $h$, there is a continuous function $g : U_1' \times X_1' \to U_2'$ such that $D_1' \overset{g}{\rightsquigarrow} D_2'$ realises $\overset{i}{\rightsquigarrow} D_1 \overset{h}{\rightsquigarrow} D_2$.*

*Proof.* Let $(\alpha_i, \iota_i, \zeta_i)$ be the assignment of $\mathcal{M}(\mathcal{C}(D_i)) = \langle \overline{X}_i, \overline{U}_i, \overline{f}_i, \overline{X}_i \times \overline{U}_i, \mathrm{id} \rangle$ into $\mathcal{M}(\mathcal{C}(D_i')) = \langle \overline{X}_i', \overline{U}_i', \overline{f}_i', \overline{X}_i' \times \overline{U}_i', \mathrm{id} \rangle$, for each $i \in [1..2]$. We assume w.l.o.g. that $h = \mathrm{id}$ and $i = \mathrm{id}$, i.e., we can consider the usual feed-forward cascade $D_1 \rightsquigarrow D_2$, by replacing $D_1$ with $D_{1,i}$ and $D_2$ with $D_{2,h}$. In that case, we have $U_2 = U_1 \times X_1$.

Define $g : \overline{U}_1' \times \overline{X}_1' \to \overline{U}_2'$ given by $g(\overline{u}', \overline{x}') = \iota_2(\overline{u}, \overline{x}) \in U_2'$ where $(\overline{x}, \overline{u}) = \zeta_1(\overline{x}', \overline{u}') \in X_1 \times U_1 = U_2$.

Define

$$
\begin{aligned}
\alpha :&(\overline{X}_1 \times \overline{X}_2) \to \mathcal{P}_+\big(\overline{X}_1' \times \overline{X}_2'\big) &&\text{as } \alpha(\overline{x}_1, \overline{x}_2) = \alpha_1(\overline{x}_1) \times \alpha_2(\overline{x}_2) \\
\iota :&\overline{U}_1 \to \overline{U}_1' &&\text{as } \iota = \iota_1 \\
\zeta :&\overline{X}_1' \times \overline{X}_2' \times \overline{U}_1' \to \overline{X}_1 \times \overline{X}_2 \times \overline{U}_1 &&\text{as } \zeta\big(\langle \overline{x}_1', \overline{x}_2' \rangle, \overline{u}_1' \big) = (a, b, c) \\
& && \text{where } (b, c, a) = \zeta_2(x_2', g\langle \overline{u}_1', \overline{x}_1' \rangle)
\end{aligned}
$$

Let $(\overline{x}_1, \overline{x}_2) \in \overline{X}_1 \times \overline{X}_2$, $\overline{u}_1 \in \overline{U}_1$ and $(\overline{x}_1', \overline{x}_2') \in \alpha\big((\overline{x}_1, \overline{x}_2)\big)$. Let $\overline{f}$ and $\overline{f}_g'$ be the dynamics functions of $\mathcal{C}(D_1) \rightsquigarrow \mathcal{C}(D_2)$ and $\mathcal{C}(D_1') \rightsquigarrow \mathcal{C}(D_2')_g$ respectively. We have that $\overline{f}_g'\big(\langle \overline{x}_1', \overline{x}_2' \rangle, \iota(\overline{u}_1)\big) = \langle \overline{x}_{1,\text{new}}', \overline{x}_{2,\text{new}}' \rangle$, where

$$\overline{x}_{1,\text{new}}' = \overline{f}_1'(\overline{x}_1', \iota_1(\overline{u}_1)) \in \alpha_1\big(\overline{f}_1(\overline{x}_1, \overline{u}_1)\big)$$

by Property I) of assignment, and

$$\overline{x}'_{2,\text{new}} = \overline{f}'_2\big(x'_2, g(\iota(\overline{u}), \overline{x}'_{1,\text{new}})\big)$$

Now, by Property II) of assignment we have $\zeta_1(\overline{x}'_{1,\text{new}}, \iota(u)) = \big(f_1(\overline{x}_1, \overline{u}_1), \overline{u}_1\big)$, since $\overline{x}_{1,\text{new}} \in \alpha_1\big(\overline{f}_1(\overline{x}_1, \overline{u}_1)\big)$. Thus

$$\overline{x}'_{2,\text{new}} = \overline{f}'_2\Big(x'_2, \iota_2\big(\overline{u}_1, \overline{f}_1(\overline{x}_1, \overline{u}_1)\big)\Big) \in \alpha_2\Big(\overline{f}_2\big(\overline{x}_2, \langle \overline{u}_1, \overline{f}_1(\overline{x}_1, \overline{u}_1)\rangle\big)\Big)$$

So, altogether $\langle \overline{x}'_{1,\text{new}}, \overline{x}'_{2,\text{new}}\rangle \in \alpha\Big(\overline{f}\big(\langle \overline{x}_1, \overline{x}_2\rangle, \overline{u}_1\big)\Big)$, so Property I) of assignment is satisfied. Now

$$\zeta\Big(\langle \overline{x}'_{1,\text{new}}, \overline{x}'_{2,\text{new}}\rangle, \iota(\overline{u}_1)\Big) = (a, b, c)$$

where $(b, c, a) = \zeta_2\Big(\overline{x}'_{2,\text{new}}, g\langle \iota(\overline{u}_1), \overline{x}'_{1,\text{new}}\rangle\Big) = \zeta_2\Big(\overline{x}'_{2,\text{new}}, \iota_2\big(\overline{u}_1, \overline{f}_1(\overline{x}_1, \overline{u}_1)\big)\Big)$. Thus

$$\zeta\Big(\langle \overline{x}'_{1,\text{new}}, \overline{x}'_{2,\text{new}}\rangle, \iota(\overline{u}_1)\Big) = \overline{f}(\langle \overline{x}_1, \overline{x}_2\rangle, \overline{u}_1)$$

and so Property II) is satisfied. We may now choose a continuous $g' : U'_1 \times X'_1 \to U'_2$ such that $\overline{g}' = g$ by Lemma 22. Then we have that $\mathcal{C}(D'_{2,g'}) = \mathcal{C}(D'_2)_g$. Overall, the cascade $D'_1 \rightsquigarrow D'_{2,g'}$ realises $D_1 \rightsquigarrow D_2$. $\quad\square$

The decomposition theorems of AAT are stated for serial cascades, while RNNs in practice usually work with feed-forward cascades. In Appendix G.2, we show how $D_1 \ltimes D_2$ can be realised by $D_1 \overset{g_1}{\rightsquigarrow} R_X \overset{g_2}{\rightsquigarrow} D_2$ for some continuous functions $g_1, g_2$, and the *repeat dynamics* $R_X$ over state-space $X$ of $D_1$.

**Definition 24.** The *repeat dynamics on state space* $X$ are the dynamics $R_X = \langle X^2, X, r_X\rangle$, where $r_X\big(\langle x_{\text{old}}, x_{\text{new}}\rangle, x\big) = \langle x_{\text{new}}, x\rangle$. $\quad\blacksquare$

Thus we have that with initial state $\langle a, b\rangle \in X^2$ and input sequence $(u_n)_{n\geq 1} \in X^\omega$, the state sequence is $\big(s_n = \langle x_{n-1}, x_n\rangle\big)_{n\geq 0} \in (X^2)^\omega$ with $x_{-1} = a, x_0 = b$. Note that a repeat dynamics is a Finite Context Dynamics.

For $\eta$-finite spaces, $R_X$ can be decomposed in terms of 2-state repead dynamics.

**Theorem 34.** *Let $X$ be an $\eta$-finite space. Then the repeat dynamics on $X$, $R_X$, are realised by a feed-forward cascade of the repeat dynamics $R_2$ on $\{0, 1\}$.*

*Proof.* Let $X_1, \dots, X_n$ be the $\eta$-components of $X$. We can think of the canonical automaton as the repeat dynamics on $\overline{X}$, $R_{\overline{X}} = \{\overline{X}^2, \overline{X}, r_{\overline{X}}\rangle$.

Consider $C_n = \overset{f_1}{\rightsquigarrow} D_1 \overset{f_2}{\rightsquigarrow} D_2 \dots \overset{f_n}{\rightsquigarrow} D_n = \langle \{0, 1\}^{2\times n}, \overline{X}, f_C\rangle$, with $D_i \equiv R_2$ for all $i \in [1..n]$, and with $f_i : \overline{X} \times \{0, 1\}^{2\times i - 1} \to \{0, 1\}$ given by

$$f_i(\overline{x}_j) = \begin{cases} 0 & \text{if } i \neq j \\ 1 & \text{if } i = j \end{cases}$$

Thus, each $D_i$ works in parallel, treating inputs $\overline{x}_i$ as 1, and others as 0. Then we can retrieve the state of $R_{\overline{X}}$ by checking which $D_i$ has 1 at the old position, and which $D_j$ has 1 at new position. This corresponds to state $\langle \overline{x}_i, \overline{x}_j\rangle$.

The assignment this corresponds to is the following: define $\alpha : \overline{X}^2 \to \mathcal{P}_+(\{0, 1\}^{2n})$ by $\alpha(\langle \overline{x}_i, \overline{x}_j\rangle) = \{E_{i,j}\}$, where $E_{i,j} \in \{0, 1\}^{2\times n}$ is s.t. $[E_{i,j}]_{1,i} = 1, [E_{i,j}]_{2,j} = 1$ and remaining entries are all 0. We also define $\iota : \overline{X} \to \overline{X}$ as the identity, and $\zeta : \{0, 1\}^{2\times n} \times \overline{X} \to \overline{X}^2 \times X$ as mapping $(E_{i,j}, \overline{x}) \mapsto (\langle x_i, x_j\rangle, \overline{x})$, with other inputs mapped arbitrarily.

$\square$

Altogether, we have a recipe for proving positive results. It is sufficient to show that an architecture can realise FLIP-FLOP, to show that it can implement all group-free functions with *serial* cascades. If it further can realize $R_2$, then it can implement all group-free functions with *feed-forward* cascades.

**Theorem 35.** *Suppose that $\eta$-finite dynamics $D$ is a realisation of* FLIP-FLOP*, and $\eta$-finite dynamics $E$ a realization of $R_2$. Then* feed-forward *cascades of $D$ and $E$ components can implement all group free functions.*

*Proof.* Let $F$ be a group-free function. By Theorem 12, $F$ is implemented by a serial cascade of FLIP-FLOP's, say $C$. By the construction in Appendix G.2, we have that $C$ is realised by a feed-forward cascade of FLIP-FLOP's and repeat semiautomata, say $C'$. By Lemma 34, each repeat semiautomaton is a feed-forward cascade of $R_2$ components. Therefore $C'$ is realised by a feed-forward cascade of FLIP-FLOP's components and $R_2$ components, say $C''$. By Lemma 33, a feed-forward cascade of $D$ and $E$ components realises $C''$, say $C'''$. Thus, by transitivity of realisability, $C'''$ realises $C$, and thus by Theorem 31, $C''$ can implement $F$. $\square$

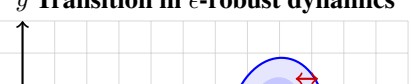

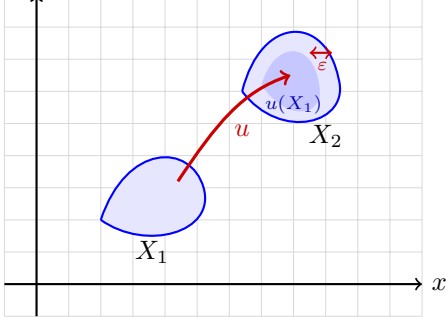

Figure 7: The image of an $\eta$-component under an $\epsilon$-robust transition lies inside the target $\eta$-component, within $\epsilon$-distance of its boundary.

## C  Robust Systems

In this appendix we introduce a central notion of *robustness* that allows us to extend Metric Automata Theory to the study of concrete finite-precision implementations.

Arithmetic operations with floating point numbers are difficult to analyse, since addition and multiplication are not exactly commutative, associative and distributive. Thus, for example, the recurrent form and the convolutional form of the SSM update are not exactly equivalent (also noted by Merrill et al. [2024]—see footnote 3 in Definition 2.1). A theoretical framework which specifies an explicit datatype either is hard to analyse, or introduces additional simplifying assumptions.

The central notion that allows us to extend Metric Automata Theory to the study of finite-precision implementations is the notion of $\epsilon$-*robustness*. Intuitively, it describes stability of the dynamics under transition perturbations.

In Section C.1 we prove Theorem 2, thus showing that robustness provides a way to connect $\eta$-finite systems to their floating-point implementations on real-world computer architectures, without requiring us to commit to any particular standard of floating-point operations.

In Section C.2 we show that robustness provides stability under perturbing the parametrs of a model which describes the dynamics. We will later present a strongly robust dynamics based on the sLSTM model, which uses a particular choice of parameters. Our results show, that in such cases the parameters may be perturbed by some amount and the robust system will retain its behaviour.

Lastly, in Section C.3 we prove Theorem 5 and further describe what kind of connecting functions are required for strongly robust $\eta$-finite cascades, by showing that 2-layer MLPs suffice.

*Robustness marks the departure of Metric Automata Theory from Classical Automata and Formal Languages Theory, allowing us to study phenomena that do not occur with discrete state-spaces.*

For completeness, we restate Definition 2 paying closer attention to the role of inputs in the notion of strong $\epsilon$-robustness.

**Definition 2.** *For $\epsilon > 0$ and $X \subseteq \Omega_X, U \subseteq \Omega_U$, dynamics $D = \langle X, U, f \rangle$ are $\epsilon$-robust (in $\Omega_X$) if, for every $x \in X$ and every $u \in U$, it holds that $\overline{B}_{\Omega_X}(f(x,u), \epsilon) \subseteq X$—i.e., $y \in X$ for all $y \in \Omega_X$ s.t. $\|f(x,u) - y\| \leq \epsilon$. Furthermore, we say that dynamics $D$ are strongly $\epsilon$-robust (in $\Omega_X$ and $\Omega_U$) if they are $\epsilon$-robust (in $\Omega_X$), each $\eta$-component of $X$ contains an $\Omega_X$-ball of radius at least $\epsilon$ and each $\eta$-component of $U$ contains an $\Omega_U$-ball of radius at least $\epsilon$.*

Note that the property of robustness is with respect to the ambient space $\Omega_X$, which contains the state space $X$. Thus, it is possible that a dynamics is $\epsilon$-robust w.r.t. some ambient space (e.g., $\mathbb{R}$), and not $\epsilon$-robust w.r.t. another ambient space (e.g., $\mathbb{C}$). This captures the property, that for a $\eta$-finite dynamics, a function approximating $f$ within $\epsilon$, and taking values in $\Omega$, will implement the same transitions.

**Lemma 36.** *Let $C = D_1 \rightsquigarrow \cdots \rightsquigarrow D_n$ be a cascade, with $D_i = \langle X_i, U \times X_{[1,i-1]}, f_i \rangle$ and $X_i \subseteq \Omega_i, U \subseteq \Omega_U$. Then $C$ is (strongly) $\epsilon$-robust w.r.t. $\Omega_1 \times \cdots \times \Omega_n$ if $D_i$ is (strongly) $\epsilon$-robust w.r.t. $\Omega_i$ for all $i \in 1..n$.*

*Proof.* By induction, it suffices to show the statement for $n = 2$. First, suppose that $D_i$ is $\epsilon$-robust for $i \in 1, 2$. Let $\langle x_1, x_2 \rangle \in X_1 \times X_2$, $u \in U$ and take $\langle y_1, y_2 \rangle \in \Omega_1 \times \Omega_2$ s.t. $||f(\langle x_1, x_2 \rangle, u) - \langle y_1, y_2 \rangle||_2 \leq \epsilon$. We have, by def of cascading

$$f(\langle x_1, x_2 \rangle, u) = \langle x_1', x_2' \rangle \quad \text{where } x_1' = f_1(x_1, u), \ x_2' = f_2\Big(x_2, \langle x_1', u \rangle\Big)$$

By definition of the $L_2$ norm, since $||\langle x_1', x_2' \rangle - \langle y_1, y_2 \rangle|| \leq \epsilon$, we also have

$$||x_1' - y_1|| \leq \epsilon \quad \text{and} \, ||x_2' - y_2|| \leq \epsilon$$

Thus, by $\epsilon$-robustness, we have that $y_i \in X_i$ for $i \in 1, 2$, and hence $\langle y_1, y_2 \rangle \in X_1 \times X_2$. All together, $C$ is $\epsilon$-robust w.r.t. $\Omega_1 \times \Omega_2$.

Suppose further that $D_1, D_2$ are strongly $\epsilon$-robust. Let $Z$ be a $\eta$-component of $X_1 \times X_2$. Then $Z$ is of the form $Z_1 \times Z_2$ for $Z_i$ $\eta$-component of $X_i$, see proof of Lemma 19. We have by strongly-robustness that $\overline{B}_{\Omega_i}(z_i, \epsilon) \subseteq Z_i$ for some $z_i \in Z_i$. By triangle inequality: $\overline{B}_{\Omega_1 \times \Omega_2}\big((z_1, z_2), \epsilon\big) \subseteq \overline{B}_{\Omega_1}(z_1, \epsilon) \times \overline{B}_{\Omega_2}(z_2, \epsilon) \subseteq Z_1 \times Z_2$. Finally, the input space of $D_1 \rightsquigarrow D_2$ is the same as the input space of $D_1$, so by strongly-robustness we have that each $\eta$-component of $U$ contains a closed $\Omega_U$-ball with radius $\epsilon$. $\qquad\square$

## C.1 Finite Datatypes and Proof of Theorem 2

We now consider approximations of dynamical systems using a finite datatype $\mathbb{D} \subseteq \Omega$. $\mathbb{D}$ can for example represent the Python `float` type. We simply consider $\mathbb{D}$ as a discrete subset of $\Omega$, abstracting away the details regarding arithmetic properties of such a datatype.

**Definition 25.** A *finite datatype* is a set $\mathbb{D} \subseteq \Omega = \mathbb{R}^d$ having finite cardinality. A *finite-datatype implementation* of a system $S$ is then a system whose input, state, and output spaces are finite datatypes, and whose dynamics and output functions are implemented using floating-point operations.

**Definition 26.** Call a set $S$ an $\epsilon$-covering of $X \subseteq \Omega$, if for all $x \in X$ there is a $s \in S$ s.t. $||x - s|| \leq \epsilon$.

**Definition 27.** Define $\texttt{int}_p^+ = \{0, \ldots, 2^{p-1} - 1\}$ to be the $p$-bit unsigned integers. Define $\texttt{int}_p = \{2^{p-1}, \ldots, 0, \ldots, 2^{p-1} - 1\}$ to be the $p$-bit signed integers. Define $\mathbb{D}_p$ to be floating point numbers with $2p$-bit significand and $p$-bit exponent:

$$\mathbb{D}_p = \Big\{ \frac{s}{2^{2p-1}} \cdot 2^e : s \in \texttt{int}_{2p}, e \in \texttt{int}_p \Big\}$$

Similarly, define $\mathbb{D}_p'$ to be floating point numbers with $p$ bits of integer precision and $p$ bits of fractional precision:

$$\mathbb{D}_p' = \Big\{ a + \frac{b}{2^p} : a \in \texttt{int}_p, b \in \texttt{int}_P^+ \Big\}$$

**Lemma 37.** *Let $X \subseteq \Omega = \mathbb{R}^d$ be compact. Then, for $p$ sufficiently large, i.e. with sufficient precision, $\mathbb{D}_p'^d$ is an $\epsilon$-covering of $X$.*

*Proof.* $X$ is a compact subspace of $\Omega$, and therefore bounded. So, there is some integer $k \geq 1$ s.t. $X \subseteq [-2^k, 2^k - 1]^d$. There is also some integer $l \geq 1$ s.t. $\epsilon/\sqrt{d} \geq 2^{-l}$. Take $p \geq \max(k, l)$. The set $\mathbb{D}_p'$ is an $2^{-p}$-cover of $[2^{-p}, 2^p - 1]$. Now for any $x \in X \subseteq [-2^p, 2^p - 1]^d$, we have that for each $i \in 1 \ldots d$ there is $y_i \in \mathbb{D}_p'$ s.t. $|[x]_i - y_i| \leq 2^{-p}$. Therefore, writing $y \in [-2^p, 2^p - 1]^d$ for $(y_1, \ldots, y_d)$

$$||x - y|| = \Big( \sum_{i=1}^d \big|[x]_i - [y]_i\big|^2 \Big) \leq \epsilon$$

Therefore $(\mathbb{D}_p')^d$ an $\epsilon$-cover of $X$. $\qquad\square$

**Lemma 38.** *Let $X \subseteq \Omega = \mathbb{R}^d$ be compact. Then, for $p$ sufficiently large, i.e. with sufficient precision, $\mathbb{D}_p^d$ is an $\epsilon$-covering of $X$.*

*Proof.* By the previous Lemma, for some $p$ we have that $\mathbb{D}'_p$ is an $\epsilon$-cover of $X$. We have for each $a \in \mathtt{int}_p, b \in \mathtt{int}_p^+$:

$$a + \frac{b}{2^p} = \frac{2^p \cdot a + b}{2^{2p+1}} \cdot 2^{p+1}$$

Now, $2^p a + b \geq 2^p(-2^p) - 2^p > -2^{2p+1}$ and $2^p a + b \leq 2^p \cdot 2^p + 2^p < 2^{2p+1}$, so that $2^p a + b \in \mathtt{int}_{2p+2}$. Since $p + 1 < 2^{p+1}$, we have $p + 1 \in \mathtt{int}_{p+1}$. So, $\mathbb{D}'_p \subseteq \mathbb{D}_{p+1}$, and therefore $\mathbb{D}_{p+1}$ is also an $\epsilon$-cover for $X$, for sufficiently large $p$. $\qquad\square$

**Definition 28.** Let $X, U$ be $\eta$-finite spaces having components $X_{[1..r]}, U_{[1..s]}$ and subspaces $X' \subseteq X$, $U' \subseteq U$, respectively. Let us consider dynamics

$$D = \langle X, U, f \rangle \quad \text{and} \quad \hat{D} = \langle X', U', \hat{f} \rangle.$$

We say that dynamics $D$ are *simulated by* dynamics $\hat{D}$, with error at most $\epsilon$, if we have that the disjointness condition (C1) holds for every $i \in [1..r]$, the disjointness condition (C2) holds for every $j \in [1..s]$, and the approximation condition (C3) holds.

(C1) $X' \cap X_i \neq \emptyset$, $\qquad$ (C2) $U' \cap U_j \neq \emptyset$, $\qquad$ (C3) $\displaystyle\sup_{x \in X', \, u \in U'} \|f(x, u) - \hat{f}(x, u)\| \leq \epsilon.$

**Lemma 39.** *Suppose $\eta$-finite dynamics $D = \langle X, U, f \rangle$ are $\epsilon$-robust, and are simulated by $\eta$-finite dynamics $\hat{D} = \langle X', U', \hat{f} \rangle$ with error $\epsilon$. Then $\hat{D}$ is a realisation of $D$.*

*Proof.* Consider the canonical semiautomata $D_A = \langle \overline{X}, \overline{U}, \overline{f} \rangle$ and $\hat{D}_A = \langle \overline{X}', \overline{U}', \overline{f}' \rangle$

Define $\alpha : \overline{X} \to \mathcal{P}_+(\overline{X}')$ as

$$\alpha([x]_{\sim_X}) = \left\{ [x']_{\sim_{X'}} \in \overline{X}' \; : \; x' \in [x]_{\sim_X} \right\}$$

which is indeed non-empty by definition of simulation, and well-defined as $X' \subseteq X$, and so if $x'_1 \sim_{X'} x'_2$ then also $x'_1 \sim_X x'_2$. Also define $\iota : \overline{U} \to \overline{U}'$ by

$$\iota([u]_{\sim_U}) = [u']_{\sim'_U} \quad \text{where } u' \in U' \cap [u]_{\sim_U} \text{ is arbitrary,}$$

and $\zeta : (\overline{X}' \times \overline{U}') \to (\overline{X} \times \overline{U})$ by

$$\zeta(\langle [x']_{\sim_{X'}}, [u']_{\sim_{U'}} \rangle) = \langle [x']_{\sim_X}, [u']_{\sim_U} \rangle$$

$\zeta$ is indeed well-defined: suppose $x'_1, x'_2 \in [x']_{\sim_{X'}}$ and $u'_1, u'_2 \in [u']_{\sim_{U'}}$. Then since $X' \subseteq X$ and $U' \subseteq U$ we also have $x'_1, x'_2 \in [x']_{\sim_X}$, since $x'_1 \sim_X x'_2$ and $u'_1, u'_2 \in [u']_{\sim_U}$, since $u'_1 \sim_U u'_2$.

Now, $(\alpha, \iota, \zeta)$ is an assignment of $\mathcal{M}(D_A)$ into $\mathcal{M}(\hat{D}_A)$: for all $[x]_{\sim_X} \in \overline{X}$ and $[u]_{\sim_U} \in \overline{U}$, and for all $[x']_{\sim_{X'}} \in \alpha([x]_{\sim_X})$ we have

$$\overline{f}'([x']_{\sim_{X'}}, \iota([u]_{\sim_U})) = [f'(x', u')]_{\sim_{X'}} \quad \text{where } [u']_{\sim_{U'}} = \iota([u]_{\sim_U})$$

On the other hand, we have

$$\alpha(\overline{f}([x]_{\sim_X}, [u]_{\sim_U})) = \alpha([f(x, u)]_{\sim_X})$$

We have that $x' \in [x]_{\sim_X}$, since $[x'] \in \alpha([x]_{\sim_X})$. We have by simulation with error at most $\epsilon$

$$\|f'(x', u') - f(x', u')\| \leq \epsilon$$

and so $f'(x', u') \in [f(x, u)]_{\sim_X}$, since $D$ is $\epsilon$-robust. Hence $\overline{f}'([x']_{\sim_{X'}}, \iota([u]_{\sim_U})) \in \alpha(\overline{f}([x]_{\sim_X}, [u]_{\sim_U}))$. Thus Part I) of definition of assignment is satisfied.

Moreover, we have

$$\zeta([x']_{\sim_{X'}}, \iota([u]_{\sim_{U'}})) = ([x']_{\sim_X}, [u']_{\sim_U}) = ([x]_{\sim_X}, [u]_{\sim_U})$$

so that Part II) of the definition is satisfied. $\qquad\square$

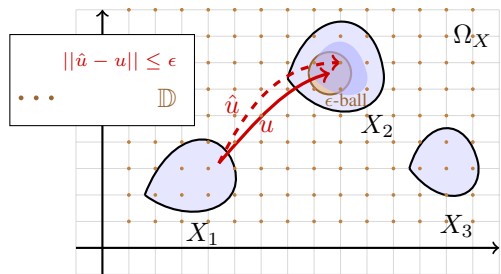

Figure 8: Given sufficient precision, the transitions of strongly $\epsilon$-robust dynamics can be realized with approximate dynamics on a finite datatype, which gives a $\epsilon$-covering for the state-space.

**Lemma 40.** *Consider $\eta$-finite dynamics $D = \langle X, U, f \rangle$, s.t. each component of $X$ and $U$ contains a closed ball of radius $\epsilon$ (in $\Omega_X, \Omega_U$ resp.)*

*Then given datatypes $\mathbb{D}_X \subseteq X, \mathbb{D}_U \subseteq U$ with sufficient precision, there is a function $\hat{f} : \mathbb{D}_X \times \mathbb{D}_U \to \mathbb{D}_X$ s.t. $\langle \mathbb{D}_X, \mathbb{D}_U, \hat{f} \rangle$ simulates $\langle X, U, f \rangle$ with error $\epsilon$.*

*Proof.* Suppose $\mathbb{D}_X$ is an $\epsilon$-covering of $X$, and $\mathbb{D}$ is an $\epsilon$-covering of $U$. Let $X_1, .., X_r, r \geq 1$ be the connected components of $X$. Let $i \in 1..r$, we have by assumption, that for some $x_i \in X_i$

$$\overline{B}(x_i, \epsilon) \subseteq X_i$$

Since $\mathbb{D}_X$ is an $\epsilon$-covering of $X$, there is some $d_i \in \mathbb{D}_X$ s.t. $\|x_i - d_i\| \leq \epsilon$, and therefore $d_i \in \overline{B}(x_i, \epsilon) \subseteq X_i$.

Similarly, there is an element of $\mathbb{D}_U$ in each component of $U$. Now, we may construct $\hat{f}$ as follows: for $x \in \mathbb{D}_X$ and $u \in \mathbb{D}_U$

$$\hat{f}(x, u) = \arg \min_{y \in \mathbb{D}_X} \|y - f(x, u)\|$$

with ties broken arbitrarily. Then, as $\mathbb{D}_X$ is an $\epsilon$-covering of $X$, $\|\hat{f}(x, u) - f(x, u)\| \leq \epsilon$ as desired. $\qquad \square$

We now have the setup, and necessary results for Theorem 2.

**Theorem 2.** *Every $\eta$-finite system with strongly robust dynamics can be implemented with floating-point operations, given sufficient precision.*

*Proof.* Apply Lemma 39 and Lemma 40 to obtain a realisation of $S$ using a finite datatype, e.g. using $\mathbb{D}_p$ or $\mathbb{D}'_p$ for sufficiently large $p$. $\qquad \square$

### C.2  Parametrised Systems

The stability of robust dynamics can also be a desirable property in the context of learning. Consider a parametrised model describing the trained model. If the system described by the model is $\epsilon$-robust and it is sufficiently smooth with respect to its parameters, then perturbing the model parameters will not change the behaviour of the system. Thus a robust system is intuitively more likely to be attained by a learning algorithm.

**Definition 29.** Let $f : \Theta \times \Omega_X \times \Omega_U \to \Omega$ be continuous. Write $f_\theta$ for the function $f(\theta, -, -)$. A dynamics parametrised by $\Theta$ is of the form $D_\theta = \langle X, U, f_\theta \rangle$.

**Theorem 41.** *(Corollary 36.20 of [Willard, 2012]) A continuous functions on a compact metric space $X$ is* uniformly continuous, *that is for all $\epsilon > 0$ there exists $\delta > 0$ such that for all $x, y \in X$ $\|x - y\| \leq \delta \implies \|f(x) - f(y)\| \leq \epsilon$.*

**Theorem 42.** *Let $\eta$-finite dynamics $D_\theta = \langle X, U, f_\theta \rangle$ be parametrised by $\Theta$, and let $\Theta$ be compact. Suppose $D_\theta$ is $\epsilon$-robust (w.r.t $\Omega_X$). Then for some $\delta > 0$, we have that for $\rho \in \Theta$ s.t. $\|\theta - \rho\| \leq \delta$ the dynamics $D_\rho = \langle X, U, f_\rho \rangle$ is well-defined. Moreover, for any system $S_\theta$ with dynamics $D_\theta$, the system $S_\rho$ obtained by switching out $D_\theta$ for $D_\rho$ has the same canonical automaton.*

*Proof.* Since $D_\theta$ is $\eta$-finite, we have that $X$ and $U$ are compact. Thus the Cartesian product $\Theta \times X \times U$ is compact. Thus, by Theorem 41 for all $\epsilon > 0$ we have some $\delta > 0$ such that for all $(\theta, x, u), (\theta, x, u) \in \Theta \times X \times U$

$$||(\theta, x, u) - (\rho, x', u')|| \le \delta \implies ||f(\theta, x, u) - f(\rho, x', u')|| \le \epsilon$$

Now, take $\rho \in \overline{B}_\Theta(\theta, \delta)$. We have for all $x \in X$ and $u \in U$ that

$$||(\theta, x, u) - (\rho, x, u)|| = ||\theta - \rho|| \le \delta$$
$$\therefore \quad ||f(\theta, x, u) - f(\rho, x, u)|| \le \epsilon$$

Thus $f(\rho, x, u) \in \overline{B}(f(\theta, x, u), \epsilon) \subseteq X$, since $D_\theta$ is $\epsilon$-robust. Moreover, letting $X_1, .., X_r$ be the components of $X$ and $U_1, .., U_s$ be the components of $U$, we have that $X \cap X_i \ne \emptyset$ for $i \in 1..r$ and $U \cap U_i$ for $i \in 1..s$. Thus $D_\rho$ simulates $D_\theta$ with error $\epsilon$.

Now, the canonical semiautomaton for $D_\theta$ is $\langle \overline{X}, \overline{U}, \overline{f_\theta} \rangle$ and the canonical semiautomaton for $D_\rho$ is $\langle \overline{X}, \overline{U}, \overline{f_\rho} \rangle$. By Lemma 39, we have that $\overline{f_\theta}$ and $\overline{f_\rho}$ give the same transitions. Therefore the two semiautomata are the exact same. Taking $S_\theta, S_\rho$ as in the statement, we see that they indeed must have the same canonical automaton. $\square$

## C.3  Robust Cascade Decomposition and Proof of Theorem 5

Coming back to connecting functions discussed in Appendix B.5, we have the following refinement of the result.

**Theorem 43.** *Let $D$ be a strongly robust $\eta$-finite dynamics, which are a realisation of* FLIP-FLOP. *Then all group-free functions can be implemented by some strongly robust* serial *cascade of $D$ components. Moreover, the connection functions in such cascade can be given by depth-2 MLPs.*

*Proof.* Say $D = \langle X, U, f \rangle$ is strongly $\epsilon$-robust. By Theorem 35, for any group-free function $F$, there is a serial cascade $C$ of $D$-components which can implement it. By Lemma 36, $C$ is also strongly robust. Say, $C = \overset{g_1}{\leadsto} D_1 \cdots \overset{g_L}{\leadsto} D_L = \langle X^L, U', f_C \rangle$, with $U'$ an $\eta$-finite space, $D_i \equiv D$ and $g_i : U' \times X^{i-1} \to U$.

Let $U_1, \ldots, U_n$ be the $\eta$-components of $U$. By strong robustness, for each $i \in [1..n]$, there is $u_i \in U_i$ s.t. $B_{\Omega_U}(u_i, \epsilon) \subseteq U_i$ By Lemma 22, we can w.l.o.g. assume that $g_i$ has its image in $\{u_1, \ldots, u_n\}$, while still inducing the same mapping $\overline{U}' \times \overline{X}^{i-1} \to \overline{U}$.

By Theorem 16, there is a MLP $N_i : \Omega_{U'} \times \Omega_X^{i-1} \to \Omega_U$ which $\epsilon$-approximates $g_i$, since $U' \times X^{i-1}$ is compact and $g_i$ continuous. For $\langle u', x_1, \ldots, x_{i-1} \rangle \in U' \times X^{i-1}$ we have $f_i(\langle u', x_1, \ldots, x_{i-1} \rangle) = u_j$ for some $j \in [1..n]$, so

$$N(\langle u', x_1, \ldots, x_{i-1} \rangle) \in B_{\Omega_U}(u_j, \epsilon) \subseteq U_j$$

Thus $N_i$ sends elements of $U' \times X^{i-1}$ to the same $\eta$-components of $U$ as $g_i$. Moreover, $N_i$ is continuous.

Overall, the canonical automaton for $\overset{g_1}{\leadsto} D_1 \overset{g_2}{\leadsto} \cdots \overset{g_L}{\leadsto} D_L$ is the same as the canonical automaton for $\overset{N_1}{\leadsto} D_1 \overset{N_2}{\leadsto} \cdots \overset{N_L}{\leadsto} D_L$. Thus the strongly robust cascade with $D$ components and MLP connections can implement $F$. $\square$

Appendix G.3 shows constructions for strongly robust $\eta$-finite xLSTM FLIP-FLOP and $R_2$ dynamics. All together, we obtain Theorem 5:

**Theorem 5** (xLSTM does start-free robustly). *All star-free languages can be recognised by xLSTM cascades, as well as by floating-point implementations of xLSTM cascades given sufficient precision.*

*Proof.* We have that there are strongly robust xLSTM dynamics that realise FLIP-FLOP and $R_2$. Thus by Theorem 43, every group-free function can be implemented by a cascade of strongly robust xLSTM dynamics. Any such cascade is itself strongly robust, by Lemma 36, and thus can be realized by floating-point operations, given sufficient precision, by Theorem 2 $\square$

Moreover, by Theorem 43 we know that for these cascades, it suffices to use MLP connecting functions. By Theorem 42 we also have that the parametrizations of sLSTM blocks which yields FLIP-FLOP and $R_2$ can also be changed, within some $\delta$, retaining the behaviour of the dynamics.

# D  Expressivity Results for State Space Models

In this Appendix we reap rewards of establishing the preliminary framework of Metric Automata Theory for $\eta$-finite dynamics. We can now prove expressivity results by establishing structural properties of dynamics, which are preserved by feed-forward cascades, and which are generally applicable.

In Section D.1 we introduce the notion of *contracting* dynamics, which describes dynamics that are not able to keep track of a state over unbounded input lengths. We use this notion to prove Theorems 3 and 4.

In Section D.2 we introduce another structural property, called *aperiodicity*. It is the $\eta$-finiteness corresponding notion to group-freeness in Finite Automata. We use aperiodicity to prove Theorem 6.

Finally, in Section D.3 we focus on the SSM parametrisation of Mamba, and prove Theorem 7.

## D.1  Contracting Dynamics and Proofs of Theorems 3 and 4

**Definition 30.** Call $\eta$-finite dynamics $\langle X, U, f \rangle$ a *contracting dynamics*, if for any initial points $x_0, x_0' \in X$ and eventually equivalent input sequences $(u_n)_{n \geq 1}, (u_n')_{n \geq 1} \subseteq U$, we have that the corresponding state sequences $(x_n)_{n \geq 1}, (x_n')_{n \geq 1} \subseteq U$ are eventually equivalent.

Thus, a for a contracting dynamics, it does not matter what state the evaluation of the inputs starts from—eventually all initial states lead to the same behaviour under a fixed input sequence. The intuition behind the name is the following—eventually all possible states that the dynamics could be in under the input sequence collapse to a single $\eta$-component.

*Example* 9. Clearly, all Finite Context Dynamics (Definition 18) are contracting. ∎

**Lemma 44.** *Let $C = D_1 \rightsquigarrow \cdots \rightsquigarrow D_n$ be a cascade of $\eta$-finite contracting dynamics. Then $C$ is a contracting dynamics.*

*Proof.* By induction, it is sufficient to show the statement for $n = 2$.

Let us consider $C = D_1 \rightsquigarrow D_2$ with $D_1 = \langle X, U, f_1 \rangle$ and $D_2 = \langle Z, U \times X, f_2 \rangle$. The dynamics function of the cascade is:

$$f(\langle x, z \rangle, u) = \Big\langle f_1(x, u), f_2(z, u') \Big\rangle \quad \text{where } u' = \langle u, f_1(x, u) \rangle$$

Consider arbitrary $\langle x_0, z_0 \rangle, \langle x_0', z_0' \rangle \in X \times Z$ and $(u_t)_{t \geq 1}, (u_t')_{t \geq 1} \in U^\omega$, eventually equivalent in $U$. Take

$$\langle x_n, z_n \rangle = (D_1 \rightsquigarrow D_2)\big(\langle x_0, z_0 \rangle, u_{[1..n]}\big); \quad \langle x_n', z_n' \rangle = (D_1 \rightsquigarrow D_2)\big(\langle x_0', z_0' \rangle, u_{[1..n]}'\big)$$

By inductive hypothesis, $D_1$ is contracting, and so since we have

$$x_n = D(x_0, u_{[1..n]}); \quad x_n' = D(x_0', u_{[1..n]}')$$

we have that $(x_n)_{n \geq 1}, (x_n')_{n \geq 1} \in X^\omega$ are eventually equivalent. Thus also $(\langle u_n, x_{n+1} \rangle)_{n \geq 1}, (\langle u_n', n_{n+1}' \rangle)_{n \geq 1} \in \big(U \times X\big)^\omega$ are eventually equivalent.

Note that we have $z_{n+1} = f_2(z_n, \langle u_n, x_{n+1} \rangle)$ and $z_{n+1}' = f_2(z_n', \langle u_n', x_{n+1}' \rangle)$. Since $D_2$ is by assumption contracting, and the two input sequence are eventually equivalent by continuity of $f_n$, we get that $(z_n)_{n \geq 1}, (z_n')_{n \geq 1} \in Z^\omega$ are eventually equivalent.

So, overall $\big(\langle x_n, z_n \rangle\big), \big(\langle x_n', z_n' \rangle\big) \in \big(X \times Z\big)^\omega$ are eventually equivalent. □

**Lemma 45.** *Suppose a $\eta$-finite Linear Recurrent Dynamics $D$ is $\epsilon$-robust. Then $D$ is contracting.*

*Proof.* Suppose that $D = \langle X, U, f \rangle$ is $\epsilon$-robust.

Let $x_0, x_0' \in X$ and $(u_n)_{n \geq 1}, (u_n')_{n \geq 1} \subseteq U$ which are eventually equivalent—say for $n \geq N$. For each component of $U$, say $U_1, \ldots, U_k$, define a representative element $r_1, \ldots, r_k$. Define $(\tilde{u}_n)_{n \geq 1} \subseteq U$ to be such that $\tilde{u}_n = r_c$ where $U_c$ is the component containing $u_{n+N}$. Thus $(\tilde{u}_n)_{n \geq 1}$ is equivalent to $(u_{n+N})_{n \geq 1}$ and $(u_{n+N}')_{n \geq 1}$.

Write $A_n = A(\tilde{u}_n)$ and $B_n = B(\tilde{u}_n)$ and $f_n(x) = f(x, \tilde{u}_n)$. For $S \subseteq \Omega$, define
$$\Delta S = \{\alpha \cdot (x - y) : \alpha \in [0, 1], x, y \in S\}$$
For $\beta \in \mathbb{R}_{\geq 0}$, write $\beta \cdot S = \{\beta \cdot s : s \in S\}$. Take $M = \sup_{x,y \in X} ||x - y||$. We have that $M$ is finite, since $X$ is compact, and hence bounded. Also, denote $X^{(0)} = X$, $X^{(n+1)} = \{f(x, \tilde{u}_n) : x \in X^{(n)}\} = \{D(x)\}$.

We have, by induction that $\Delta(X^{(n)}) \subseteq \left(\frac{M}{M+2n\epsilon}\right) \cdot \Delta(X)$: for $n = 0$ this is immediate.

For $n \geq 1$, by inductive hypothesis we have $\Delta(X^{(n-1)}) \subseteq \left(\frac{M}{M+2(n-1)\epsilon}\right) \cdot \Delta(X)$. Consider $u \neq 0$, $u \in \Delta(X^{(n)})$. Take $v = \frac{u}{||u||}$. We have that
$$u = \beta \cdot (f_n(x) - f_n(y)) = \beta \cdot A_n(x - y)$$
for some $x, y \in X^{(n-1)}$ and $\beta \in [0, 1]$. We have that $x - y \in \Delta(X^{(n-1)}) \subseteq \left(\frac{M}{M+2(n-1)\epsilon}\right) \cdot \Delta(X)$, so for some $x', y' \in X$ we have
$$x' - y' = \left(\frac{M}{M + 2(n-1)\epsilon}\right)^{-1} \cdot (x - y)$$

Now:
$$\left|\left|f_n(x') - (f_n(x') + \epsilon \cdot v)\right|\right| = \epsilon \quad \text{and} \quad \left|\left|f_n(y') - (f_n(y') + \epsilon \cdot v)\right|\right| = \epsilon$$
so by robustness, $f_n(x') + \epsilon \cdot v \in X$ and $f_n(y') - \epsilon \cdot v$. Thus
$$\begin{aligned}
\Delta X \ni (f_n(x') + \epsilon \cdot v) &- (f_n(y') - \epsilon \cdot v) \\
&= f_n(x') - f_n(y') + 2\epsilon \cdot v \\
&= A_n(x' - y') + 2\epsilon \cdot v \\
&= \left(\frac{M}{M+2(n-1)\epsilon}\right)^{-1} \cdot A_n(x - y) + 2\epsilon \cdot v \\
&= \left(\left(\frac{M}{M+2(n-1)\epsilon}\right)^{-1} \cdot \beta^{-1} + \frac{2\epsilon}{||u||}\right) \cdot u
\end{aligned}$$

So, we have $u = \gamma \cdot l$ for some $l \in \Delta X$ and
$$\begin{aligned}
\gamma^{-1} &= \left(\frac{M}{M+2(n-1)\epsilon}\right)^{-1} \cdot \beta^{-1} + \frac{2\epsilon}{||u||} \\
&= \frac{M + 2(n-1)\epsilon}{M} \cdot \beta^{-1} + \frac{2\epsilon}{||u||} \\
\text{as } \beta^{-1} \geq 1 \text{ and } ||u|| \leq M \quad &\geq \frac{M + 2(n-1)\epsilon}{M} + \frac{2\epsilon}{M} \\
&= \frac{M + 2n\epsilon}{M}
\end{aligned}$$

So $u \in \left(\frac{M+2n\epsilon}{M}\right) \cdot \Delta(X)$, and thus indeed $\Delta(X^{(n)}) \subseteq \left(\frac{M}{M+2n\epsilon}\right) \cdot \Delta(X)$. Therefore $\sup_{x,x' \in X^{(n)}} ||x - x'|| \to 0$ as $n \to \infty$.

Now, consider the state-sequences $\left(D(x_0, u_{[1..n]})\right)_{n \geq 1}$, $\left(D(x_0', u_{[1..n]}')\right)_{n \geq 1}$. We have by Lemma 24
$$\begin{aligned}
D(x_0, u_{[1..(n+N)]}) &= D(x_N, u_{[(N+1)..(n+N)]}) \\
&\sim_X D(x_N, \tilde{u}_{[1..n]})
\end{aligned}$$
and similarly $D(x_0', u_{[(1+N)..(n+N)]}') \sim_X D(x_N', \tilde{u}_{[1..n]})$. Now, $D(x_N', \tilde{u}_{[1..n]}), D(x_N, \tilde{u}_{[1..n]}) \in X^{(n)}$. Thus we have
$$\left|\left|D(x_N', \tilde{u}_{[1..n]}) - D(x_N, \tilde{u}_{[1..n]}) \in X^{(n)}\right|\right| \to 0 \text{ as } n \to \infty$$

Therefore, eventually $D(x_N', \tilde{u}_{[1..n]})$ and $D(x_N, \tilde{u}_{[1..n]}) \in X^{(n)}$ are in the same $\eta$-component of $X$. $\qquad \square$

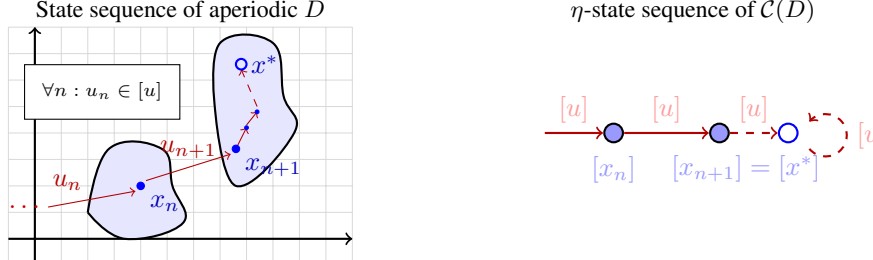

State sequence of aperiodic $D$ $\qquad$ $\eta$-state sequence of $\mathcal{C}(D)$

Figure 9: State sequence of aperiodic dynamics under iterated input always $\eta$-converges.

**Theorem 3** (Non-robustness of LRDs). *Suppose an $\eta$-finite LRD $D$ is such that its canonical semiautomaton $D_A$ has at least two states, and an input inducing an identity transformation. Then $D$ cannot be $\epsilon$-robust for any $\epsilon > 0$.*

*Proof.* Let $D = \langle X, U, f \rangle$ be an $\eta$-finite LRD, such that its canonical semiautomaton $D = \langle \overline{X}, \overline{U}, \overline{f} \rangle$ has at least two distinct $\eta$-states, say $\overline{x}, \overline{x}'$, and an input $\overline{u}$ inducing identity transformation on $\overline{X}$.

For contradiction suppose that $D$ is robust. Then by Lemma 45, $D$ is contracting. Thus for $x_0 \in \overline{x}, x_0' \in \overline{x}'$ and $u \in \overline{u}$ we have that the sequences $\left(x_n = D(x_0, u^n)\right)_{n \geq 1}, \left(x_n' = D(x_0', u^n)\right)_{n \geq 1} \in X^\omega$ are eventually equivalent. Since $[u]_{\sim_U} = \overline{u}$ induces the identity transformation of $\overline{X}$, we have that the corresponding sequences $([x_n]_{\sim_X})_{n \geq 1}, ([x_n']_{\sim_X})_{n \geq 1} \in \overline{X}^\omega$ are constant, equal $\overline{x}, \overline{x}'$ respectively. Thus necessarily $\overline{x} = \overline{x}'$. This is a contradiction. $\qquad \square$

**Lemma 46.** *Contracting dynamics cannot implement the state-sequence function of* FLIP-FLOP.

*Proof.* Consider a system $S$ with some encoder $\mathrm{enc} : \{\mathtt{set}, \mathtt{reset}, \mathtt{id}\} \to U$ and decoder $\mathrm{dec} : Y \to \{\mathtt{low}, \mathtt{high}\}$. Suppose that the dynamics $D = \langle X, U, f \rangle$ of $S$ are contracting. Consider $x_0 \in X$ and input sequences $(u_n)_{n \geq 1}, (u_n')_{n \geq 1} \subseteq U$, given by

$$u_1 = h; \quad u_1' = l; \quad u_n = u_n' = i \quad \text{for } n > 1$$

They are eventually equivalent, and so the corresponding state sequences $x_n = D(x_0, \langle u_{1..n} \rangle)$ and $x_n' = D(x_0, \langle u_{1..n}' \rangle)$ are also eventually equivalent. Thus

$$\mathrm{dec} \circ S\big( \mathrm{enc}(u_{1...n}) \big) = \mathrm{dec} \circ S\big( \mathrm{enc}(u_{1...n}') \big) \in \{\mathtt{high}, \mathtt{low}\}$$

for large enough $n$, since $\{\mathtt{high}, \mathtt{low}\}$ is a discrete space.

However, the two sequences of inputs correspond to different flip flop states - thus $D$ cannot be a dynamics for a system that implements a flip flop. $\qquad \square$

**Theorem 4** (LRDs cannot do FLIP-FLOP robustly). FLIP-FLOP *cannot be implemented by a cascade of $\eta$-finite $\epsilon$-robust LRDs for any $\epsilon > 0$.*

*Proof.* A cascade of such LRDs is contracting by Lemmas 45 and 44. Thus, by Lemma 46, it cannot implement FLIP-FLOP. $\qquad \square$

### D.2 Aperiodic Dynamics and Proof of Theorem 6

**Definition 31.** For a $\eta$-finite space $X$, we say a sequence $(x_n)_{n \geq 1} \subseteq X$ $\eta$-converges in $X$, if eventually all its terms lie in the same $\eta$-component of $X$.

If the sequence of states of a system $\eta$-converges, it means that the behaviour of that system is eventually the same.

**Definition 32.** Call a $\eta$-finite dynamics $D = \langle X, U, f \rangle$ *aperiodic*, if for all $x_0 \in X$ and input sequences $(u_n)_{n \geq 1} \subseteq U$ $\eta$-convergent in $U$, we have that the corresponding state sequence $(x_n)_{n \geq 1} \subseteq X$ is $\eta$-convergent in $X$.

An example of a aperiodic dynamics is given by the FLIP-FLOP dynamics. An input sequence that $\eta$-converges must eventually be constantly `set` or `reset`. In that case, the state is eventually `high`, `low` respectively.

**Lemma 47.** *Let $D$ be a $\eta$-finite Linear Recurrent Dynamics, with $A(u)$ having all its eigenvalues being non-negative. Then $D$ is aperiodic.*

*Proof.* This is a similar argument as for Theorem 1 in Grazzi et al. [2025], with some simplifications stemming from the fact that we can use associativity of linear operations freely.

Let $D = \langle X, U, f \rangle$ be an $\eta$-finite Linear Recurrent Dynamics, with $X \subseteq \mathbb{R}^d$, s.t. $A(u)$ has all its eigenvalues being real, for all $u \in U$. Say $f(x, u) = A(u) \cdot x + B(u)$.

Consider a sequence $(u_n)_{n \geq 1} \in U^\omega$, $\eta$-convergent in $U$, and $x_0 \in X$. Let $\left( x_n = D(x_0, u_{1\ldots n}) \right)_{n \geq 1} \in X^\omega$ be the corresponding state sequence. We have some $N$ s.t. for $n \geq N$ all $u_n$ are contained in the same component of $U$, we may pick a representative $r \in U$ of that component.

Write $A = A(r), B = B(r)$. By Lemma 25, we have for $n \geq N$ that

$$x_{n+N} \sim_X x'_n = D(x_N, r^{n-N})$$

We consider the state sequence in the diagonalized space of $A$. Write $A = P^{-1}JP$ for the Jordan normal form of $A$. Here $J$ is block diagonal, with say blocks $J_1, \ldots, J_s$, $J_b \in \mathbb{R}^{m_b \times m_b}$ being a Jordan Block with $\lambda_b$ on the diagonal being an eigenvalue of $A$, and $1$ on the right off-diagonal. Also $P \in \mathbb{R}^{d \times d}$, since all eigenvalues of $A$ are real.

Take $\bar{x}_n = Px'_n$, then we have

$$\begin{aligned}
\bar{x}_{n+1} &= P(Ax'_n + B) = P(AP^{-1}\bar{x}_n + B) \\
&= P(P^{-1}JPP^{-1}\bar{x}_n + B) = J\bar{x}_n + PB
\end{aligned}$$

We will consider the difference $z_n = \bar{x}_{n+1} - \bar{x}_n$. Unrolling the recurrence we get

$$z_n = J^n(J\bar{x}_0 - \bar{x}_0) = J^n z_0$$

The $i$-th entry of this difference, where $i$ is in say the $b$-th block of $J$, is

$$[z_n]_i = \sum_{j=i}^{m_b} \lambda_b^{n+i-j} \binom{n}{j-1} [z_0]_j$$

This is of the form considered in Lemma 64. Thus, $[z_n]_i \in \mathbb{R}$ is eventually monotone, and so it either converges in $\mathbb{R}$ or is unbounded as $n \to \infty$.

Now, if $z_n \to 0$, that is $[z_n]_i$ for all $i \in [1..d]$, then we have that also, by continuity of linear maps, $x'_{n+1} - x'_n = P^{-1}z_n \to 0$, so that $x'_n$ must eventually be in the same component of $X$ by Lemma 20. Therefore also $(x_n)_{n \geq 1}$ is $\eta$-convergent in $X$.

Otherwise, one of the entries of $z_n$ either is unbounded, or converges to a non-zero limit. In both cases, the corresponding entry of $x_n$ is unbounded as $n \to \infty$, and so this is impossible in a $\eta$-finite space $X$.

Overall, this shows that $D$ must be aperiodic. $\qquad\square$

**Lemma 48.** *Let $D = \langle X, U, f \rangle$ be a $\eta$-finite Finite Context Dynamics. Then $D$ is aperiodic.*

*Proof.* Let $l$ be the context length of $D$. Let $x_0 \in X$ and $(u_n)_{n \geq 1} \in U^\omega$ be $\eta$-convergent in $U$. Let $\bar{u} \in U$ lie in the component of $U$ which contains the tail of $(u_n)_{n \geq 1}$, say for $n \geq N$. For $n \geq N + l$ we have that $u_{n-l+1}, \ldots, u_n \sim \bar{u}$, and so

$$x_n = C(\langle u_{n-l-1\ldots n} \rangle) \sim C(\bar{u}^l)$$

Thus $x_n$ is in the component of $X$ containing $C(\bar{u}^l)$. $\qquad\square$

**Lemma 49.** *Let $C = D_1 \rightsquigarrow \cdots \rightsquigarrow D_k$ be a cascade of $\eta$-finite aperiodic dynamics $D_1, \ldots, D_k$. Then $C$ is aperiodic.*

*Proof.* By induction, is is sufficient to show the statement for $n = 2$.

Let us consider $C = D_1 \rightsquigarrow D_2$ with $D_1 = \langle X, U, f_1 \rangle$ and $D_2 = \langle Z, U \times X, f_2 \rangle$. The dynamics function of the cascade is:

$$f(\langle x, z \rangle, u) = \Big\langle f_1(x, u), f_2(z, u') \Big\rangle \quad \text{where} \ \ u' = \langle u, f_1(x, u) \rangle$$

Consider a sequence $(u_t)_{t \geq 1} \in U^\omega$ $\eta$-convergent in $U$, and $\langle x_0', x_0 \rangle \in X' \times X$.

As $D_1$ is aperiodic, the corresponding sequence $(x_n)_{n \geq 1} \subseteq X^\omega$ is $\eta$-convergent in $X$. Equivalently, $(x_{t+1})_{t \geq 1}$ is $\eta$-convergent in $X$. Moreover, then the sequence $(u_n' = \langle u_n, x_{n+1} \rangle)_{n \geq 1}$ is $\eta$-convergent in $U \times X$. Since $D_2$ is aperiodic, the sequence $(z_n)_{n \geq 1} \in Z$ is therefore $\eta$-convergent in $Z$.

All together, $(\langle x_n, z_n \rangle)_{n \geq 1}$ is $\eta$-convergent in $X \times Z$. $\Diamond$ $\hfill\square$

**Theorem 50.** *$\eta$-finite dynamics are aperiodic if and only if their canonical semiautomaton is group-free*

*Proof.* Let $D = \langle X, U, f \rangle$ have canonical semiautomaton $D_A = \langle \overline{X}, \overline{U}, \overline{f} \rangle$

($\Rightarrow$) First, suppose that $D_A$ is not group-free. By Theorem 13, there exist some $S \subseteq \overline{X}$ and $\overline{u} \in \overline{U}$ s.t. $\overline{f}(-, \overline{u})$ induces a non-trivial permutation on $S$. That is, since $S$ is a finite set, we have $s \in \overline{X}$ s.t. $D_A(s, \overline{u}^n) \neq D_A(s, \overline{u}^{n+1})$ for all $n \geq 1$. Here $\overline{u}^n$ denotes the word of length $n$ consisting of repeated symbol $\overline{u}$.

Take $u \in U$ s.t. $[u]_{\sim_U} = \overline{u}$ and $x \in X$ s.t. $[x]_{\sim_X} = s$. Then, we have that for all $n \geq 1$ that

$$\big[D(x, u^n)\big]_{\sim_X} \neq \big[D(x, u^{n+1})\big]_{\sim_X}$$

The input sequence $(u^n)_{n \geq 1}$ is $\eta$-convergent in $U$, but the corresponding state sequence $(D(s, u^n))_{n \geq 1}$ is not. Thus, $D$ is not aperiodic.

($\Leftarrow$) Now, suppose that $D_A$ is group free. By Theorem 12, $D_A$ can be realized by a *serial* cascade of FLIP-FLOPs, say $C$. We also have, that $C$ can be realized by a *feed-forward* cascade $C'$ of FLIP-FLOPs and repeat semiautomata, all of which are aperiodic (as repeat semiautomata are FCDs). Thus by Lemma 49, $C'$ is aperiodic. It remains to show that dynamics realised by aperiodic dynamics are also aperiodic.

Let $(\alpha, \iota, \zeta)$ be an assignment of $D_A$ into $C'$. Consider an $\eta$-convergent input sequence $(\overline{u}_n)_{n \geq 1} \subseteq \overline{U}$ and $\overline{x}_0 \in \overline{X}$, with the corresponding state sequence $(\overline{x}_n = D_A(\overline{x}_0, \overline{u}_{[1..n]}))_{n \geq 0} \subseteq \overline{X}$. Since $(\overline{u}_n) \subseteq \overline{U}$ is $\eta$-convergent, it is in fact eventually constant, since $\overline{U}$ is a discrete space.

Since $C'$ realizes $D_A$, by Theorem 14, we have, for $\overline{x}_0' \in \alpha(\overline{x}_0)$

$$\mathcal{M}(D_A)\big(\overline{x}_0, \overline{u}_{[1..n]}\big) = \zeta \circ \mathcal{M}(C)(\overline{x}_0', \iota(\overline{u}_{[1..n]}))$$

where $\mathcal{M}(D_A), \mathcal{M}(C)$ are the canonical machines for $D_A, C$, respectively. Now, $(\overline{u}_n)$ is eventually constant and so also $(\iota(\overline{u}_n))$ is eventually constant. $C$ is aperiodic, and so the sequence $C(\overline{x}_0', \iota(\overline{u}_{[1..n]}))$ is $\eta$-convergent (and thus eventually constant, as $C$ is a semiautomaton). All together

$$\mathcal{M}(D_A)\big(\overline{x}_0, \overline{u}_{[1..n]}\big) = \zeta \circ \mathcal{M}(C)(\overline{x}_0', \iota(\overline{u}_{[1..n]})) = \zeta\Big(C(\overline{x}_0', \iota(\overline{u}_{[1..n]})), \iota(\overline{u}_n)\Big)$$

by def. of canonical machines, and therefore this sequence is also eventually constant.

Thus the state sequence $D_A\big(\overline{x}_0, \overline{u}_{[1..n]}\big)$ itself is eventually constant.

Equivalently, by Lemma 26, for any $s\eta$-convergent sequence $(u_n) \subseteq U$ and $x_0 \in X$ the state sequence $(D(x_0, u_{[1..n]})) \subseteq X$ is $\eta$-convergent, and so $D$ is indeed aperiodic. $\hfill\square$

### D.3 Parametrisation of Mamba and Proof of Theorem 7

Sarrof et al. [2024] show that any star-free language can be recognized by an SSM like Mamba (Gu et al. [2022]), using the Krohn and Rhodes Theorem from Algebraic Automata Theory. However, in their construction, they assume that gates of the form $A(u) = \mathbf{0}$ can be used, which is not the case for architectures utilizing strictly positive parametrization, like Mamba.

We show in Construction 3 a modified $\eta$-finite system construction, which only requires gates with diagonal entries in the range $[\epsilon, 1]$, for a suitable $\epsilon > 0$. As it turns out, further restricting diagonal entries to lie in $(-1, 1)$ makes it impossible to implement a flip flop.

Mamba ([Gu et al., 2022]) parametrization is of the form

$$A(u) = \text{Diag}\big( \exp\left(-\Delta_u \odot \exp(z_u)\right) \big) \quad \text{where} \quad z_u \in \mathbb{R}^d, \Delta_u \in (0, \infty)^d$$

and $\odot$ is the element-wise product $\mathbb{R}^d \times \mathbb{R}^d \to \mathbb{R}^d$. This gives $-\big[\Delta_u \odot \exp(z_u)\big]_i < 0$ for $i \in 1 \ldots d$, and thus $A(u)_i \in (0, 1)$ for $i \in 1 \ldots d$. We will show in this section that an SSM using Mamba blocks cannot implement a flip flop for unbounded .

However, experimental results in [Sarrof et al., 2024] show that this architecture does well in experimental evaluations and demonstrates length generalization for star-free modelling tasks. For tasks involving periodic modelling, the model fails to length generalize. This motivates us to investigate the geometric complexity of the state space when evaluated on sequences of bounded length in Appendix E.

**Construction 3.** There is a $\eta$-finite system with Linear Recurrent Dynamics with diagonal entries in $[\epsilon, 1]$, for some $\epsilon > 0$, which realize FLIP-FLOP dynamics.

Take $\epsilon = 1/4$. Consider $X = X_l \cup X_h \subseteq \mathbb{R}$, where

$$X_l = \bar{B}(1, \epsilon); \quad X_h = \bar{B}(2, \epsilon)$$

Then $X_{q_0}, X_l, X_h$ are the components of $X$, and $X$ is $\eta$-finite. Take $U$, $e : \{s, r, i\} \to U$ and $f : X \times U \to X$ to be such that

$$f(x, e(\sigma)) = A_\sigma \cdot x + B_\sigma \quad \text{where} \quad (A_\sigma, B_\sigma) = \begin{cases} (1, 0) & \text{if } \sigma = i \\ (\epsilon/4, 1) & \text{if } \sigma = r \\ (\epsilon/4, 2) & \text{if } \sigma = s \end{cases}$$

We have $X \subseteq \bar{B}(0, 2 + \epsilon)$, and so $(\epsilon/4 \cdot -)(X) \subseteq \bar{B}(0, \epsilon/4 \cdot (2 + \epsilon)) \subseteq \bar{B}(0, \epsilon)$. Thus we see that $f$ maps $X$ to $X_l$ under input $r$ and to $X_h$ under input $s$. Under input $i$, $f$ acts as identity. Thus these dynamics indeed realize FLIP-FLOP, through assignment that identifies with $\alpha$ mapping $\texttt{high} \mapsto X_h, \texttt{low} \mapsto X_l$, $\iota$ mapping $\texttt{set} \mapsto s, \texttt{reset} \mapsto r, \texttt{id} \mapsto i$ and $\zeta$ mapping $X_l$ to $\texttt{low}$ and $X_h$ to $\texttt{high}$.

**Lemma 51.** *Let $D = \langle X, U, f \rangle$ be an $\eta$-finite Linear Recurrent dynamics with $A(u)$ diagonal, with entries in $(-1, 1)$ for all $u \in U$. Then $D$ is contracting.*

*Proof.* Let $x_0, x_0' \in X$ and $(u_n)_{n \geq 1} \subseteq U$. For each component of $U$, say $U_1, \ldots, U_k$, define a representative element $r_1, \ldots, r_k$. Define $(u_n')_{n \geq 1} \subseteq U$ to be such that $u_n' = r_c$ where $U_c$ is the component containing $u_n$. Thus $(u_n')_{n \geq 1}$ is equivalent to $(u_n)_{n \geq 1}$, and $(u_n')_{n \geq 1}$ takes finitely many values $r_1, \ldots, r_k$.

Now, consider $A_1, \ldots, A_k$, where $A_c = A(r_c)$. For each $c \in [1..r]$, let $\lambda_c$ be the largest size eigenvalue of $A_c$. Then we have $|\lambda_c| < 1$, and

$$\|A_c \cdot x\|_2 \leq |\lambda_c| \cdot \|x\|_2 \quad \forall x \in X$$

Let $\lambda \in \arg\max_{c \in 1..r} |\lambda_c|$, then we have $|\lambda| < 1$ and

$$\|A(r_c) \cdot x\|_2 \leq |\lambda| \cdot \|x\|_2 \quad \forall x \in X, c \in 1..k$$

Now, we have that for the state sequences $(x_n)_{n \geq 1}, (x'_n)_{n \geq 1}$ corresponding to initial states $x_0, x'_0$ resp., and the input sequence $(u'_n)_{n \geq 1}$, the following holds:

$$
\begin{aligned}
\|x_n - x'_n\|_2 &= \left\| \big(A(u'_n) \cdot x_{n-1} + B(u'_n)\big) - \big(A(u'_n) \cdot x'_{n-1} + B(u'_n)\big) \right\|_2 \\
&= \left\| A(r_c) \cdot (x_{n-1} - x'_{n-1}) \right\|_2 \quad \text{for some } c \in [1..k] \\
&\leq |\lambda| \cdot \left\| x_{n-1} - x'_{n-1} \right\|_2 \\
&\leq \ldots \\
&\leq |\lambda|^n \cdot \left\| x_0 - x'_0 \right\|_2 \to 0 \quad \text{as} \quad n \to \infty
\end{aligned}
$$

Thus eventually $x_n$ and $x'_n$ must be in the same component of $X$. $\qquad \square$

Altogether, we arrive at the following result (for $\eta$-finite dynamics), restated here more precisely than in the main body.

**Theorem 7.** *SSMs with Mamba parametrisation cannot recognise* FLIP-FLOP *as $\eta$-finite systems.*

*Proof.* Mamba blocks are feed-forward cascades of LRDs of the type considered in Lemma 51 and convolution blocks (FCDs)—see Figure 6. Thus $\eta$-finite feed-forward cascades of Mamba blocks are contracting, and so by Lemma 46, cannot implement FLIP-FLOP. $\qquad \square$

# E   Geometrically Constrained Systems

In this appendix, we depart the setting of $\eta$-finiteness, and explore *geometrically-constrained systems (GCSs)*. This setting allows for systems implementing functions beyond regular, but shares many properties with the $\eta$-finite setting. We develop the theory of GCS to explain empirical capabilities of Mamba, and to showcase the flexibility and generalizability of Metric Automata Theory.

In Section E.1 we develop a notion analogous to *aperiodicity* from Section D.2. We then prove Theorem 9.

In Section E.2 we introduce a generalisation of $\eta$-finiteness, called *weak $\eta$-finiteness*. We use it to argue that the cascade decomposition results for $\eta$-finite dynamics still apply to dynamics with convex-covering state-spaces.

In Section E.3 we show that $\eta$-finite dynamics are a special case of convex-constrained dynamics. Finally, we show a construction of a FLIP-FLOP using a Mamba convex-constrained SSM, and argue using weakly $\eta$-finiteness that Theorem 8 holds.

**Definition 33.** For $\Omega = \mathbb{R}^d$ or $\mathbb{C}^d$, we call $C \subseteq \Omega$ a *convex-covering* if $C$ is a finite union of open, convex sets in $\Omega$. We say that $X \subseteq \Omega$ is *convex-covered* by $C$ if $X \subseteq C$.

We say $X$ is *convex-separated* by $C$ if (i) it is convex-covered by $C$ and (ii) each path-connected component of $C$ contains at most one path-connected component of $X$. ∎

Note: any convex set in $\Omega = \mathbb{R}^d$ or $\mathbb{C}^d$ is path-connected. Thus any convex-covering $C$ has finitely many path-connected components.

**Definition 34.** Let $\Omega = \mathbb{R}^d$ or $\Omega = \mathbb{C}^d$, and let $C \subseteq \Omega$. We say that dynamics $D = \langle X, U, f \rangle$ are *convex-covered* by $C$ if $X$ is convex-covered by $C$. We define a *system geometrically-constrained by $C$* as a tuple $S_C = \langle X, U, f, C, x_0, Y, h \rangle$, where its dynamics $\langle X, U, f \rangle$ is a dynamics convex-covered by $C$, $x_0 \in X$ is the initial state, and $h : C \times U \to Y$ is the continuous output function. ∎

The difference between a shortcut system and a system is that the dynamics function is defined only on $X$, while the output function is define on the convex-covering $C$.

We extend the definition of implementing a function to shortcut systems: $S_C$ implements $F : \Sigma^+ \to \Gamma$ with encoder $\text{enc} : \Sigma \to U$ and decoder $\text{dec} : \text{Im } h \to \Gamma$ if $\text{enc}, \text{dec}$ are continuous and $F(w) = \text{dec} \circ S\big(\text{enc}(w)\big)$.

**Construction 4.** Consider Linear Recurrent Dynamics with state-space $X = \mathbb{Z}$, input space $U = \{a, b\}$ and dynamics function $f(n, a) = n + 1; \quad f(n, b) = n - 1$. The space $C = (-\infty, -0.5) \cup (-0.5, 0.5) \cup (0.5, \infty)$ is a convex-covering for this dynamics. We may define the output function $h : C \to \{0, 1\}$ to map points in $(-\infty, -0.5) \cup (0.5, \infty)$ to 0 and points in $(-0.5, 0.5)$ to 1. Picking initial state $x_0 = 0$ , we have that this GCS outputs 0 precisely when the input has the same number of $a$s and $b$s. This recognizes the language

$$\{w \in \{a, b\}^+ : w \text{ has as many } a\text{s as } b\text{s.}\},$$

whose dynamics can be interpreted as a counter, with $a$ corresponding to $+1$ and $b$ corresponding to $-1$.

**Lemma 52.** *For a cascade $\mathcal{D} = D_1 \rightsquigarrow \cdots \rightsquigarrow D_n$ with $D_i$ convex-covered/convex-separated by $C_i$ we have that $C$ is convex-covered/convex-separated by $\mathcal{C} = C_1 \times \cdots \times C_n$*

*Proof.* Suppose $D_i$ is convex-covered by $C_i$ for $i \in [1..n]$. First, $C_1 \times \cdots \times C_n$ is indeed a convex-covering. A product of convex sets is convex, and so a product of finite unions of convex sets is also a finite union of convex sets (by commutativity of set product and union, see proof of Lemma 19). Thus, $X_1 \times \cdots \times X_n \subseteq \mathcal{C}$ and $\mathcal{D}$ is convex-covered by $\mathcal{C}$.

Now, suppose further that $D_i$ is convex-separated by $C_i$ for $i \in [1..n]$. The path-connected components of $\mathcal{C}$ are of the form $\prod_{i=1}^{n} G_i$, where $G_i$ is a path-connected component of $C_i$. Similarly, path-connected components of $X = X_1 \times \cdots \times X_n$ are of the form $\prod_{i=1}^{n} Z_i$ where $Z_i$ is a path-connected component of $X_i$.

We have that $\prod_{i=1}^{n} Z_i$ intersects $\prod_{i=1}^{n} G_i$ precisely when $Z_i$ intersects $G_i$ for each $i \in [1..n]$. Hence, there is exactly one component of $\mathcal{C}$ intersecting $\prod_{i=1}^{n} Z_i$, i.e., $\mathcal{C}$ convex-separates $\mathcal{D}$. □

We begin by defining a restricted type of cascade. This model corresponds more to the idea of joining the cascade components by their respective output function. Thus, we require that the connection between sequential blocks respects convex-coverings.

**Definition 35.** A *constrained* cascade $D_1 \overset{C_1}{\rightsquigarrow} \cdots \overset{C_{n-1}}{\rightsquigarrow} D_n$ w.r.t. covering $C_1 \times \cdots \times C_n$ is a dynamics $D_1 \rightsquigarrow \cdots \rightsquigarrow D_n$, where $D_i = \langle X_i, U \times C_{[1..(i-1)]}, f_i \rangle$ and $D_i$ is convex-covered by $C_i$.

We can think of a constrained cascade as a feed-forward cascade with connections $D_1 \overset{g_1}{\rightsquigarrow} \cdots \overset{g_{n-1}}{\rightsquigarrow} D_n$ where each $g_i$ is continuous on $U \times C_{[1..i-1]}$.

## E.1 Aperiodic Convex-covered Dynamics and Proof of Theorem 9

We define an analogous notion of aperiodicity for convex-covered dynamics. First we extend the notion of $\eta$-convergence to convex-coverings.

**Definition 36.** For a space $X$, we say a sequence $(x_n)_{n \geq 1} \in X^\omega$ *PC-converges* in $X$, if eventually all its terms lie in the same path-connected component of $X$. ∎

This is an identical notion to $\eta$-convergence, but we give it a different name, since it applies to non-$\eta$-finite spaces.

**Definition 37.** Call dynamics $D = \langle X, U, f \rangle$ *aperiodic* w.r.t. convex-covering $C$, if $D$ is convex-covered by $C$ and if for every sequence $(u_n)_{n \geq 1} \in U^\omega$ PC-convergent in $U$ and $x_0 \in X$, the state sequence $\big(D(x_0, u_{1...n})\big)_{n \geq 1} \in X^\omega \subseteq C^\omega$ is PC-convergent in $C$.

Note the difference in definition: we require that the state sequence is eventually in the same component of $C$, instead of the same component of $X$!

**Lemma 53.** *Let $\mathcal{D} = D_1 \rightsquigarrow \cdots \rightsquigarrow D_n$ be a cascade s.t. $D_i$ is aperiodic w.r.t. convex-covering $C_i$ for $i \in [1..n]$. Then $\mathcal{D}$ is aperiodic w.r.t. convex-covering $\mathcal{C} = C_1 \times \cdots \times C_n$.*

*Proof.* Analogous to proof of Lemma 49, applied to the cascade $D'_1 \rightsquigarrow \cdots \rightsquigarrow D'_n$, where $D'_i = \langle C_i, U \times C_{[1,...i-1]}, f_i \rangle$. □

**Definition 38.** We call a function $F : \Sigma^+ \to \Gamma$ *alternating* if, for some $\sigma \in \Sigma$, the sequence $\big(F(\sigma^n)\big)_{n \geq 1} \in \Gamma^\omega$ changes value infinitely many times. ∎

**Theorem 54.** *Let $D$ be a dynamics aperiodic w.r.t. convex-covering $C$. Let $S_C$ be a shortcut system constrained by $C$ with dynamics $D$. Then $S_C$ can not implement any alternating function.*

*Proof.* Say $D = \langle X, U, f \rangle$ and $S_C = \langle X, U, f, x_0, C, Y, h \rangle$. Suppose for contradiction that $S_C$ with encoder $\mathrm{enc} : \Sigma \to U$ and decoder $\mathrm{dec} : \mathrm{Im}\, h \to \Gamma$ implements an alternating function $F : \Sigma^+ \to \Gamma$.

Let $\sigma \in \Sigma$ be a symbol such that $\big(F(\sigma^n)\big)_{n \geq 1}$ changes value infinitely many times. Since $D$ is aperiodic w.r.t. $C$ we have that $\big(D(x_0, \mathrm{enc}(\sigma)^n)\big)_{n \geq 1} \subseteq X \subseteq C$ is eventually in the same path-connected component of $C$. As $\mathrm{dec} \circ h : C \times U \to \Gamma$ is continuous we thus have that

$$F(\sigma^n) = \mathrm{dec} \circ h\Big(D\big(x_0, \mathrm{enc}(\sigma^n)\big), \mathrm{enc}(\sigma)\Big)$$

is eventually in the same path-connected component of $\Gamma$, i.e. eventually constant. This is a contradiction. □

We now introduce an elementary theorem about convex sets in $\mathbb{R}^d$ (or $\mathbb{C}^d$).

**Theorem 55** (Minkowski's Hyperplane Separation Theorem). *Let $A, B \subseteq \mathbb{R}^d$ be two disjoint, non-empty convex sets. If both are open, then there exists a non-zero vector $v \subseteq \mathbb{R}^d$ and constant $c \in \mathbb{R}$ s.t.*

$$\langle a, v \rangle > c \quad and \quad \langle b, v \rangle < c \quad \forall a \in A, b \in B$$

*with $\langle \cdot, \cdot \rangle$ being the dot product.*

*Proof.* By Section 2.5.1 of [Boyd and Vandenberghe, 2006 - 2004], we have that there exists a non-zero vector $v \subseteq \mathbb{R}^d$ and constant $c \in \mathbb{R}$ s.t.

$$\langle a, v \rangle \geq c \quad \text{and} \quad \langle b, v \rangle \leq c \quad \forall a \in A, b \in B$$

Now, these inequalities in fact must be strict. For contradiction suppose that $\langle a, v \rangle = c$ for some $a \in A$. Since $A$ is open, we have that for some $\epsilon > 0$ $B_{\mathbb{R}^d}(a, \epsilon) \subseteq A$. Thus $a + \epsilon \cdot \frac{v}{||v||_2^2} \in A$ ($||v||_2 \neq 0$ as $v$ is a non-zero vector). But then $\langle a + \epsilon \cdot \frac{v}{||v||_2^2}, v \rangle = a + \epsilon > a$ by linearity of the dot product. Similarly for $B$. $\qquad \square$

**Theorem 9.** *Let $D$ be an $\eta$-finite Linear Recurrent Dynamics, with its state-transition gates having all non-negative eigenvalues. Let $C$ be a covex-regular covering of $D$. Then $D$ is aperiodic w.r.t. $C$.*

*Proof.* Let $D = \langle X, U, f \rangle$ be a Linear Recurrent Dynamics, with $X \subseteq \mathbb{R}^d$, convex-covered by $C$, s.t. $A(u)$ has all its eigenvalues being real, for all $u \in U$. Say $f(x, u) = A(u) \cdot x + B(u)$.

Consider a sequence $(u_n)_{n \geq 1} \in U$, state-convergent in $U$, and $x_0 \in X$. Let $(x_n = D(x_0, u_{1...n})) \subseteq X$ be the corresponding state sequence. We have some $N$ s.t. for $n \geq N$ all $u_n$ are contained in the same component of $U$, we may pick a representative $r \in U$ of that component.

Write $A = A(r), B = B(r)$. By Lemma 25, we have for $n \geq N$ that

$$x_{n+N} \sim_X x'_n = D(x_N, r^{n-N})$$

Like in proof of Theorem 47, we consider the state sequence in the diagonalized space of $A$. Write $A = P^{-1}JP$ for the Jordan normal form of $A$. Here $J$ is block diagonal, with say blocks $J_1, ..., J_s$, $J_b \in \mathbb{R}^{m_b \times m_b}$ being a Jordan Block with $\lambda_b$—eigenvalue of $A$—on the diagonal, and 1 on the right off-diagonal.

Define $y_n = x_{n+1} - x_n$ and $y'_n = P(x'_{n+1} - x'_n)$, then

$$\begin{aligned} y'_{n+1} &= P \cdot (x'_{n+2} - x'_{n+1}) \\ &= P \cdot (Ax'_{n+1} + B - Ax'_n - B) \\ &= PA \cdot (x'_{n+1} - x'_n) = Jy'_n \end{aligned}$$

Thus, unrolling the recurrence we get

$$y'_n = J^n y'_0$$

The $i$-th component of $y'_n$, where $i$ is in say the $b$-th block of $J$, is

$$[y'_n]_i = \sum_{j=i}^{m_b} \lambda_b^{n+i-j} \binom{n}{j-1} [y'_0]_j$$

The binomial coefficients are polynomial in $n$. Thus we may write $[y'_n]_i = \sum v_j \cdot n^{b_j} \cdot a_j^n$, where $b_j \in \mathbb{Z}_{\geq 0}$ and $a_j = \lambda_b \geq 0$, which is of the form in Lemma 64. Since $y_n = Py'_n$, we have

$$[y_n]_i = \sum_{j=1}^{d} [P]_{i,j} \cdot [y'_n]_j$$

which again is of the form in Lemma 64.

Now, for contradiction suppose that $x'_n$ is not state-convergent in $C$. Then, since $C$ has finitely many components, there are two distinct components of $C$, say $C_1, C_2$ such that $x'_n$ is in both $C_1$ and in $C_2$ infinitely often. Furthermore, since $C_1, C_2$ are finite unions of open convex sets, there are convex, open sets $S_1, S_2$ which are disjoint, non-empty, and $x'_n$ is in both $S_1$ and $S_2$ infinitely often (*).

By Theorem 55, there is a non-zero vector $v \in \mathbb{R}^d$ and constant $c \in \mathbb{R}$ s.t. $\langle s_1, v \rangle > c \quad \forall s_1 \in S_1$ and $\langle s_2, v \rangle > c \quad \forall s_2 \in S_2$.

Thus, $\langle x'_n, v \rangle > c$ infinitely often, and $\langle x'_n, v \rangle < c$ infinitely often.

We have

$$\langle y_n, v\rangle = \sum_{i=1}^{d} v_i \cdot [y_n]_i$$

is again in the form from Lemma 64. Thus it is eventually monotone. Therefore eventually $\langle y_n, v\rangle \leq 0$, in or $\langle y_n, v\rangle \geq 0$. By linearity of the inner product

$$\langle y_n, v\rangle = \langle x_{n+1}, v\rangle - \langle x_n, v\rangle$$

Thus, eventually also $\langle x_n, v\rangle$ is monotone—contradiction with (*). □

## E.2 Weakly $\eta$-finite Dynamics

In this section we introduce the topological notion of *connectedness*, as well as the necessary results to establish the finite state properties of GCSs where the state-space coincides with the convex-covering.

**Definition 39.** A topological space $X$ is called *disconnected*, if there are disjoint non-empty sets $H, K$ in $X$ such that $X = H \cup K$. Then $X$ is called *connected* if it is not disconnected.

Connectedness is, as it turns out, a generalization of path-connectedness.

**Fact E.2.1.** (Theorem 27.2, [Willard, 2012]) Every path-connected space is connected.

Similarly to compactness and path-connectedness, connectedness is preserved by continuous mappings and products.

**Fact E.2.2.** (Theorem 26.2, [Willard, 2012]) The continuous image of a connected space is connected.

**Fact E.2.3.** (Theorem 26.10, [Willard, 2012]) A nonempty product space is connected iff each factor space is connected.

Similarly to path-connectedness, connectedness induces an equivalence on the space.

**Definition 40.** For $x \in X$, define $C_x$ as the union of connected subspaces of $X$ containing $x$. We call it the *C-component* at $x$. We write $x \approx_X y$ when $y \in C_x$.

Note, that in [Willard, 2012] C-components are simply referred to as *components*.

**Fact E.2.4.** $\approx_X$ is an equivalence relation, partitioning $X$ into maximal (with respect to inclusion) connected subspaces of $X$. $C_x$ is the equivalence class of $\approx_X$ containing $x$. See Theorem 26.7 and Definition 26.11 of [Willard, 2012] for details.

**Fact E.2.5.** (Theorem 26.12, [Willard, 2012]) The C-components of $X$ are closed in $X$.

Thus, we think of C-components as a partition of the space that is a coarsening of the path-connected components. For an example of a space that has one C-connected component and 2 path-connected components, see the *topologist's sine curve* (Example 27.3, [Willard, 2012]).

**Definition 41.** We call a space $X$ *weakly $\eta$-finite*, if it has finitely many C-components.

*Example* 10. Any finite alphabet is weakly $\eta$-finite, with each symbol being in a separate C-component. ∎

Our goal now is to show that weakly $\eta$-finiteness enjoys the same favourable theoretical properties as $\eta$-finiteness.

**Lemma 56.** *A continuous image of a weakly $\eta$-finite space is weakly $\eta$-finite.*

*Proof.* Let $C_1, \ldots, C_n$ be the C-components of $X$, and let $f : X \to Y$ be continuous. Each $f(C_i)$ is connected, and so $\mathrm{Im}\, f$ is a union of finiely many connected spaces $f(C_1), \ldots, f(C_n)$. Thus, the equivalence classes of $\approx_{\mathrm{Im}\, f}$ must be unions of these images. Thus $\approx_{\mathrm{Im}\, f}$ must have finitely many equivalence classes. □

**Lemma 57.** *The Cartesian product $X \times Y$ space of weakly $\eta$-finite spaces is weakly $\eta$-finite. The C-components of $X \times Y$ are the products of C-components of $X$ and C-components of $Y$.*

*Proof.* Let $C_1, \ldots, C_n$ and $E_1, \ldots, E_{,}$ be the C-components of $X, Y$ respectively. We have $X = \bigcup_{i=1}^{n} C_i, Y = \bigcup_{j=1}^{m}$ and so

$$X \times Y = \Big( \bigcup_{i=1}^{n} C_i \Big) \times \Big( \bigcup_{j=1}^{m} E_j \Big) = \bigcup_{i=1}^{n} \bigcup_{j=1}^{m} C_i \times E_j$$

By Fact E.2.2 each $C_i \times E_j$ is connected. Thus, the C-components of $X \times Y$ are unions of the products $C_i \times E_j$. Now, fix $i \in [1..n], j \in [1..j]$. Let $Z$ be the C-component of $X \times Y$ containing $C_i \times E_j$. consider the projection map $\pi_X : X \times Y \to X$. As the projection is continuous, the image, $\pi_X(Z)$ is connected in $X$. Moreover, $C_i \in \pi_X(Z)$. Thus, as $C_i$ is a maximal connected subspace of $X$, we have $C_i = \pi_X(Z)$. Similarly, considering the projection $\pi_Y : X \times Y \to X$, we have $E_j = \pi_X(Z)$. Since $C_i \times E_j \subseteq Z$, we therefore must have $C_i \times E_j = Z$. Therefore $X \times Y$ has finitely many C-components, and they are the products of C-components of $X$ and C-components of $Y$. $\qquad\square$

**Lemma 58.** *Let $X$ be weakly $\eta$-finite and $\Sigma$ be a finite alphabet. Then $f : X \to \Sigma$ is continuous if and only if it is constant on the C-components of $X$.*

*Proof.* ($\Rightarrow$) Let $f : X \to \Sigma$ be continuous. Let $C$ be a C-component of $X$. By Fact E.2.2, $f(C) \subseteq \Sigma$ is connected, and so $f(C) = \{\sigma\}$ for some $\sigma \in \Sigma$. I.e., $f$ is constant on the C-components of $X$.

($\Leftarrow$) Let $f : X \to \Sigma$ be constant on the C-components. Let $Y \subseteq \Sigma$ be closed. Then $f^{-1}(Y) \subseteq X$ must be a union of finitely many C-components, since $X$ is weakly $\eta$-finite. By Fact E.2.5, we have that each C-component is closed, and therefore also $f^{-1}(Y)$ is closed, as a finite union of closed sets. Thus $f$ is continuous. $\qquad\square$

Now, we have all the properties needed to carry out the arguments in Appendix B.3.

**Definition 42.** We call dynamics $D = \langle X, U, f \rangle$ *weakly $\eta$-finite* if $X$ and $U$ are weakly $\eta$-finite. We call a system $S$ *weakly $\eta$-finite* if its dynamics are weakly $\eta$-finite.

By Lemma 57, we immediatly have that cascades of weakly $\eta$-finite dynamics are weakly $\eta$-finite.

*Example* 11. $\eta$-finite dynamics are weakly $\eta$-finite. $\qquad\blacksquare$

**Theorem 59.** *A convex-covering $C$ is weakly $\eta$-finite, with its C-components coinciding with its path-connected components.*

*Proof.* Let $C_1, \ldots, C_n$ be path-connected components of $C$. Each $C_i$ is a union of finitely many open (in $\mathbb{R}^d$) convex sets, and so is also open. Let $Z$ be a C-component of $C$. Then $Z$ is a union of the path-connected components, and so $Z$ is also open. An open, connected subspace of $\mathbb{R}^d$ is path-connected, see Corollary 27.6 of [Willard, 2012]. Thus $Z$ must actually be one of the path-connected components. $\qquad\square$

**Lemma 60.** *Let $D = \langle X, U, f \rangle$ be a geometrically-contrained system, convex-covered by $C$, with $X = C$. Then $D$ is weakly $\eta$-finite, and the C-components of $X$ are the path-connected components.*

*Proof.* $C$ has finitely many path-connected components, and so it is weakly $\eta$-finite, since path-connectedness implies connectedness. Now, each C-component of $C$ is a union of the path-connected components, all of which are open in $\Omega = \mathbb{R}^d$. Hence each C-component of $C$ is open in $\Omega$. By Corollary 27.6 of [Willard, 2012], C-components of $C$ are therefore path-connected. Thus the path-connected components and C-connected components of $C$ coincide. $\qquad\square$

Since a C-component has to be mapped by a continuous function into a single C-component, we have that a version of Lemma 24 also holds for weakly $\eta$-finite dynamics. For a weakly $\eta$-finite system $S = \langle X, U, f, x_0, Y, h \rangle$ and weakly $\eta$-finite dynamics $D = \langle X, U, f \rangle$, we can thus define the analogous canonical automata

$$\mathcal{C}_{\text{weakly}}(S) = \langle X/_{\approx_X}, U/_{\approx_U}, \tilde{f}, [x_0]_{\approx_X}, \operatorname{Im} h/_{\approx_{\operatorname{Im} h}}, \tilde{h} \rangle$$
$$\mathcal{C}_{\text{weakly}}(D) = \langle X/_{\approx_X}, U/_{\approx_U}, \tilde{f} \rangle$$

with $\tilde{f} : \big([x]_{\approx_X}, [u]_{\approx_U}\big) \mapsto [f(x,u)]_{\approx_X}$ and $\tilde{h} : \big([x]_{\approx_X}, [u]_{\approx_U}\big) \mapsto [h(x,u)]_{\approx_{\mathrm{Im}\, h}}$.

Similarly, replacing path-equivalence $\sim$ with C-component-equivalence $\approx$ in Lemmas 26, 27, 28 and Theorem 1, we get that the canonical automata of weakly $\eta$-finite systems have the same capability in terms of implementing functions.

Likewise, the realization results of Appendix B.4 and Appendix B.5 carry over to the setting of weakly $\eta$-finiteness. Thus we may apply the structural theorems of Algebraic Automata Theory in the case of weakly $\eta$-finite dynamics. We defer exploring the properties of weakly $\eta$-finite dynamics in detail to future work.

### E.3 $\eta$-finite Systems as GCSs and Proof of Theorem 8

We start by showing that $\eta$-finite dynamics that are convex-separated by $C$ can implement exactly the same functions in a $\eta$-finite system as in a GCS constrained by $C$.

**Lemma 61.** *Suppose $\eta$-finite dynamics $D$ are convex-separated by $C$. The following are equivalent:*

- *There is a system with dynamics $D$ that can implement $F : \Sigma^+ \to \Gamma$.*

- *There is a shortcut system $S_C$ constrained by $C$ with dynamics $D$ that can implement $F : \Sigma^+ \to \Gamma$.*

*Proof.* ($\Rightarrow$) Let $S = \langle X, U, f, x_0, Y, h \rangle$ be a system with dynamics $D$ that implements $F$ with some encoder $\mathrm{enc} : \Sigma \to U$ and decoder $\mathrm{dec} : Y \to \Gamma$.

Let $C_1, ..., C_s$ be the path-connected components of $C$. Fix $\gamma \in \Gamma$ and define $h' : C \times U \to \Gamma$ as follows: for $i \in 1 \ldots s$, if $C_i \cap X = \emptyset$, take $h'(c, u) = \gamma$, where $\gamma \in \Gamma$. If $C_i \cap X \neq \emptyset$, take $h'(c, u) = \mathrm{dec} \circ h(x, u)$ for $(c, u) \in C_i \times U$, where $x \in X_i$. This is well-defined: For all $x, x' \in C_i \cap X$, since $C$ is a convex-separator of $X$, we have that $x$ and $x'$ are in the same path-connected component of $X$. Therefore necessarily $\mathrm{dec} \circ h(x, u) = \mathrm{dec} \circ h(x', u)$.

*Want to show*: $h'$ is continuous. Let $\big((c_n, u_n)\big)_{n \geq 1} \subseteq C \times U$ be a sequence converging to $(c, u) \in C \times U$. Then $(c_n)_{n \geq 1}$ converges to $c$ in $C$ and $(u_n)_{n \geq 1}$ converges to $u$ in $U$.

Let $C_i$ be the component that contains $c$. Since $C_i$ is open, there is some $\epsilon > 0$ s.t. $B_\Omega(c, \epsilon) \subseteq C_i$. Since $c_n \to c$, we must have that eventually $(c_n)$ lies in $B_\Omega(c, \epsilon) \subseteq C_i$.Similarly, let $U_j$ be the $\eta$-component of $U$ that contains $u$. Then, by Lemma 20, as $u_n \to u$, we must have that eventually $(u_n)$ lies in $U_j$. Thus eventually $\big(\langle c_n, u_n \rangle\big)_{n \geq 1}$ lies in $C_i \times U_j$. By definition of $h'$, it is constant on $C_i \times U_j$. Thus $\big(h'(u_n, c_n)\big)_{n \geq 1}$ is eventually equal $h'(c, u)$.

Now, define $S_c = \langle X, U, f, x_0, C, Y, h' \rangle$. As $h' : C \times U \to Y$ is continuous, this is a well-def. shortcut system constrained by $C$. Moreover, since $h'$ constrained to $X \times U$ is equal to $\mathrm{dec} \circ h$, we have that $S_C$ with encoder $\mathrm{enc}$ and decoder $\mathrm{id} : \Gamma \to \Gamma$ implement $F$.

($\Leftarrow$) Let $S_c = \langle X, U, f, x_0, C, Y, h \rangle$ be a shortcut constrained by $C$. Suppose that $S_C$ implements $F$ with some encoder $\mathrm{enc} : \Sigma \to U$ and $\mathrm{dec} : Y \to \Gamma$. Then taking $h : X \times U \to Y$ to be the restriction of $h$, we get that the system $S = \langle X, U, f, x_0, Y, h' \rangle$ with encoder $\mathrm{enc}$ and decoder $\mathrm{dec}$ implements $F$. $\qquad \square$

**Lemma 62.** *Let $X$ be a $\eta$-finite space. Then $X$ is convex-separated by some convex-covering $C$.*

*Proof.* Let $X_1, \ldots, X_k$ be the components of $X$. Take

$$\delta = \min_{1 \leq i < j \leq n} \ \inf_{x_i \in X_i, x_j \in X_j} d(x_i, x_j)$$

Then we have $\delta > 0$ by Lemma 20. Define

$$C_i^\delta = \{ B(x_i, \delta/2) \mid x_i \in X_i \}$$

Then $C_i^\delta$ is an open cover of $X_i$. Since $X_i$ is compact, by definition of compactness there is a finite subcover $\bar{C}_i^\delta \subseteq C_i^\delta$ which also covers $X_i$. Moreover, by definition of $\delta$, this subcover does not intersect other components of $X$. Taking $C_i = \bigcup \bar{C}_i^\delta$ we have that $C = C_1 \cup \cdots \cup C_k$ is a convex-covering that convex-separates $X$.

$\qquad \square$

**Construction 5.** FLIP-FLOP dynamics can be implemented by a Linear Recurrent Dynamics with entries in $[\delta, 1 - \delta]$, for some $\delta > 0$.

Let $\epsilon < 1$. Take $D = \langle X, U, f \rangle$ with $X = X_l \cup X_h$, where $X_l = (-1, 0)$, $X_h = (0, 1)$ and $U, f$ such that:

$$f(x, e(\sigma)) = A_\sigma \cdot x + B_\sigma \quad \text{where } \langle A_\sigma, B_\sigma \rangle = \begin{cases} \langle 1 - \epsilon, 0 \rangle & \text{if } \sigma = i \\ \langle \epsilon/4, -1/2 \rangle & \text{if } \sigma = l \\ \langle \epsilon/4, 1/2 \rangle & \text{if } \sigma = h \end{cases}$$

With output function $X_l \mapsto \texttt{low}$ and $X_h \mapsto \texttt{high}$, this implements FLIP-FLOP. The set $C = X$ is a convex-covering of this dynamics.

Hence, Mamba can implement FLIP-FLOP as a constrained system, and so constrained cascades of Mamba blocks can implement any star-free language.

**Corollary 63.** *$\eta$-finite dynamics are in particular convex-separated dynamics, and implement the same functions in $\eta$-finite systems and in GCSs.*

**Theorem 8.** *SSMs with Mamba parametrisation can recognise all star-free languages as GCSs.*

*Proof.* By Construction 5, there is a Mamba block dynamics $D$, with a convex-covering state space, and $\eta$-finite input space, that realise FLIP-FLOP as weakly $\eta$-finite dynamics. A Mamba block can also have a convolution, and so there is a Mamba block dynamics $E$, with a convex-covering state space, and $\eta$-finite input space, that realise $R_2$ as weakly $\eta$-finite dynamics (details omitted. Also a sLSTM-like $\eta$-finite construction is possible, see Appendix G.3). Thus, by weakly $\eta$-finite analogue of Theorem 14, all group-free functions can be realized by feed-forward cascades of $D$ and $E$ components. Such cascades are actually constrained cascades of Mamba block GCSs, since the convex-coverings of $D$ and $E$ coincide with their state-spaces. $\qquad\square$

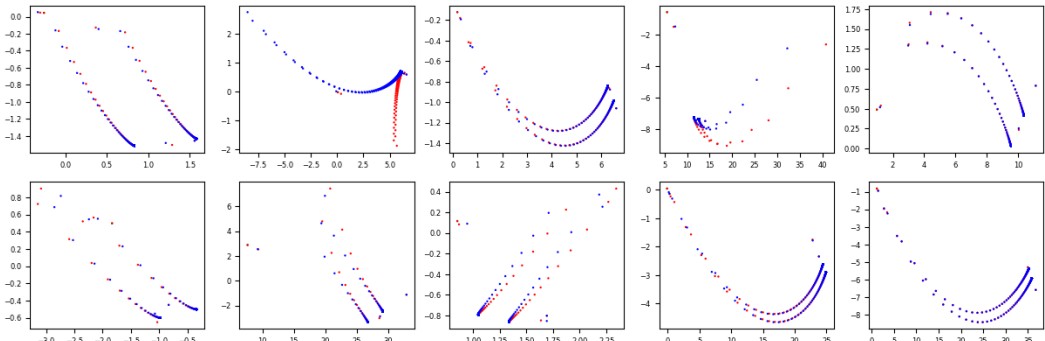

Figure 10: FLIP-FLOP task [Liu et al., 2023]. PCA of a trained 1-layer Mamba states for each channel: red and blue are state sequences under `i0` inputs, starting from `w1` and `w0` respectively. After ≈1000 inputs, both state sequences give the same predictions on the read instruction `r`, incorrectly. The 'doubled' state trajectories are due to each transition consisting of 2 input tokens.

# F  Details of The Experiments

We have created visualizations based on the [Liu et al., 2023] FLIP-FLOP task. The dataset is available at `https://huggingface.co/datasets/synthseq/flipflop/`. The objective of the task is to predictively model a sequence of instructions of the form $sx$, where $s \in$ `w,r,i`, $x \in$ `0,1`. `w` indicates that the next symbol is to be stored, `r` indicates that the next symbol should be the retrieved value and `i` indicates no action. The specific task we trained on corresponds to the "clean" prediction mode, where only prediction following an `r` instruction need to be predicted. We note that the aim of our experiments was to obtain empirical evidence of Mamba having contracting dynamics, and a comprehensive experimental study is beyond the scope of our paper.

We trained 1-layer Mamba on sequence lengths 32, 64, and 512, observing similar state-collapse phenomena, as predicted by our results. Additionally [Sarrof et al., 2024] note that in their experiments Mamba needed more training steps to converge than reported by Liu et al. [2023] for an LSTM. This is another evidence towards the influence of robustness on stability of training.

The code used to perform the experiments is based on the repository shared in Grazzi et al. [2025], with some environment modifications to make it work on the 2025-04-09 Google Colab release. The forked repository is available at `https://github.com/adankow/unlocking_state_tracking`, with a Google Colab notebook file containing the set-up, simple training loop, and hidden state visualisation code.

# G    Additional Proofs and Constructions

## G.1    Monotone Sequence Lemma

**Lemma 64.** *Let $d \geq 1$, $a_1, ..., a_d \geq 0$, $b_1, .., b_d \in \mathbb{Z}_{\geq 0}$ and $v_1, ..., v_d \in \mathbb{R}$. The sequence*

$$x_n = \sum_{i=1}^{d} v_i \cdot n^{b_i} \cdot a_i^n$$

*is eventually monotone.*

*Proof.* If all $v_i = 0$, then $x_n = 0$ for all $n$, in particular the sequence is monotone. Otherwise, we may assume that $v_i \neq 0$ for all $i$, and that

$$a_i > a_{i+1} \quad \text{or} \quad a_i = a_{i+1} \text{ and } b_i > b_{i+1}$$

If $a_1 = 0$, then again $x_n = 0$, and it is monotone. Otherwise, we can take $d_1 : 1 \leq d_1 \leq d$ such that

$$a_i = a_1 \text{ for } 1 \leq i \leq d_1 \quad \text{and } a_i < a_1 \text{ for } i \geq d_1 + 1$$

We may write

$$x_n = a_1^n \cdot P(n) + \sum_{i=d_1+1}^{d} v_i \cdot n^{b_i} \cdot a_i^n$$

where $P(n)$ is the polynomial $\sum_{i=1}^{d_1} v_i \cdot n^{b_i}$.

*Case 1: $a_1 \neq 1$.* We have $a_1 > 0$ and

$$\frac{x_n}{a_1^n} = P(n) + \sum_{i=d_1+1}^{d} v_i \cdot n^{b_i} \cdot (a_i/a_1)^n$$

We have that $(a_i/a_1) \to 0$ as $n \to \infty$, since $a_1 > a_i$ for $d_1 + 1 \leq i \leq d$. On the other hand, $P(n)$ is a non-zero polynomial, since its leading term is $v_1 \cdot n^{b_1}$ and $v_1 \neq 0$, and so $P(n) \to \pm\infty$ as $n \to \infty$. Thus, $x_n \neq 0$ for sufficiently large $n$. Moreover,

$$\frac{x_{n+1}}{x_n} = \frac{a_1^{n+1}}{a_1^n} \cdot \frac{(n+1)^{b_1}}{n^{b_1}} \cdot \frac{P(n+1)/(n+1)^{b_1} + \sum_{i=d_1+1}^{d} v_i(n+1)^{b_i-b_1}(a_i/a_1)^{n+1}}{P(n)/n^{b_1} + \sum_{i=d_1+1}^{d} v_i \cdot n^{b_i-b_1}(a_i/a_1)^n}$$

We have $P(n)/n^{b_1} \to v_1$ as $n \to \infty$, since $v_1 \cdot n^{b_1}$ is the leading term of $P(n)$. Also $n^{b_i-b_1}$ grows at most polynomially, while $(a_i/a_1)^n$ goes to 0 exponentially, since $a_i < a_1$ for $d_1 + 1 \leq i \leq d$. Therefore $\sum_{i=d_1+1}^{d} v_i \cdot n^{b_i-b_1}(a_i/a_1)^n \longrightarrow 0$ as $n \to \infty$. Lastly we have $\frac{(n+1)^{b_1}}{n^{b_1}} \to 1$ as $n \to \infty$. All together

$$\lim_{n\to\infty} \frac{x_{n+1}}{x_n} = a_1 \cdot 1 \cdot \frac{v_1 + 0}{v_1 + 0} = a_1$$

In particular, eventually $x_n$ is positive, or eventually it is negative. There are 4 cases:

- If $a_1 \in (0, 1)$ and $x_n$ is positive eventually, then $x_n$ is decreasing eventually.

- If $a_1 \in (1, \infty)$ and $x_n$ is positive eventually, then $x_n$ is increasing eventually.

- If $a_1 \in (0, 1)$ and $x_n$ is negative eventually, then $x_n$ is increasing eventually.

- If $a_1 \in (1, \infty)$ and $x_n$ is negative eventually, then $x_n$ is decreasing eventually.

Case 2: $a_1 = 1$. We proceed by induction on $b_1$. If $b_1 = 0$, then necessarily $d_1 = 1$, and $P(n) = v_1$. Then we have by Case 1 that $x_n - P(n) = x_n - v_1$ is eventually monotone, and so also $x_n$ is eventually monotone.

For the inductive step, consider

$$y_n = x_{n+1} - x_n$$

$$= P(n+1) - P(n) + \sum_{i=d_1+1}^{d} v_i \cdot a_i^n \cdot \left(a_i(n+1)^{b_i} - n^{b_i}\right)$$

We can again write $\sum_{i=d_1+1}^{d} v_i \cdot a_i^n \cdot \left(a_i(n+1)^{b_i} - n^{b_i}\right)$ as $\sum_{i=1}^{d'} v_i' \cdot n^{b_i'} \cdot (a_i')^n$, with $a_i' < a_1 = 1$. On the other hand $Q(n) = P(n+1) - P(n)$ is a polynomial with leading coefficient of degree $< b_1$. Thus we may apply inductive hypothesis to

$$y_n = Q(n) + \sum_{i=1}^{d'} v_i' \cdot n^{b_i'} \cdot (a_i')^n$$

to conclude that $y_n$ is eventually monotone. Thus, either $x_{n+1} - x_n = y_n \leq 0$ eventually, or $x_{n+1} - x_n = y_n \geq 0$ eventually. Hence $x_n$ is eventually monotone.

$\square$

## G.2   Sequential Cascade Construction

The serial cascade can be realised in terms of the feedforward cascade $\rightsquigarrow$. Consider $i \in 1, 2$ and $D_i = \langle X_i, U_i, f_i \rangle$. Define the *repeat* dynamics on $X_1$ to be the system $R_{X_1} = \langle X_1^2, U \times X_1, r \rangle$, with $r$ given by

$$r\big(\langle x_1, x_2 \rangle, \langle u, x_3 \rangle\big) = \langle x_2, x_3 \rangle \quad \forall x_1, x_2, x_3 \in X_1, u \in U$$

Thus $R_{X_1}$ can delay the propagation of the state of $D_1$ by one time step. Also, define the modified dynamics $D_2' = \langle X_2, U \times X_1^3, f_2' \rangle$, with $f_2'$ given by

$$f_2'\big(x_2, \langle u, x_1, x_{1,old}, x_{1,new} \rangle\big) = f_2\big(x_2, \langle u, x_{1,old} \rangle\big)$$

Note that $R_{X_1}$ is equivalent to the usual repeat dynamics over $X_1$, $\langle X_1^2, X_1, r_X \rangle$, but with input function $(u, x) \mapsto x$.

Now, the feed-forward cascade $D_1 \rightsquigarrow R_{X_1} \rightsquigarrow D_2'$ is well-defined, and has the following transitions:

$$f'\big(\langle x_1, x_{1,old}, x_{1,new}, x_2 \rangle, u\big) = \langle x_1', x_{1,old}', x_{1,new}', x_2' \rangle \quad \text{where}$$

$$x_1' = f_1(x_1, u); \quad x_{1,old}' = x_{1,new}; \quad x_{1,new}' = x_1';$$

$$x_2' = f_2'\big(x_2, \langle u, x_1', x_{1,old}', x_{1,new}' \rangle\big) = f_2\big(x_2, \langle u, x_{1,new} \rangle\big)$$

Now, suppose we have system $S = \langle X_1 \times X_2, U, f, (x_{1,0}, x_{2,0}), Y, h \rangle$ with dynamics $D_1 \ltimes D_2$. Then there is a system $S'$ with dynamics $D_1 \rightsquigarrow R_{X_1} \rightsquigarrow D_2$ which realises $S$: take $S' = \langle X_1^3 \times X_2, U, f', x_0', Y, h' \rangle$ with $x_0' = (x_{1,0}, x_{1,0}, x_{1,0}, x_{2,0})$, $h'(\langle x_{1,1}, x_{1,2}, x_{1,3}, x_2 \rangle, u) = h(\langle x_{1,1}, x_2 \rangle, u)$ and take

$$\alpha : X_1 \times X_2 \to \mathcal{P}_+(X_1^3 \times X_2)$$

$$\alpha(x_1, x_2) \mapsto \{(x_1, x_{old}, x_1, x_2) : x_{old} \in X_1\}$$

Take $\iota : U \to U$ and $\zeta : Y \to Y$ to be the identities. We then have for all $(x_1, x_2) \in X_1 \times X_2$, $u \in U$ and $x' \in \alpha((x_1, x_2))$:

$$f'(x', \iota(u)) = f'(\langle x_1, x_{old}, x_1, x_2 \rangle, u)$$

$$= \langle x_1', x_1, x_1', x_2' \rangle \in \alpha((x_1', x_2'))$$

where $x_1' = f_1(x_1, u)$ and $x_2' = f_2\big(x_2\langle u, x_1 \rangle\big)$, so that $(x_1', x_2') = f\big((x_1, x_2), u\big)$. Moreover $x_0' \in \alpha(x_0)$.

Finally, we have

$$\zeta \circ h'(x', \iota(u)) = h'(x', u) = h'(\langle x_1, x_{old}, x_1, x_2 \rangle, u) = h(\langle x_1, x_2 \rangle, u) = h(x, u)$$

so that indeed $S'$ is a realisation of $S$. Note, that we did not need to introduce any new transitions on $X_1$ or $X_2$ in order to carry out this construction. In particular, if $D_1$ and $D_2$ are linear recurrent dynamics, then $D_1, D_2'$ are linear recurrent dynamics. Also $R_{X_1}$ is a Finite Context Dynamics.

## G.3 Robust Flip-Flop realisations

Recall the sLSTM parametrisation: the state space of a sLSTM is $\mathbb{R}^3$, and the input space is $\mathbb{R}^d$ for some $d \geq 1$. The dynamics function of the form $(\langle c, n, h \rangle, u) \mapsto \langle f_c(\langle c, n, h \rangle, u), f_n(\langle c, n, h \rangle, u), f_h(\langle c, n, h \rangle, u) \rangle$, where

$$f_c(\langle c, n, h \rangle, u) = \psi(l_f(h, u)) \cdot c + \exp(l_i(h, u)) \cdot \varphi(l_z(h, u))$$
$$f_n(\langle c, n, h \rangle, u) = \psi(l_f(h, u)) \cdot n + \exp(l_i(h, u))$$
$$f_h(\langle c, n, h \rangle, u) = \sigma(l_o(h, u)) \cdot \frac{f_c(\langle c, n, h \rangle, x)}{f_n(\langle c, n, h \rangle, x)}$$

where each $l_s : s \in o, i, z, f$ is a function of the form $w_s^t \cdot u + r_s \cdot h + b_s$, for $w_s \in \mathbb{R}^d$, $r_s, b_s \in \mathbb{R}$, $\psi$ is either $\exp$ or $\sigma$, and $\varphi$ is $\tanh$.

### G.3.1 Strongly robust sLTSM FLIP-FLOP realization

We present a construction for a one layer sLSTM FLIP-FLOP, which is strongly robust. The key idea is to only use the $h$ state to implement the dynamics. Then, we can use Theorem 42, and similar arguments involving uniform continuity, to extend the construction to be strongly robust in the states $h, c, n$ and the input space $u$. We shall present the arguments in more detail here, to demonstrate how robustness can be used to prove properties of systems, in particular how to extend robustness to strong robustness.

Let $\psi = \sigma$. Set $w_s = \mathbf{0}$ and $r_s = 0$ for $s = f, i, z$. Set $b_f = -3$, $b_z = 2$, $b_i = 0$. Then we have $l_f \equiv -3, l_i \equiv 0, l_z \equiv 2$. Thus the updates simplify as

$$f_c(\langle c, n, h \rangle, u) = \sigma(-3) \cdot c + \exp(0) \cdot \tanh(2) = \sigma(-3) \cdot c + \tanh(2)$$
$$f_n(\langle c, n, h \rangle, u) = \sigma(-3) \cdot n + \exp(0) = \sigma(-3) \cdot n + 1$$
$$f_h(\langle c, n, h \rangle, u) = \sigma(l_o(h, u)) \cdot \frac{f_c(\langle c, n, h \rangle, x)}{f_n(\langle c, n, h \rangle, x)} = \sigma(l_o(h, u)) \cdot \tanh(2) \in [0, 1]$$

Finally, take $d = 1$ and $l_o(h, u) = u + 10h - 5$.

For now, let us fix $c$ as $c^* = \frac{\tanh(2)}{1 - \sigma(-3)} \approx 1.01202$ and $n$ as $n^* = \frac{1}{1 - \sigma(-3)} \approx 1.049787$, i.e. the fix points of the linear recurrences given by $f_c$ and $f_n$. Then we have that

$$\sigma(-3) \cdot c^* + \tanh(2) = c^* \quad \text{and} \quad \sigma(-3) \cdot n^* + 1 = n^*$$

Moreover, $\frac{f_c(\langle c^*, n^*, h \rangle, x)}{f_n(\langle c^*, n^*, h \rangle, x)} = \frac{c^*}{n^*} = \tanh 2$, so that the update for $h$ simplifies as

$$f(h, u) := f_h(\langle c^*, n^*, h \rangle, u) = \sigma(u + 10h - 5) \cdot \tanh(2)$$

We can set $U = \{u_{\mathtt{set}}, u_{\mathtt{reset}}, u_{\mathtt{id}}\}$, with $u_{\mathtt{set}} = 8$, $u_{\mathtt{reset}} = -8$ and $u_{\mathtt{id}} = 0$, and $H_{\mathtt{low}} = [-0.05, 0.2]$, $H_{\mathtt{high}} = [0.8, 1.05]$

Now, for $h \in [0, 1]$ we have

$$f(\langle c, n, h \rangle, u_{\mathtt{set}}) = \sigma(8 + 10h - 5) \cdot \tanh(2)$$
$$\geq \sigma(3) \cdot \tanh(2) \approx 0.9183$$

Therefore $f(\langle c, n, h \rangle, u_{\mathtt{set}}) \in [0.85, 1]$. Similarly

$$f(\langle c, n, h \rangle, u_{\mathtt{reset}}) = \sigma(-8 + 10h - 5) \cdot \tanh(2)$$
$$\leq \sigma(-3) \cdot \tanh(2) \approx 0.04572$$

Therefore $f(\langle c, n, h \rangle, u_{\mathtt{reset}}) \in [0, 0.05]$. Now, for $h \leq 0.2$

$$\sigma(10h - 5) \cdot \leq \sigma(2 - 5) \approx 0.047426 < 0.05$$

and so $f(\langle c, n, h \rangle, u_{\mathtt{id}}) \in [0, 0.05]$. Also for $h \geq 0.8$

$$\sigma(10h - 5) \cdot \tanh(2) > 0.95 \cdot 0.9 = 0.855$$

and so $f(\langle c, n, h\rangle, u_{\texttt{id}}) \in [0.8, 1]$. Thus we see that the dynamics

$$\Big\langle H = H_{\texttt{low}} \cup H_{\texttt{high}}, U, f = (h, u) \mapsto f_h\big(\langle c^*, n^*, h\rangle, u\big)\Big\rangle$$

realise the FLIP-FLOP dynamics, and is $\eta$-finite and $\epsilon$-robust, for $\epsilon = 0.05$. Furthermore, we can modify the input space $U$, to make it *strongly $\epsilon$-robust*.

Consider $U' = [0, 10]$. $H \times U'$ is compact, and $f$ is continuous on $H \times U'$, so by Theorem 41 it is uniformly continuous on $H \times U'$. In particular, for $\epsilon' = \epsilon/2$, there exists $\delta > 0$ such that

$$||(h, u) - (h', u')|| \leq \delta \implies ||f(h, u) - f(h', u')|| \leq \epsilon'$$

for all $(x, u), (x', u') \in X' \times U'$. Thus, we may take $\delta' = \min(\delta, 1)$ and $U'' = [u_{\texttt{set}} \pm \delta'] \cup [u_{\texttt{reset}} \pm \delta'] \cup [u_{\texttt{id}} \pm \delta']$. Now, consider $h \in H$, $u \in U''$ and $h' \in \mathbb{R}$ such that $||h' - f(h, u)|| \leq \epsilon'$. We have $||u - u'|| \leq \delta'$ for some $u' \in \{u_{\texttt{set}}, u_{\texttt{reset}}, u_{\texttt{id}}\}$, and so

$$\big||f(h, u) - f_h(h, u')\big|| \leq \epsilon'$$

All together

$$\begin{aligned}
\epsilon = \epsilon' + \epsilon' &\geq ||h' - f(h, u)|| + ||f(h, u) - f(h, u')|| \\
&\geq \big||\big(h' - f(h, u)\big) + \big(f(h, u) - f(h, u')\big)\big|| \\
&= ||h' - f(h, u')||
\end{aligned}$$

Since $(h, u') \in H \times U$ and $\langle H, U, f\rangle$ is $\epsilon$-robust, we get that $h' \in H$. Hence $f$ also gives a well defined dynamics function $H \times U'' \to H$, which moreover is $\epsilon'$-robust. Thus, we have $\langle H, U'', f\rangle$ is $\eta$-finite and strongly $\min(\epsilon', \delta')$-robust. It also realizes FLIP-FLOP, since the input components induce the same $\eta$-transitions as $\{u_{\texttt{set}}, u_{\texttt{reset}}, u_{\texttt{id}}\}$ by path-connectedness.

Finally, we extend the dynamics to $c$ and $n$. We can see $f$ as parametrized by $\theta \in [c^* \pm 0.5], \rho \in [n^* \pm 0.5]$, given by

$$f_{\theta, \rho} = \sigma(u + 10h - 5) \cdot \frac{\theta}{\rho}$$

So, $f = f_{c^*, n^*}$. We see that $f_{\theta, \rho}$ is continuous in $\theta$ and $\rho$, and $[c^* \pm 0.5] \times [n^* \pm 0.5]$ is compact. Thus by Theorem 42, there is some $\gamma > 0$ such that $f_{\theta, \rho}$ induces the same function $\overline{H} \times \overline{U''} \to \overline{H}$ as $f_{c^*, n^*}$. Also, similarly to how we extended $U$ to $U''$, we can choose $\gamma$ such that the resulting dynamics are always $\epsilon/4$-robust

Lets take $X = H \times C \times N$ where $C = [c^* \pm \gamma]$ and $N = [n^* \pm \gamma]$. We have that the sLSTM dynamics gives a well-defined, *robust* dynamics function $X \times U \to X$: we already have that the restriction of the dynamics to the $h$ component is robust. For the $c$ and $n$ components, since $\sigma(-3) < 1$, the state updates given by $f_c$ and $f_n$ (which are independent of $u$) are contractions towards $c^*$ and $n^*$ respectively, with rate $\sigma(-3)$. Thus $f_c$ sends $C = [c^* \pm \gamma]$ to $[c^* \pm \gamma \cdot \sigma(-3)]$ and $f_n$ sends $N = [n^* \pm \gamma]$ to $[n^* \pm \gamma \cdot \sigma(-3)]$. All together, the sLSTM dynamics are strongly $\min(\epsilon/4, \delta', \gamma(1 - \sigma(3)))$-robust, and realize FLIP-FLOP.

### G.3.2 Strongly robust sLSTM repeat dynamics

To realize any repeat semiautomata, as defined in Appendix G.2, it is sufficient to realize the two state repeat semiautomaton $R_2 = \langle \{0, 1\}^2, \{0, 1\}, r\rangle$, with $r(\langle x_{\text{old}}, x_{\text{new}}\rangle, x) = \langle x_{\text{new}}, x\rangle$.

Here, the construction is extremely similar to the FLIP-FLOP one. We first show a robust dynamics on just the $h$ cell, using $f(h, u) = \sigma(u + 10h - 5) \cdot \tanh(2)$ which realize $R_2$. Then we can use the same argument as before to extend it to strongly robust dynamics on all 3 cells.

We can use the $h$ cell to represent $x_{\text{new}}$, by simply reusing the previous strongly robust construction for setting the high and low state, with dynamics function $f(h, u) = f_h(\langle c^*, n^*, h\rangle, u)$, state space $H$ and input space $[u_{\texttt{set}} \pm \delta'] \cup [u]$. We then have that for some $\gamma > 0$ for all $c \in [c^* \pm \gamma]$ and $n \in [n^* \pm \gamma]$ the dynamics function $f_h(\langle c, n, h\rangle, u)$ still performs

Define $X_{00} = [-0.01, 0.015]$, $X_{01} = [0.02, 0.05]$, $X_{10} = [0.95, 0.98]$, $X_{11} = [0.985, 1, 01]$ and $u_0 = -8.1, u_1 = 8.1$. Note that $X = X_{00} \cup X_{01} \cup X_{10} \cup X_{11}$ has 4 $\eta$-components. Also, define

$X_0 = X_{01} \cup X_{10}$ and $X_1 = X_{10} \cup X_{11}$. In our construction $X_{ab}$ will correspond to the state of $R_2$ after the last two inputs were $ab$, $a, b \in \{0, 1\}$.

We have

$$f(0.95, u_1) = \sigma(8.1 + 9.5 - 5) \approx 0.999997$$
$$f(1.01, u_1) = \sigma(8.1 + 10.1 - 5) \approx 0.999998.$$

As $\sigma$ is increasing, we therefore have $f(X_1, u_1) \subseteq [0.99999, 1] \subset X_{11}$. Similarly, we have

$$f(-0.01, u_1) = \sigma(8.1 - 0.1 - 5) \approx 0.9526$$
$$f(0.05, u_1) = \sigma(8.1 + 0.5 - 5) \approx 0.9734.$$

Therefore $f(X_0, u_1) \subseteq [0.952, 0.974] \subset X_{10}$. Similarly for $u_0$, we have

$$f(0.95, u_0) = \sigma(-8.1 + 9.5 - 5) \approx 0.0265$$
$$f(1.01, u_0) = \sigma(-8.1 + 10.1 - 5) \approx 0.0474.$$

Therefore $f(X_1, u_0) \subseteq [0.025, 0.0475] \subset X_{01}$. Similarly

$$f(-0.01, u_0) = \sigma(-8.1 - 0.1 - 5) \approx 0.000001$$
$$f(0.05, u_0) = \sigma(-8.1 + 0.5 - 5) \approx 0.000003.$$

Therefore $f(X_0, u_0) \subseteq [0, 0.000004] \subset X_{00}$. Thus $\langle X, \{u_0, u_1\}, f \rangle$ are well-defined dynamics and the 4 $\eta$-components correspond to 4 possible values for the last 2 inputs. Hence clearly they can realize $R_2$. Moreover, the dynamics are strongly robust. The remainder of the argument is the same as for the FLIP-FLOP construction.

### G.3.3 Strongly robust Elman-RNN FLIP-FLOP construction

The following is a modification of a construction in [Knorozova and Ronca, 2024a]. Consider the dynamics function

$$f(x, u) = \tanh(2 \cdot x + u)$$

for $x, u \in \mathbb{R}$. We have that for all $x, u$, $f(x, u) \in [-1, 1]$. Define $X_{\texttt{low}} = [-1.1, \tanh(-1)]$, $X_{\texttt{high}} = [\tanh(1), 1.1]$. Note that $\tanh(1) \approx 0.76159$, $\tanh(-1) \approx -0.76159$

We have

$$f(-1.1, 4) = \tanh(-2.2 + 4) \approx 0.9468$$
$$f(1.1, 4) = \tanh(2.2 + 4) \approx 0.999992$$

As $\tanh$ is increasing, we have $f([-1.1, 1.1], 4) \subseteq [0.9467, 0.999993] \subset X_{\texttt{high}}$. Similarly, $f([-1.1, 1.1], -4) \subseteq [-0.999993, -0.9467] \subset X_{\texttt{low}}$. Moreover

$$f(\tanh(1), 0) = \tanh(2 \cdot \tanh(1)) \approx 0.909$$
$$f(1.1, 0) = \tanh(2 \cdot 1.1) \approx 0.9757$$

Thus, $f(X_{\texttt{high}}, 0) \subseteq [0.908, 0.9757] \subset X_{\texttt{high}}$. Similarly $f(X_{\texttt{low}}, 0) \subseteq [-0.9757, -0.909] \subset X_{\texttt{low}}$. Thus we see that, taking $X = X_{\texttt{low}} \cup X_{\texttt{high}}$, $u_{\texttt{set}} = 4$, $u_{\texttt{reset}} = -4$, $u_{\texttt{id}} = 0$, the $\eta$-finite dynamics $\langle X, \{u_{\texttt{set}}, u_{\texttt{reset}}, u_{\texttt{id}}\}, f \rangle$ are well-defined, and realize FLIP-FLOP. Also clearly they are robust. Now, by the same argument as for the sLTSM FLIP-FLOP realisation, we can extend the input space, using Theorem 16 and Theorem 42, to obtain a strongly robust construction.

## H    Further Discussion on Related Work

Sarrof et al. [2024] show that, in the finite-precision setting, regular languages that can be modelled by diagonal linear-recurrences with non-negative entries—like Mamba—are precisely the star-free languages. The setting differs from ours, in that it allows finite fractional precision, but unbounded number of integer bits. With that, a number of positive expressivity results for counter languages is given. The empirical experiments show that SSMs indeed can model such languages on in-distribution lengths, but with limited length-generalisation. The finite-precision arguments in this work are not fully formal, essentially ignoring the error of the linear dynamics carried out in finite precision. Weiss et al. [2018] use the same finite-precision setup with unbounded integer bits to show that ReLU-activated Elman-RNN and LSTM can implement counting behaviour, while Elman-RNNs with squashing activations and GNU [Cho et al., 2014, Chung et al., 2014] cannot.

Grazzi et al. [2025] extend Mamba and DeltaNet parametrisations to allow for gates with negative eigenvalues. The work proves that linear recurrences, with gates having non-negative eigenvalues, are restricted to modelling star-free recurrent languages in the *finite-precision* setting. Their framework differs from [Sarrof et al., 2024] and ours, assuming that the linear recurrence is computed in convolutional form in some finite datatype $\mathbb{D}$, with some operations carried out in infinite precision before casting back into $\mathbb{D}$. This setting is more explicit in its assumptions than [Sarrof et al., 2024], but not generalisable to other types of recurrence.

Merrill et al. [2024] use the parallelisability aspect of SSMs to obtain an expressivity classification in terms of Circuit Complexity in the *log-precision setting*, i.e., precision logarithmic in the input length. Assuming a particular datatype is used to carry out the operations, it shows that SSMs, including Mamba, can be simulated in the TC-0 circuit class. Thus Mamba is unable to solve the $S_5$ word problem, even with log-precision, under the widely-accepted conjecture that TC-0 $\neq$ NC-1. The log-precision framework offers a unique perspective on the drawbacks of parallelism of SSMs.

The Turing-completeness capabilities of Elman-RNNs as *offline models* of computation are studied in [Siegelmann and Sontag, 1995, Kilian and Siegelmann, 1996, Hobbs and Siegelmann, 2015, Chung and Siegelmann, 2021]; differently, we study RNNs as online models, reading input elements as they arrive. A form of *asymptotic expressivity* of RNNs is studied in [Merrill et al., 2020], when weights tend to infinity; differently, we consider actual weights. A rich literature surveyed in [Strobl et al., 2024] focuses on the expressivity of Transformers, that constitute an alternative to RNNs as they also operate on sequences.

