# OpenReview forum: "Metric Automata Theory: A Unifying Theory of RNNs"
_NeurIPS.cc/2025/Conference — NeurIPS 2025 poster_

### Official Review · Reviewer_ycc4 · 2025-06-17

**Clarity:** 2
**Significance:** 2
**Originality:** 3
**Rating:** 5
**Confidence:** 4

**Summary:**

## Summary
In this paper the authors propose a unifying theory to mainly study the expressivity of Recurrent Neural Networks (RNNs) from a language-theory point of view. The principle linking continuous dynamical systems to finite-state machines such as Automata, is the discretization of latent space (hidden space). In this work, the authors turn to the tools of algebra/topology to partition the latent space into disjoint subspaces forming equivalence classes. Based on this approach, the authors derive a number of notions such as $\eta-$finiteness, $\eta-$convergence, $\epsilon-$rebustness, to name just a few. These notions form the basis of theoretical results on the expressivity power of RNNs, for example, the non-$\epsilon-$rebustness of SSMs, a widespread class of recurrent neural networks.

**Questions:**

## Questions

- Have you thought of a way of analyzing the latent space to determine equivalence classes more accurately? If not, do you have in mind a construction of an RNN that has a latent space that is well split into equivalence classes, and that is not the trivial split derived from finite precision?

- The previous question brings me to this one. Do you think there is a non-trivial way to apply the Krohn-Rhodes theorem to RNNs, from a practical point of view? There are algorithms for doing cascade decomposition of semigroups, but the compléxité is often of type $n^n$, which is not tractable at all when the discretization of the latent space is derived from finite precision.

**Ethical Concerns:**

["NO or VERY MINOR ethics concerns only"]

**Final Justification:**

Rephrase the article so that ideas for proof techniques appear in the body of the text. Possibly illustrate key concepts to make the article more easily recognizable and digestible for readers.

**Limitations:**

The fact that any RNN implemented in finite precision is $\eta-$finite is not in doubt, but if we take a vanilla RNN with $h_{t+1} = \sigma\left(Ux_t + Wh_t + b\right)$, where the dimension of the vector $h_t$ is $d\geq1$. The sigmoidal function has values in $[0,1]$, and the interval $[0,1]$ in finite precision is finite, so $\Gamma := Card([0,1])$. So if there is not a more refined way of determining $\eta$ once we want to associate a state machine to the RNN, we will potentially end up with a number of states $\approx \Gamma^d$.

**Quality:**

3

**Strengths And Weaknesses:**

## The strengths
- The article is well written.
- The subject matter is complicated and of great interest to the community.
- The ambition of the work is to propose a unifying theory for RNN studies, and indeed the proposed theoretical framework has the potential to fulfill this role.
## Weaknesses
- There are no illustrations. For a mathematician, the concepts are accessible, but this may not be the case for practitioners who have no grounding in abstract algebra. I think an illustration or two to represent equivalence classes would be a big plus.
- From a mathematical point of view, the work is rigorous and all the proofs, in my view, are valid. Nevertheless, the results demonstrated are not innovative. All the proofs are based on known techniques and the assertions are not surprising.
The authors propose the notion of $\eta-$finitness, and provide several constructions of xLSTM and Elman RNN that are $\eta-$finite and $\epsilon-$robust. Nevertheless, it is not clear, outside the framework of finite precision, whether RNNs have the characteristic of being $\eta-$finite.
- Empirical validation comes from other works, and concerns only SSM-type models.
- In my opinion, the 9-page format is not at all appropriate for this work. A large part of the notions and proofs ended up in the appendix. I feel that a theoretical article should at least provide a sketch of proof of the theorems announced, in the article section.

---

> ### Author Rebuttal · Authors · 2025-07-30
>
> Thank you for your in depth review. We are glad that you found our work well-written, and that it has the potential to be unifying for the field.
>
> We address your questions, and then concerns in order. However, we feel there might be some overall confusion related to etafiniteness, so we first provide some clarification on this aspect.
> We look forward to your further feedback and thoughts.
>
> ## $\eta$-finiteness
> $\eta$-finiteness is primarily a framework for describing RNN expressivity, and deriving results for concrete finite precision implementations. $\eta$-finiteness is a mathematical property of a continuous system, regardless of whether it is realised in finite precision. It is worth noting that systems with infinite cardinality state space can be $\eta$-finite.
>
> ## Questions
> > Have you thought of a way of analyzing the latent space to determine equivalence classes more accurately? If not, do you have in mind a construction of an RNN that has a latent space that is well split into equivalence classes, and that is not the trivial split derived from finite precision?
>
> We feel there might be some misunderstanding related to the notion of etafiniteness. See clarification above.
>
> For an example of such an RNN that is not linear, consider the construction of FlipFlop for a $\tanh$ activated neuron, Section G.3.3 in the appendix. Its state space has 2 equivalence classes: $[-1.1, \tanh(-1)]\approx [-1.1, -0.76]$ and $[\tanh(1), 1.1] \approx [0.76, 1.1]$. Now, for a finite precision implementation, we can still assign these equivalence classes, as long as the error with which tanh is computed is less than $0.1$, due to robustness of the dynamics. Note that in this case, the RNN has the state-space well-split (consistent with the dynamics) that is not the trivial split derived from finite precision.
>
> > The previous question brings me to this one. Do you think there is a non-trivial way to apply the Krohn-Rhodes theorem to RNNs, from a practical point of view? There are algorithms for doing cascade decomposition of semigroups, but the compléxité is often of type $n^n$, which is not tractable at all when the discretization of the latent space is derived from finite precision.
>
> We are not sure there is much practical applicability of computing the full decomposition. Aside from computational constraints, Krohn-Rhodes does not tell us e.g. what the complexity of the connecting functions should be. However, just knowing what prime factors the transition semigroup has can be helpful in picking the architecture. For instance, knowing that the largest factor has $N$ states tells us that it is realisable by a cascade of $N$-state automata. This could potentially guide decisions about dimension size of hidden states, tokenization or quantization algorithms. [Grazzi et al., 2025] on page 7 discusses the connection between the number of tokens per transition and the GH matrix depth required for the transition—e.g. having 2 tokens, like in “3+2+4=4” essentially allows the model two steps per transition.
>
> ## Further Concerns
>
> > There are no illustrations.
>
> We plan on adding figures to the main body, to build more visual intuition for the concepts and proof techniques presented. We will add improved versions of the figures present in the supplementary material, as well as new ones. The existing figures we will include in the main body are:
>
> * Figure 3 (page 17 supplementary), visualising correspondence between the continuous $\eta$-finite dynamics and the canonical semiautomaton. The last transition will be changed to land in the same component.
>
> * Figure 5 (page 25 supplementary), visualising a concrete datatype covering of a robust dynamics. To the right, we wil merge Figure 4 (page 22 supplementary) together with a symbolic definition of robustness.
>
> * Figure 6 (page 30 supplementary), visualising aperiodicity and the key argument behind Theorem 4
>
> * Figure 1 (main body page 8) visualising empirical behaviour of a trained Mamba model on the FlipFlop task. To the right we will add a diagram of the FlipFlop transitions.
>
> The new figure we will include in the main body is:
>
> * A figure visualising transitions of a GCS together with the hyperplanes which piecewise separate the state space. To the right we will visualise the key argument behind Theorem 9.
>
> We will also add figures visualising the constructions in the appendix. Any further suggestions for changes to the figures, or new figures to be added to the appendices, would be greatly appreciated.
>
> > Nevertheless, the results demonstrated are not innovative. All the proofs are based on known techniques and the assertions are not surprising.
>
>
> We would like to emphasise that our approach is innovative and leads to novel results, including surprising ones. We expand on this by discussing concrete points.
>
> * First, our result on Mamba being unable to implement FlipFlop is in contrast to the existing literature. This is a surprising and important fact, which was missed in the existing experimental evaluation due to limited length of considered inputs. This new result calls for a rethinking of the empirical validation process for unbounded-length expressivity results, as the current approach of testing for limited length-generalisation is inadequate and only shows the partial picture.
>
> * Second, our results on robustness and geometrically-constrained systems (GCS) introduce novel insights into existing experimental results on RNNs. In particular, GCS allows, for the first time, to describe the mode of limited length-generalisation which Mamba displays on star-free tasks such as FlipFlop.
>
> * Third, our framework is innovative and has the potential of accelerating future research, as well as making it more rigorous. We demonstrate this by proving existing results of [Sarrof et al., 2024] and [Grazzi et al., 2024] within the etafiniteness framework. While these particular results are known, within our framework we are able to simplify the proof technique. Moreover, the proofs in [Sarrof et al., 2024] had an issue of asserting without proof that convergence of the analytical form of the state sequence implies it being eventually constant, on account of finite precision, which is not necessarily the case. On the other hand, the set up of [Grazzi et al., 2024] required simplifying assumptions about finite-precision arithmetic. Our approach’s contribution is making the intuitive arguments in Sarrof rigorous, and moreover being able to obtain results on concrete finite precision implementations via robustness.
>
>
> >Nevertheless, it is not clear, outside the framework of finite precision, whether RNNs have the characteristic of being $\eta$-finite.
>
> Determining whether a given model is etafinite is an interesting point for further study. Please see also the first paragraph of our response to Reviewer sV21.
>
> >In my opinion, the 9-page format is not at all appropriate for this work.
>
> The main objective of our work is to present a new unifying framework for RNN research, and motivate its capabilities to explain empirical results and inform future research, so that hopefully it gains adoption in the wider community. Now, we think that the conference format, even though not capable of containing the entire technical work, will expose our framework to a wider audience, increasing its chances of becoming a unified framework for future studies.
>
> Now, we hope that the main body can achieve two objectives. The first one is to provide a digestible overview of our main results, to demonstrate the capabilities of MAT, as well as to share our new insights. The second one is to serve as a starting point to the supplementary material, for those interested in theoretical expressivity study of RNNs, so that they can further investigate the introduced notions, or invent new frameworks within MAT, such as GCS.
>
> >  I feel that a theoretical article should at least provide a sketch of proof of the theorems announced, in the article section.
>
> We plan to reduce the size of Section 2 Preliminaries in favour of giving more details on the proofs and techniques used in Section 4. Additionally, the figures we will include in the main body illustrate key ideas behind some of the proofs.
>
> > Empirical validation comes from other works, and concerns only SSM-type models.
>
> It is true that we focus on developing the theoretical framework, deferring new experiments to future work. It should be noted that we also consider results for xLSTM and LSTM, in addition to SSM-type models.
>
> > [...] So if there is not a more refined way of determining $\eta$ once we want to associate a state machine to the RNN, we will potentially end up with a number of states $\approx \Gamma^d$.
>
> We feel there might be some misunderstanding related to the notion of $\eta$-finiteness. See clarification above. Please also note that $\eta$ is not a variable, unlike in the case of $\epsilon$-robustness or $\epsilon$-nets. It is just a name, like $\beta$-reduction.
>
> Importantly, the number of $\eta$-states is not immediately related to the numerical precision used to implement the dynamics.
>
> Regarding the vanilla RNN within etafinite framework, regardless of any finite datatype considerations, it is possible to implement FlipFlop where we have 2 equivalence classes—the construction is analogous to the $\tanh$ one mentioned above. Furthermore, this construction can be shown to be robust, and hence by our theory the equivalence classes will be valid in finite precision representation, given sufficient precision. Explicitly, the floating points lying within one of the two equivalence classes will be mapped in a consistent way. Thus it is possible to consider such a finite-precision system as having 2 meaningful states.
>
> ## References
> * Sarrof et al., The expressive capacity of State Space Models: A formal language perspective. (NeurIPS), 2024.
> * Grazzi et al., Unlocking state-tracking in linear RNNs through negative eigenvalues. (ICLR), 2025.

---

> > ### Comment · Reviewer_ycc4 · 2025-08-04
> > **Score update**
> >
> > Thank you very much for your very detailed reply. I don't agree with your remark that there are illustrations in the appendix. The conference format is 9 pages and some technical stuff in the appendix, but in your case it seems to be a 50+ page article cut in two... Nevertheless, this is a rigorous piece of work, and you have shown me your intention to adjust the format to make it more suitable.
> >
> > Furthermore, my questions concerning the practical application of the concepts and the KR decomposition of NRNs are intended to draw attention to the fact that theoretical tools should as far as possible be practical tools. The framework you propose is very interesting for studying expressivity, but perhaps there are bridges to be built with the interpretability of RNNs? The cascade decomposition of an RNN would enable us, for example, to have an explanation of how the RNN works, or at least a clearer understanding.
> >
> > I raise my rating to 5

---

### Official Review · Reviewer_MndJ · 2025-07-01

**Clarity:** 3
**Significance:** 3
**Originality:** 3
**Rating:** 5
**Confidence:** 3

**Summary:**

The paper introduces Metric Automata Theory, a novel framework that connects classical automata theory to continuous dynamical systems, particularly focusing on recurrent neural networks (RNNs). It establishes connections between this framework and automata theory to systematically analyze the expressivity of various state-based neural models.

**Questions:**

Could the authors elaborate more generally on how they envision this framework being utilized in future research? In particular, aside from the demonstrated connection via the "FLIP-FLOP" task, what other connections to automata theory might be explored within this new theoretical setting? Making the case that this framework can be extended convincingly, and adding such a section to the paper, will probably result in a raised score.

**Ethical Concerns:**

["NO or VERY MINOR ethics concerns only"]

**Final Justification:**

I am convinced by the author's claim regarding the importance of the generic connection to automata theory, and therefore, as mentioned in my original review, I raise my score to a 5.

Less importantly, I am also glad the authors will include a limitation section and improve the preliminaries section, as I think this will greatly benefit the paper.

**Limitations:**

I feel that the paper would benefit from having a dedicated limitations section, perhaps placed in the appendix. Consolidating caveats (such as those highlighted in the strengths and weaknesses above) into a single location would help readers better understand the context and scope of the presented results.

**Paper Formatting Concerns:**

None.

**Quality:**

4

**Strengths And Weaknesses:**

**Strengths**:
1. This paper presents an interesting and powerful framework.
2. The paper is (at least subjectively) interesting. I enjoyed reading it.
3. The stated results are strong, insightful, and broadly applicable to a variety of state-based models, including widely-used architectures such as xLSTMs and SSMs.
4. Although I'm not deeply familiar with all prior related works, the connections drawn are neat and provide a valuable perspective.

**Weaknesses**:
1. The preliminaries and introduction of dynamics are math-heavy and quite general. The paper's preliminaries are quite heavy. Which is understandable for a paper presenting a new framework, but Iwasn't entirely convinced that all elements presented were strictly necessary. While establishing a connection to automata theory is interesting, it's unclear whether this warrants a fully separate framework.  Specifically, the primary practical usage demonstrated is through the "FLIP-FLOP" task, and it's not clear how broadly this connection or generality could be leveraged beyond this specific application. In general, when introducing a new framework, it would be beneficial if the authors explicitly outlined potential future directions, better explained why such generality is necessary
2. A minor point (much less important then point 1): The results on expressivity assume non-negative eigenvalues for SSMs. However, many widely-used SSM architectures, including LRU, S4, and occasionally Mamba, make use of complex-valued eigenvalues. Prior work has demonstrated that allowing complex parameterizations significantly impacts model expressivity (e.g., "Provable Benefits of Complex Parameterizations for Structured State Space Models" and "Universality of Linear Recurrences Followed by Non-linear Projections: Finite-Width Guarantees and Benefits of Complex Eigenvalues"). It might thus be worthwhile to mention explicitly that restricting to non-negative eigenvalues constitutes a limitation of the current theory.

---

> ### Author Rebuttal · Authors · 2025-07-30
>
> Thank you for your review of our work and your detailed questions. We are very glad you found our work interesting and insightful.
>
> We address your question and then further concerns. We are looking forward to your reply and further questions.
>
> ## Question
> > Could the authors elaborate more generally on how they envision this framework being utilized in future research? In particular, aside from the demonstrated connection via the "FLIP-FLOP" task, what other connections to automata theory might be explored within this new theoretical setting? Making the case that this framework can be extended convincingly, and adding such a section to the paper, will probably result in a raised score.
>
> __Connection to Automata Theory__. The connection we develop between continuous systems and Automata Theory in the paper serves two main purposes discussed below. Furthermore, Metric Automata Theory is a generalisation of Automata Theory to continuous systems, which is of independent interest.
>
> * The first purpose is using decomposition theorems of AAT to analyse cascades of systems. In our paper, FlipFlop cascade decomposition has a particular importance, because it characterises star-free languages. However, the developed framework is not restricted to FlipFlops and star-free languages. For instance, the modular counting construction in [Grazzi et al., 2025], Theorem 6, is still valid in the etafinite set-up, and so it gives us a positive result for a different class of regular languages. In general, our framework allows for applying AAT to the study of RNN expressivity, in terms of all prime automata, including FilpFlop, modular counters and many others.
>
> * The second reason is more core to the theory as a whole, and perhaps is not emphasised enough throughout the main body. Automata Theory is key in describing the behaviour of a continuous dynamical system in a way that is suitable for theoretical analysis. In particular, the notion of canonical automaton allows us to abstract away the details of behaviour that do not affect expressive capacity.
>
> As an example, realisability is an algebraic automata concept which is used in proofs of Theorem 2 (realisability of robust systems with sufficient precision) and Theorem 42 (on robust parametrised systems stability), even though the results themselves are not about automata. The general continuous setting is not suitable for considering an analogous definition of realisability, because we do not actually care about realising the local behaviour of a dynamics, just the global behaviour. Thus, the connection to automata is actually __a core tool we use to develop an algebraic theory of continuous systems__.
>
> We will adjust Section 3.1 to better explain the key part that the connection to automata theory plays in MAT.
>
> __Future Research__. The framework we set up opens up many avenues for future research in connection to automata theory. Krohn and Rhodes decomposition can be further utilised to derive expressivity results using semigroup decomposition hierarchies. Additionally, this connection is central to the overall theory of $\eta$-finite systems and robustness. We especially see robustness as being of practical interest and as a subject of future research.
>
> We also believe it has great applicability to natural language tasks, for developing insights on how the way automata arise in the tasks influence models performance. For instance, Theorem 4 states that LRDs cannot robustly implement FlipFlop. However, if we remove the identity transition, that is no longer the case. Essentially, only some kinds of transitions are problematic, and the models performance might improve if it is possible to remove such inputs. Another interesting avenue of future research is on how tokenization affects the models ability to perform state-tracking and realise automata transitions. For example, [Grazzi et al., 2025](paragraph under Theorem 3, page 7) note that allowing more input symbols per transition (eg “3+2+4=4”) allows simpler gates to implement automata.
>
> The Geometrically-constrained systems theory also utilises the framework of canonical automata, e.g. for Theorem 8. We believe the GCS theory is of theoretical and practical interest, independently of the theory of $\eta$-finiteness and finite state automata. In particular, we believe it is suitable for investigating expressivity of general formal languages, beyond regular. We are interested in investigating, in future work, counter languages (see e.g., Construction 1) within GCS, e.g. as mentioned in relation to Mamba performance, described in lines 385-388.
>
> We will add a dedicated Future Research section to the main body to better emphasise these applications.
>
> ## Further Concerns
>
> > The results on expressivity assume non-negative eigenvalues for SSMs.
> > It might thus be worthwhile to mention explicitly that restricting to non-negative eigenvalues constitutes a limitation of the current theory.
>
> Most results we present indeed relate to SSMs with non-negative eigenvalues. However, e.g., Theorem 4 applies to SSMs with both real and complex gates and states, without restriction on the eigenvalues. Indeed, a feature of our framework is the ability to work with arbitrary metric spaces.
>
> In fact, one of the key advantages of the etafinite approach to finite precision is the ability to rigorously consider the state as being a subspace of various ambient spaces, and to e.g., consider the diagonalized space for a particular linear transformation. Such approaches are problematic when working with fixed datatypes, cf. [Grazzi et al., 2025] proof of Theorem 1, where the simplifying arithmetic assumption is needed for diagonalization.
>
> We will improve the articulation of Section 4.1 to better convey the generality of the results.
>
> > I feel that the paper would benefit from having a dedicated limitations section, perhaps placed in the appendix. Consolidating caveats (such as those highlighted in the strengths and weaknesses above) into a single location would help readers better understand the context and scope of the presented results.
>
> Based on feedback, we intend to add a separate limitations section and future work section in the main body. The main points we will discuss in the limitations section are the following:
>
> * __Use of the dynamical system formalism__: this approach yields an elegant and general framework and enables powerful techniques of analysis. However, it comes with a few  draw-backs. Firstly, requiring continuity throughout means that the main work of assigning meaning to the states is done in selecting the state space $X$. Further, the dynamics $f$ needs to have codomain $X$. This can make verifying constructions tedious. In the context of learning parameters for $f$, this also means that as the parameters vary, the state space must change accordingly. Hence, analysing a parametrised model requires considering a family of dynamical systems, which could be further developed as a separate framework within MAT. Nonetheless, the current formulation of MAT allows for indirect analysis of learning stability, via the notion of robustness (as discussed in Section 3.2 on robustness).
> * __Focus on unbounded length expressivity__: Most of the framework and techniques presented, aside from robustness, study asymptotic behaviour of RNNs. We do not aim to determine e.g. how the number of $\eta$-components or layers needed grows with input length. Similarly, Krohn-Rhodes decomposition techniques are not suitable for e.g., determining the minimal number of components needed.
> * __Focus on state-space transformations__: Our framework does not directly model the input and output function complexity in the network, which also influences the expressive capacity of the model. We can still indirectly control what transformations are allowed in the network by imposing additional constraints aside from continuity, e.g., like in the geometrically-constrained setting. When we consider cascade constructions, generally connecting functions implementable by MLPs suffice, thanks to separability of components.
>
> We would like to note that, in our view, points related to the limitations of Algebraic Automata Theory can __potentially motivate a renewed development of the field__, in conjunction with modern expressivity studies. We plan on including a more detailed comparison with alternative frameworks, mentioned in the Related Work section, in the appendix.
>
> >The preliminaries and introduction of dynamics are math-heavy and quite general. The paper's preliminaries are quite heavy.
>
> To address that, we plan to reduce the content of Section 2 Preliminaries, in favour of giving more details on the proofs and techniques used in Section 4.
>
> ## References
> * Riccardo Grazzi, Julien Siems, Arber Zela, Jörg K. H. Franke, Frank Hutter, and Massimiliano Pontil. Unlocking state-tracking in linear RNNs through negative eigenvalues. In Proceedings of the Thirteenth International Conference on Learning Representations (ICLR), 2025.

---

> > ### Comment · Reviewer_MndJ · 2025-08-02
> > **Raising score to 5**
> >
> > I thank the authors for their thoughtful and detailed rebuttal.
> >
> > I am convinced by the author's claim regarding the importance of the generic connection to automata theory, and therefore, as mentioned in my original review, I raise my score to a 5.
> >
> > Less importantly, I am also glad the authors will include a limitation section and improve the preliminaries section, as I think this will greatly benefit the paper.

---

### Official Review · Reviewer_sV21 · 2025-07-03

**Clarity:** 2
**Significance:** 3
**Originality:** 3
**Rating:** 4
**Confidence:** 2

**Summary:**

This paper introduces Metric Automata Theory (MAT) that applies to dynamical systems, with the specific case of Recurrent Neural Networks (RNN) typically working on sequences of inputs.
The paper presents $\eta$-finiteness (for a set, being an union of compacts called $\eta$-component, for a dynamic, having input and state space $\eta$-finite) $\eta$-robustness for a dynamic (the transition function always end up in the state space (typically $\eta$-finite), even if some noise $\varepsilon$ is added to it) and geometrically-constrained systems.
The robustness allows to cover the cases of finite precision implementations without resorting to the general argument of floats being discrete.
Expressivity results of Vanilla-RNNs, xLSTM and SSM such as MAMBA are derived using MAT, and compared to empirical behaviors for language recognition.

**Questions:**

- To what extent can the concepts, e.g. finiteness or robustness, be used on actual learned model?

**Ethical Concerns:**

["NO or VERY MINOR ethics concerns only"]

**Final Justification:**

This paper is interesting and the discussion period was productive. I am for acceptance, provided the restructuration of the paper is done to improve its exposition.

**Limitations:**

yes

**Quality:**

3

**Strengths And Weaknesses:**

The proposed theory is interesting to take a discrete viewpoint on continuous dynamical systems and understand the behavior of RNNs for language recognition.
While I could not check all the derivations, the paper seem sound.

The paper introduces many definitions and theorems and it is easy to lose track of the articulation between these and their implications. For instance, some theorems suppose $\eta$-finiteness and some constructions ensure it, but it is unclear to me if the concepts can be used on trained/learned models beyond these constructions (e.g. can we say or prove that a given model is $\eta$-finite).

The paper is probably too dense for the current format (there are 50 pages of supplementary material, which I could not check): while it reads relatively well given the amount of definitions and concepts, the body of the paper would have benefited from illustrations of the main concepts (following some of the figures from the supplementary).

Remarks

- l87: I might be missing a point of definition but how can a continuous function have a finite codomain?
- l359: about "predicted by Th. 8", actually Th. 8 allows mamba to represent the language ("expressivity" / "can recognize") it but, as far as I understand, nothing is stated about the learnability (the fact that mamba will actually manage to learn the parameters)

Local comments

- l167: "give" -> "given"
- l272: "start-free" -> "star-free"

---

> ### Author Rebuttal · Authors · 2025-07-30
>
> Thank you for your detailed review. We greatly appreciate your in-depth questions and attention to detail.
>
> We address your question, and then further concerns. We look forward to your response and further suggestions for improving our work.
>
> ## Question
> > To what extent can the concepts, e.g. finiteness or robustness, be used on actual learned model?
>
> The main focus of the paper is to establish a framework to analyse the expressive capacity of RNNs, which concerns whether a certain class of networks admits instances capable of performing certain tasks. Thus, the introduced notions and theorems provide notions and properties that altogether constitute such a framework for the study of expressivity and hence, their primary application is not to learning. Nonetheless, we argue in the paper that they are useful tools for analysing empirical phenomena in learning and for guiding practical decisions. We understand the articulation can be improved, and will do so. Below we provide further discussion for the two central notions: $\eta$-finiteness and robustness.
>
> **$\eta$-finitess.** We have developed etafiniteness primarily as a notion for defining our version of the finite-precision framework, while being a more general and easily analisable property. Note, that $\eta$-finiteness is not necessarily a desirable or undesirable property for a model. Nonetheless, it could be used to analyse the geometry of the state space of trained models, e.g., the number of meaningful “states” a learned model seems to have. For example, using the same approach as was used to obtain Figure 1 (page 8 main body), we can examine how the state sequence of the model looks when processing an input. We have found that for a trained 1 layer Mamba on the FlipFlop task the point clouds for the 1 and 0 states generally form two distinct, linearly separated regions, when looking at the PCA of the inner state channels. This comes with the caveat that not all of the channels show such separation, and sometimes none have perfect linear separability, when looking at the 2d PCA plot. Nonetheless, we can observe that the inner state exhibits “etafiniteness” with this very simple approach. We will add such a visualisation to the supplementary material, to better visualise the connection to actual models behaviour.
>
> **Robustness.** Robustness gives us a way to further connect this notion to concrete finite precision systems, without any simplifying assumptions about floating point arithmetic. We also argue that it is a desirable stability property, separately from the finite precision framework. As for practical applications of robustness, we mainly envision it driving design decisions behind model architectures and training algorithms. For example, the inherent non-robustness of linear RNNs suggests that the solutions that may be learned for the model's parametrisation are very sensitive, especially when it comes to length-generalisation abilities. When it comes to determining what state components a trained model has and whether the state transitions display robustness, we have not pursued this direction, since our work focuses on expressivity. However, the ability to do so could give valuable insight into verifying length-generalisation abilities.
>
> ## Further Concerns
> > The paper is probably too dense for the current format (there are 50 pages of supplementary material, which I could not check): while it reads relatively well given the amount of definitions and concepts, the body of the paper would have benefited from illustrations of the main concepts (following some of the figures from the supplementary).
>
> We plan on adding figures to the main body, to build more visual intuition for the concepts and proof techniques presented. We will add improved versions of the figures present in the supplementary material, as well as new ones. The existing figures we will include in the main body are:
>
> * Figure 3 (page 17 supplementary), visualising correspondence  between the continuous $\eta$-finite dynamics and the canonical semiautomaton. The last transition will be changed to land in the same component.
>
> * Figure 5 (page 25 supplementary), visualising a concrete datatype covering of a robust dynamics. To the right, we wil merge Figure 4 (page 22 supplementary) together with a symbolic definition of robustness.
>
> * Figure 6 (page 30 supplementary), visualising aperiodicity and the key argument behind Theorem 4
>
> * Figure 1 (main body page 8) visualising empirical behaviour of a trained Mamba model on the FlipFlop task. To the right we will add a diagram of the FlipFlop transitions.
>
> The new figure we will include in the main body is:
>
> * A figure visualising transitions of a GCS together with the hyperplanes which piecewise separate the state space. To the right we will visualise the key argument behind Theorem 9.
>
> We will also add figures visualising the constructions in the appendix. Any further suggestions for changes to the figures, or new figures to be added to the appendices, would be greatly appreciated.
>
> >l87: I might be missing a point of definition but how can a continuous function have a finite codomain?
>
> For a trivial example, any function $X\to \set{y_0}$ is continuous–since it is constant, and $\set{y_0}$ is finite. Alternatively, consider $X_1= [0,1]$ and $X_2=[2,3]$. Similarly, a function on $X_1\cup X_2$ mapping $X_1$ to $1$ and $X_2$ to $2$ is continuous.
>
> The key point is that a continuous function can act independently on the disconnected components of the domain.
>
> >l359: about "predicted by Th. 8", actually Th. 8 allows mamba to represent the language ("expressivity" / "can recognize") it but, as far as I understand, nothing is stated about the learnability (the fact that mamba will actually manage to learn the parameters)
>
> The reviewer is right, the wording is off here. We will change this to more accurately state that  in practice Mamba learns star-free tasks, with length-generalisation—consistent with its expressivity described by Theorem 8.

---

> > ### Comment · Reviewer_sV21 · 2025-08-04
> >
> > I appreciate the reviewers answers and discussions.
> > I will keep my weak-accept score (based on the fact the the paper needs much restructuration) which I believe won't hurt the paper decision.
> > I encourage the authors to be very careful in the process of making the final version of the manuscript.
> >
> > NB: For the continuous / finite codomain, I wrongly (at that point) had in mind functions with convex domain. So the domain in key (and hence the PCA separability etc). While revising your manuscript, please consider this misunderstanding of mine and possible clarifications.

---

### Official Review · Reviewer_RX1U · 2025-07-04

**Clarity:** 3
**Significance:** 4
**Originality:** 4
**Rating:** 5
**Confidence:** 2

**Summary:**

This paper proposes Metric Automata Theory (MAT), a novel theoretical framework that generalises automata theory to continuous dynamical systems and allows the study of expressivity in both finite-precision and beyond finite-precision for a variety of model families. Under MAT, the authors show that xLSTMs can recognize all star-free regular languages, while SSMs with non-negative eigenvalues cannot, emphasising the importance of non-linear recurrences in expressivity. Finally, the authors demonstrate that Mamba with a piecewise-linearly separable state space can approximate star-free languages with some length extrapolation capabilities, however still fails on tasks such as Parity.

**Questions:**

1. Can Metric Automata Theory be used to also study the expressivity of RNNs beyond star-free languages, such as context-free ones? In lines 382-388 you briefly discuss this.

**Ethical Concerns:**

["NO or VERY MINOR ethics concerns only"]

**Final Justification:**

I think this is a relevant paper which will be valuable towards developing more expressive recurrent models in the future. I will keep my acceptance score.

**Limitations:**

The authors do not explicitly discuss the limitations of their theoretical framework, but I would encourage them to do so.

**Paper Formatting Concerns:**

No formatting issues.

**Quality:**

4

**Strengths And Weaknesses:**

**Strengths:**
- The paper provides a novel and elegant generalization of Automata Theory to continuous systems, offering a unified framework for analyzing the expressivity and robustness of several recurrent models. Under this framework, the authors prove that non-linear models, such as xLSTM, can recognize all star-free regular languages, whilst linear models cannot. I find these results very relevant with implications on future research in designing recurrent models.
- Afterwards the authors extend their framework via the notion of Geometrically-Constrained Systems, which addresses empirical phenomena like length generalization, and explains why SSMs with Mamba parameterization cannot learn alternating functions such as Parity.
- The theoretical claims are backed up by empirical evaluations from the authors, as well as empirical results from related work, further emphasizing the validity of the theoretical framework.
- The paper is very well written and contains an extensive supplementary material with the necessary background and further details on their theoretical results. The authors also review the related work properly, relating it to their results as well.

**Weaknesses:** (Minor)
- The paper briefly mentions the implications of robustness for training but does not deeply explore how the framework could guide the design of training algorithms or architectures. I would be curious to know the authors' opinion on this point.
- The empirical evaluation is limited to a specific task and SSM model. Otherwise, the authors refer to results from previous work (which is totally fine). I would be interested to see the results using xLSTM on star-free languages to back up the theory.

Overall, I found the work that the authors have done in this paper really impressive and even though my background limits me in deeply understanding some of the theory and concepts introduced in this submission, I was able to grasp them in high-level and understand the implications it can have in the field, especially in designing more expressive models in the future. To this end, I recommend acceptance.

---

> ### Author Rebuttal · Authors · 2025-07-30
>
> Thank you for your review of our work. We greatly appreciate your recognition of our work as novel and highly relevant to the field and future model design.
>
> We address your question, and then concerns raised in your review in order of appearance, and we are keen to hear your further thoughts and questions.
>
> ## Question
> > Can Metric Automata Theory be used to also study the expressivity of RNNs beyond star-free languages, such as context-free ones?
>
> Metric Automata Theory can certainly be used to study RNNs beyond star-free languages, and as a matter of fact beyond regular languages.
>
> * First, the Geometrically-constrained systems (GCS) framework we present showcases that MAT is not limited to the finite-precision setup. Specifically, Construction 1 (line 330) shows a system implementing an unbounded counter with infinitely many states, which recognizes the context-free language where the number of $a$s equals the number of $b$s.
>
> * In lines 382-388,    we question the ability  of Mamba to implement “counter-like” functions. Although we have not developed a formal proof of this claim yet, our hypothesis is that Mamba transitions being contracting—as defined in lines 295-299—prevents it from realising such languages. Note that the dynamics in Construction 1 are impossible for Mamba, since they utilises the 1-d identity gate on state $n$.
>
> * We are planning on formally investigating whether this is the case in future work using the GCS framework, as we believe it is the right tool for explaining length-generalisation abilities of SSMs.
>
> ## Further Concerns
>
> > The paper briefly mentions the implications of robustness for training but does not deeply explore how the framework could guide the design of training algorithms or architectures. I would be curious to know the authors' opinion on this point.
>
> As we will further discuss in the Limitations response, the notion of Robustness allows us to draw indirect conclusions about learning. In our opinion it is an interesting avenue of future experiments.
>
> * On the architecture side, it motivates the use of non-linear activations and indicates potential issues involving SSM layers. xLSTM successfully combines linear and non-linear nodes. However in the original xLSTM paper [Beck et al., 2024] Chomsky hierarchy results reported show that the mixed architecture actually significantly underperforms the non-linear only architecture—see Figure 4, Cycle Nav, Mod Arithmetic w/o Brackets, Mod Arithmetic w Brackets, Solve Equation. We suspect that SSM layers are trickier to train because of lack of robust state tracking parametrisations.
>
>
> * Moving onto training algorithms, during our (limited) evaluations we noticed that on the FlipFlop task Mamba tends to converge slowly on the out of sample validation set, while sLSTM tends to converge suddenly. Additionally, Mamba would tend to move away from the local solution and dramatically decrease validation accuracy if no early stopping was in place. Again, we suspect that this is due to the linear RNN solution being unstable under perturbation of parameters, and the non-linear solution being robust according to our definition and thus stable under perturbation of parameters. However to definitively attribute training behaviour of these models to robustness phenomena would require thorough experiments and considering hyperparameters of the used algorithms. We think this is best left for future work.
>
> > I would be interested to see the results using xLSTM on star-free languages to back up the theory.
>
> We agree it would be valuable to explicitly compare performance of xLSTM with Mamba on star-free tasks. We can expect both to be able to model star-free tasks based on their expressive capacity, the remaining question being how well they can attain these solutions via training. Nonetheless, we can look at existing experiments with (the classical) LSTM.
>
> * Bhattamishra suite results reported in [Bhattamishra et al., 2020] for the classic LSTM architecture, show that it achieves perfect accuracy on star-free tasks, both on in-distribution and out-distribution sequence lengths (Table 9). Meanwhile, the suite results reported in [Sarrof et al., 2024] for Mamba are the same, except for the D_12 task, where the in-distribution/out-distribution accuracy was 93.65/93.35 for 1 layer, 99.9/95.55 for 2 layers and  99.99/99.85 for 3 layers. (Table 3).
>
> * As for the  [Liu et al., 2023] FlipFlop task, in Section 4, paragraph FlipFlop [Sarrof et al., 2024] notes that Mamba converged in 3 times as many steps as LSTM, as reported by  [Liu et al., 2023].
>
> Again, we think that further experiments are a valuable direction for future work, to further explore advantages of LSTM architecture on star-free tasks.
>
> > The authors do not explicitly discuss the limitations of their theoretical framework, but I would encourage them to do so.
>
> Based on feedback, we intend to add a separate limitations section and future work section in the main body. The main points we will discuss in the limitations section are the following:
>
> * __Use of the dynamical system formalism:__ this approach yields an elegant and general framework and enables powerful techniques of analysis. However, it comes with a few  draw-backs. Firstly, requiring continuity throughout means that the main work of assigning meaning to the states is done in selecting the state space $X$. Further, the dynamics $f$ needs to have codomain $X$. This can make verifying constructions tedious. In the context of learning parameters for $f$, this also means that as the parameters vary, the state space must change accordingly. Hence, analysing a parametrised model requires considering a family of dynamical systems, which could be further developed as a separate framework within MAT. Nonetheless, the current formulation of MAT allows for indirect analysis of learning stability, via the notion of robustness (as discussed in Section 3.2 on robustness).
> * __Focus on unbounded length expressivity:__ Most of the framework and techniques presented, aside from robustness, study asymptotic behaviour of RNNs. We do not aim to determine e.g. how the number of $\eta$-components or layers needed grows with input length. Similarly, Krohn-Rhodes decomposition techniques are not suitable for e.g., determining the minimal number of components needed.
> * __Focus on state-space transformations:__ Our framework, and indeed Algebraic Automata Theory, does not directly model the input and output function complexity in the network, which also influences the expressive capacity of the model. We can still indirectly control what transformations are allowed in the network by imposing additional constraints aside from continuity, e.g., like in the geometrically-constrained setting. When we consider cascade constructions, generally connecting functions implementable by MLPs suffice, thanks to separability of components.
>
> We would like to note that, in our view, points related to the limitations of Algebraic Automata Theory can __potentially motivate a renewed development of the field__, in conjunction with modern expressivity studies. We plan on including a more detailed comparison with alternative frameworks, mentioned in the Related Work section, in the appendix.
>
> ## References
> * Maximilian Beck, Korbinian Pöppel, Markus Spanring, Andreas Auer, Oleksandra Prudnikova, Michael Kopp, Günter Klambauer, Johannes Brandstetter, and Sepp Hochreiter. xLSTM: Extended long short-term memory. In Advances in Neural Information Processing Systems 38: Annual Conference on Neural Information Processing Systems (NeurIPS), 2024.
>
> * S. Bhattamishra, K. Ahuja, and N. Goyal. On the ability and limitations of transformers to recognize
> formal languages. In Proceedings of the 2020 Conference on Empirical Methods in Natural
> Language Processing (EMNLP), pages 7096–7116, 2020.
>
> * Yash Sarrof, Yana Veitsman, and Michael Hahn. The expressive capacity of State Space Models: A formal language perspective. In Proceedings of the Thirty-Eighth Annual Conference on Neural Information Processing Systems (NeurIPS), 2024.
>
> * Bingbin Liu, Jordan T. Ash, Surbhi Goel, Akshay Krishnamurthy, and Cyril Zhang. Exposing attention glitches with flip-flop language modeling. In Thirty-seventh Conference on Neural Information Processing Systems, 2023

---

> > ### Comment · Reviewer_RX1U · 2025-08-04
> >
> > Thank you for your detailed response. I think this is a relevant paper which will be valuable towards developing more expressive recurrent models in the future. I will keep my acceptance score.

---

### Comment · Area_Chair_djHf · 2025-08-03
**author-reviewer discussion**

Dear reviewers,

The rebuttal provided by the authors is now available.
It is now time to read authors' answers and check if your questions/issues were addressed and possibly engage discussion to ask for more information as soon as possible.
I also encourage you to read the other reviews to see the point of view of the other reviewers.

Thanks to reviewer MndJ who already provided a feedback to the authors.

Many thanks for your participation.

Best regards,
AC.

---

### Note · Authors · 2025-08-15

We sincerely thank the reviewers for engaging with our work, for in-depth reviews and for valuable feedback. We are glad that the reviewers found our work novel and elegant (rev. RX1U), interesting (rev. sV21), insightful and broadly applicable (rev. MndJ) and with the potential to become unifying to RNN studies (rev. ycc4).

The feedback has helped us in modifying the main body, to improve presentation and to address the reviewers’ points. The main changes we will introduce in the final version are the following:
* We reduce the content of Section 2 Preliminaries, in favour of giving more details on the proofs and techniques used in Section 4.
* Additionally we add figures to the main body, as outlined in the rebuttal to reviewers sV21 and ycc4, to build more visual intuition for the concepts and proof techniques presented.
* We include dedicated sections on Limitations and Future Work, containing the points discussed in rebuttals to reviewers RX1U and MndJ.

We also modify sections introducing novel notions to better articulate their applicability and address other points raised during the discussions.

Once again, thank you for the time and effort spent reviewing our work.

---

### Decision · Program_Chairs · 2025-09-17

**Decision:**

Accept (poster)

**Comment:**

This theoretical paper introduces a generalization of classic automata theory to continuous dynamical systems named Metric Automata Theory.  This theory provides a unified way  to study the expressiveness of families of models based on Recurrent Neural Networks including xLSTMs and State Space Models. The paper provides results studying the capacity of some models to recognize of approximate star-free regular languages. An important aspect of the framework is its capacity of studying expressivity in both finite-precision and beyond finite precision.

Based on the reviews, the following strengths have been identified:
-Novel and elegant generalization of automata theory.
-Strong and novel results.
-Interesting discrete viewpoint on continuous dynamical systems.
-Helpful to understand the behavior of RNNs for language recognition.

The following weaknesses have also been identified:
-paper very dense with 50pages of appendix, sometimes hard to follow.
-lack of illustrations and explanations.
-the implication of the results for learning was slightly discussed but not really explored.
-rather limited experimental evaluation.

During rebuttal, authors have provided multiple answers to the different remarks raised and their answers were globally satisfying for most of the reviewers. During final evaluation and discussion:
-Reviewer RX1U thinks that this is relevant paper and proposes acceptance.
-Reviewer sV21 was favorable to acceptance provided that a restructuring of the paper is done to improve its exposition.
-Reviewer MndJ was convinced by the author's claim regarding the importance of the generic connection to automata theory, he appreciates the scope and strength of the contribution. He is also highly favorable to the inclusion of a limitation section and requires to improve the preliminaries section, to improve the presentation of the paper.
-Reviewer ycc4 appreciated the solidity of the contribution and the perspective of unification theory for RNN studies. He also required to rephrase the article so that ideas for proof techniques appear in the body of the text and possibly to illustrate key concepts to make the article more easily recognizable and digestible for readers.

Overall, 3 reviewers propose direct acceptance and the other one leans toward weak acceptance which shows a clear consensus towards acceptance.
If the weaknesses raised were still present, they are outweighed by the correctness, originality, and usefulness of the results for developing more expressive recurrent models.
I read the paper and I agree that the contribution is solid providing a nice supporting theory for studying recurrent models.

I propose then accept for this paper.
Nevertheless, I urge the authors to take into account the requests made by the reviewers to improve the presentation of the paper, notably those made by reviewers sV21, MndJ and ycc4. The authors final remarks make me confident on the fact that the authors will update their paper accordingly.